# T cell responses to SARS-CoV-2 spike cross-recognize Omicron

Roanne Keeton[1,2], Marius B. Tincho[1,2], Amkele Ngomti[1,2], Richard Baguma[1,2], Ntombi Benede[1,2], Akiko Suzuki[1,2], Khadija Khan[3,4], Sandile Cele[3,4], Mallory Bernstein[3,4], Farina Karim[3,4], Sharon V. Madzorera[5,6], Thandeka Moyo-Gwete[5,6], Mathilda Mennen[7], Sango Skelem[7], Marguerite Adriaanse[7], Daniel Mutithu[7], Olukayode Aremu[7], Cari Stek[1,7], Elsa du Bruyn[1,7], Mieke A. Van Der Mescht[8], Zelda de Beer[9], Talita R. de Villiers[9], Annie Bodenstein[9], Gretha van den Berg[9], Adriano Mendes[10], Amy Strydom[10], Marietjie Venter[10], Jennifer Giandhari[11], Yeshnee Naidoo[11], Sureshnee Pillay[11], Houriiyah Tegally[11], Alba Grifoni[12], Daniela Weiskopf[12], Alessandro Sette[12,13], Robert J. Wilkinson[1,7,14,15,16], Tulio de Oliveira[11,17], Linda-Gail Bekker[1,7,18], Glenda Gray[19], Veronica Ueckermann[20], Theresa Rossouw[8], Michael T. Boswell[20], Jinal N. Bhiman[5,6], Penny L. Moore[1,5,6,21], Alex Sigal[3,4,22], Ntobeko A. B. Ntusi[1,7,14,23], Wendy A. Burgers[1,2,14,24 ✉] & Catherine Riou[1,2,14,24 ✉]

The SARS-CoV-2 Omicron variant (B.1.1.529) has multiple spike protein mutations[1,2] that contribute to viral escape from antibody neutralization[3–6] and reduce vaccine protection from infection[7,8]. The extent to which other components of the adaptive response such as T cells may still target Omicron and contribute to protection from severe outcomes is unknown. Here we assessed the ability of T cells to react to Omicron spike protein in participants who were vaccinated with Ad26.CoV2.S or BNT162b2, or unvaccinated convalescent COVID-19 patients ($n = 70$). Between 70% and 80% of the CD4$^+$ and CD8$^+$ T cell response to spike was maintained across study groups. Moreover, the magnitude of Omicron cross-reactive T cells was similar for Beta (B.1.351) and Delta (B.1.617.2) variants, despite Omicron harbouring considerably more mutations. In patients who were hospitalized with Omicron infections ($n = 19$), there were comparable T cell responses to ancestral spike, nucleocapsid and membrane proteins to those in patients hospitalized in previous waves dominated by the ancestral, Beta or Delta variants ($n = 49$). Thus, despite extensive mutations and reduced susceptibility to neutralizing antibodies of Omicron, the majority of T cell responses induced by vaccination or infection cross-recognize the variant. It remains to be determined whether well-preserved T cell immunity to Omicron contributes to protection from severe COVID-19 and is linked to early clinical observations from South Africa and elsewhere[9–12].

The newest SARS-CoV-2 variant of concern, designated Omicron[1], was first described on 26 November 2021 from sequences from Botswana, Hong Kong and South Africa[2]. Omicron is responsible for the current surge of infections in South Africa, and is becoming globally dominant. The variant has more than 30 mutations in the spike protein compared with the ancestral strain and a substantial ability to evade the neutralizing

antibody response[3–6]. This is associated with greater capacity for reinfection[13], as well as lower early estimates of vaccine effectiveness against symptomatic disease[7,8]. SARS-CoV-2-specific T cells have a role in modulating COVID-19 severity. A study of acute COVID-19 using combined measurement of CD4$^+$ T cells, CD8$^+$ T cells and neutralizing antibodies has suggested that co-ordination of these three arms of the adaptive

[1]Institute of Infectious Disease and Molecular Medicine, University of Cape Town, Observatory, Cape Town, South Africa. [2]Division of Medical Virology, Department of Pathology, University of Cape Town; Observatory, Cape Town, South Africa. [3]Africa Health Research Institute, Durban, South Africa. [4]School of Laboratory Medicine and Medical Sciences, University of KwaZulu-Natal, Durban, South Africa. [5]National Institute for Communicable Diseases of the National Health Laboratory Service, Johannesburg, South Africa. [6]SA MRC Antibody Immunity Research Unit, School of Pathology, Faculty of Health Sciences, University of the Witwatersrand, Johannesburg, South Africa. [7]Department of Medicine, University of Cape Town and Groote Schuur Hospital; Observatory, Cape Town, South Africa. [8]Department of Immunology, University of Pretoria, Pretoria, South Africa. [9]Tshwane District Hospital, Tshwane, South Africa. [10]Centre for Viral Zoonoses, Department of Medical Virology, University of Pretoria, Pretoria, South Africa. [11]KwaZulu-Natal Research Innovation and Sequencing Platform, University of KwaZulu-Natal, Durban, South Africa. [12]Center for Infectious Disease and Vaccine Research, La Jolla Institute for Immunology, La Jolla, CA, USA. [13]Department of Medicine, Division of Infectious Diseases and Global Public Health, University of California, San Diego (UCSD), La Jolla, CA, USA. [14]Wellcome Centre for Infectious Diseases Research in Africa, University of Cape Town, Observatory, Cape Town, South Africa. [15]Department of Infectious Diseases, Imperial College London, London, UK. [16]The Francis Crick Institute, London, UK. [17]Centre for Epidemic Response and Innovation, Stellenbosch University, Stellenbosch, South Africa. [18]Desmond Tutu HIV Centre, University of Cape Town, Cape Town, South Africa. [19]South African Medical Research Council, Cape Town, South Africa. [20]Department of Internal Medicine, University of Pretoria and Steve Biko Academic Hospital, Pretoria, South Africa. [21]Centre for the AIDS Programme of Research in South Africa, Durban, South Africa. [22]Max Planck Institute for Infection Biology, Berlin, Germany. [23]Cape Heart Institute, Faculty of Health Sciences, University of Cape Town; Observatory, Cape Town, South Africa. [24]These authors contributed equally: Wendy A. Burgers, Catherine Riou. ✉e-mail: wendy.burgers@uct.ac.za; cr.riou@uct.ac.za

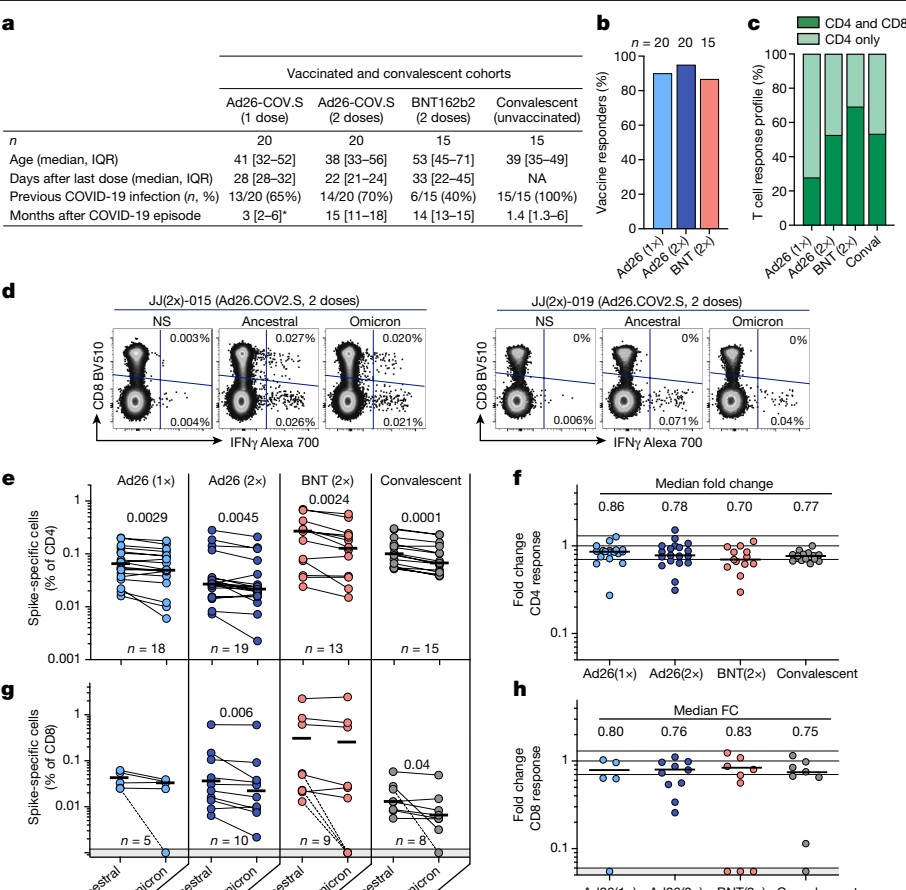

**Fig. 1 | T cell response to the ancestral and Omicron SARS-CoV-2 spike after vaccination and in unvaccinated COVID-19 convalescent patients.** **a**, Clinical characteristics of the study groups. *Data from after Covid-19 infection were available for only 6 out of the 13 participants who received one dose of Ad26.COV2.S. **b**, The proportion of participants exhibiting an ancestral spike-specific CD4+ T cell response after vaccination with one or two doses of Ad26.COV2.S or two doses of BNT162b2. **c**, The profile of the ancestral spike-specific T cell response in vaccinees and convalescent (conval) individuals. **d**, Representative examples of IFN-γ production in response to ancestral and Omicron spike in two individuals who received two doses of

Ad26.COV2.S. **e**, **g**, Frequency of spike-specific CD4+ (**e**) and CD8+ T cells (**g**) producing any of the measured cytokines (IFN-γ, IL-2 or TNF) in response to peptide pools representing ancestral and Omicron spike protein. Bars represent the median of responders. Differences between SARS-CoV-2 variants were calculated using a two-tailed Wilcoxon paired test. **f**, **h**, Fold change in the frequency of spike-specific CD4+ (**f**) and CD8+ T cells (**h**) between ancestral and Omicron spike responses. Bars represent medians. No significant differences were observed between groups using a Kruskal–Wallis test with Dunn´s multiple comparisons post test. The number of participants included in each analysis is indicated on the graphs.

response leads to lower disease severity[14]. A greater CD8+ T cell response in blood and highly clonally expanded CD8+ T cells in bronchoalveolar lavage were observed in convalescent patients who experienced mild or moderate disease compared with severe disease[15,16], and CD8+ T cells provided partial protective immunity in the context of suboptimal antibody titres in macaques[17]. In this study, we include 138 participants grouped according to their vaccination and COVID-19 status, in order to determine whether T cells generated in response to vaccination or previous SARS-CoV-2 infection could cross-recognize Omicron and to define the profile of T cell responses in Omicron-infected patients compared with those infected with other variants of concern.

## T cell cross-reactivity to Omicron

We examined T cell responses in participants who had received one or two doses of the Ad26.COV2.S vaccine (Johnson and Johnson/Janssen, $n = 20$ per group), two doses of the BNT162b2 mRNA vaccine (Pfizer–BioNTech, $n = 15$ per group), or who had recovered from infection ($n = 15$ per group) (Fig 1a, Extended Data Table 1a, b). Convalescent donors were examined a median of 1.4 months (interquartile range (IQR): 1.3–6 months) after mild or asymptomatic infection. More than

85% of vaccinees generated a T cell response to vaccination, measured 22–32 days after the last dose (Fig. 1b). Both vaccination and infection induced spike-specific CD4+ T cell responses, whereas a CD8 response was less consistently detected (Fig. 1c). We measured cytokine production (IFN-γ, IL-2 and TNF) by intracellular cytokine staining in response to peptide pools covering the full Wuhan-1 spike protein (ancestral) and the Omicron spike (Fig. 1d, Extended Data Fig. 1a).

The levels of CD4+ T cell responses to Omicron spike were consistently and significantly lower than those responsive to ancestral spike in all groups tested (Fig. 1e). This translated to a median decrease of 14–30% of the CD4 response to Omicron, as demonstrated by fold change (Fig. 1f). Similar results were observed for the CD8+ T cell response (Fig. 1g, h): vaccinees who had received two doses of Ad26.COV2.S and convalescent donors exhibited a significantly lower frequency of Omicron spike-specific CD8+ T cells, although the other groups did not. There was a median reduction of 17–25% in the CD8 response to Omicron compared with the ancestral virus. Of note, a fraction of responders (5 out of 32, 15%) exhibited a loss of CD8+ T cell recognition of Omicron (Figure 1g, Extended Data Fig. 1b), probably reflecting specific human leukocyte antigen (HLA) molecules being adversely affected by mutations in particular CD8 epitopes[18].

In parallel, we measured the neutralizing activity against ancestral and Omicron spike from the plasma of the same participants who received BNT162b2 ($n = 10$) or two doses of Ad26.COV.S ($n = 19$) (Extended Data Fig. 2). As previously described[3,5,6], Omicron escapes the SARS-CoV-2 neutralizing antibodies generated after BNT162b2 vaccination. Here we present neutralizing responses to Omicron after two doses of Ad26.COV2.S (Extended Data Fig. 2b), demonstrating diminished neutralization capacity compared with D614G ancestral virus and the Beta variant. Comparison of the fold change in T cell responses and neutralizing antibodies targeting ancestral or Omicron spike further emphasizes the preservation of the T cell response, even when neutralization is severely reduced.

Mutations in variant epitopes have the potential to affect the functional capacity of cells[19]. Thus, we compared the polyfunctional profiles of T cells in vaccinees and convalescent individuals and demonstrate similar capacities for cytokine co-expression across all groups for both ancestral and Omicron-specific T cells (Extended Data Figs. 3a, b, 4a, b). Notably, there were also no differences in the polyfunctional profiles between ancestral and Omicron spike for either CD4+ or CD8+ T cells (Extended Data Figs. 3c, 4c), indicating the absence of a functional deficit in cross-reactive Omicron T cell responses. We also compared Omicron spike responses to other variants of concern in Ad26.CoV2.S vaccinees, by testing spike peptide pools corresponding to the viral sequences of the Beta and Delta strains (Extended Data Fig. 5a). There were no significant differences in cross-reactive CD4+ and CD8+ T cell responses between Beta, Delta and Omicron (Extended Data Fig. 5b), with the exception of a greater decrease in the Omicron CD4 response compared with Beta in people who had received two doses of Ad26.COV2.S. Of note, whereas previous SARS-CoV-2 infection in vaccinees was associated with a higher frequency of spike-specific T cells (Extended Data Fig. 6a), it had no impact on Omicron cross-reactivity (Extended Data Fig. 6b). Overall, these results show that CD4+ and CD8+ T cell recognition of Omicron spike is largely preserved compared with the ancestral strain, and is similar to other variants of concern carrying fewer mutations.

## T cell response to different variants

The SARS-CoV-2 epidemic in South Africa has been characterized by four virologically distinct infection waves (Fig. 2b). This enabled us to compare T cell responses in patients infected with SARS-CoV-2 during the current fourth epidemic wave, dominated by Omicron, with those infected in previous waves dominated by ancestral (wave 1, $n = 17$), Beta (wave 2, $n = 16$) and Delta (wave 3, $n = 16$) variants (Fig. 2a). In addition to extensive mutations in spike, Omicron has 20 additional mutations in other proteins which could also result in T cell escape. Therefore, we measured the frequency of CD4+ and CD8+ T cells reactive towards ancestral spike, nucleocapsid and membrane proteins, all major targets of the T cell response[20]. We studied SARS-CoV-2-infected patients who were hospitalized with COVID-19 (Fig. 2a). These recently hospitalized patients, recruited between 1 December 2021 and 15 December 2021 ($n = 19$), had no previous history of COVID-19 and were unvaccinated. Omicron infection was inferred by spike gene target failure (SGTF)[21] in nine of these patients. Although swabs were unavailable for the remainder, with Omicron accounting for more than 90% of sequences from South Africa at the time of recruitment and 98% in Tshwane from where the samples originated (Fig. 2b), there was a high probability of Omicron infection in all of these patients.

Despite differences in age, disease severity and co-morbidities across the infection waves (Fig. 2a, Extended Data Table 1c), T cell responses directed at spike, nucleocapsid and membrane proteins in wave 4 patients were of similar magnitude as those in patients infected with other SARS-CoV-2 variants in previous waves (Fig. 2c, d). The frequency of responders also did not differ markedly across the waves. Of note, we did not find any association between the absence of detectable CD4+ T cell responses and the time post COVID-19 diagnosis or disease severity. Furthermore, the magnitude of Omicron spike-specific CD4 responses mounted by those infected in wave 4 was highly comparable to those against ancestral spike (Fig. 2e), suggesting that the CD4 responses mostly target conserved epitopes in spike. Using data from the Immune Epitope Database (https://www.iedb.org), we assessed the frequency of T cell recognition of experimentally-confirmed epitopes spanning the entire spike protein. Data show that Omicron spike mutations occur in regions poorly targeted by CD4+ T cells, but are more common in regions frequently targeted by CD8+ T cells (Extended Data Fig. 7).

To gain deeper insight into the recognition of variable spike epitopes by CD8+ T cells, we also performed in silico analysis to define predicted HLA class I restriction for Omicron variable epitopes (Extended Data Table 2). Six confirmed spike epitopes containing Omicron mutations (A67V/Δ69–70, G142D/Δ143–145, S373P, S375F, D614G, P681H and N764K) would be detrimentally affected for binding to specific class I alleles, four of which were located at positions that recorded a frequency of recognition greater than 10%. However, we also found another seven confirmed epitopes that contained Omicron mutations (T95I, S371L/S373P/S375F, K417N, G446S, Q493R, N764K and L981F) but had no effect on class I binding compared with the ancestral sequence, five of which were located at positions with a frequency of recognition greater than 10%. Overall, this suggests that although some Omicron mutations may mediate escape from specific HLA-restricted CD8+ T cells, not all mutations appear to have an impact on class I binding.

## Discussion

Here we measured the ability of individuals to cross-recognize Omicron spike following vaccination, prior infection or both. We also studied unvaccinated individuals with no history of previous infection, whose first encounter with spike was with the Omicron variant. We demonstrate that vaccination and infection induce robust CD4+ and CD8+ T cell responses that largely cross-react with Omicron, consistent with recent work from our laboratory and others on limited T cell escape by Beta, Delta and other variants[22–24]. Despite extensive neutralization escape against Omicron[5], 70–80% of the T cell response is cross-reactive. In contrast to neutralizing antibody epitopes, T cell epitopes are abundant and located across the entire spike protein[20], suggesting that the majority of SARS-CoV-2 spike-specific T cell responses are directed against conserved epitopes and that SARS-CoV-2 viral evasion from T cells may be limited.

Of note, Omicron mutations appear to abolish CD8+ T cell recognition in 5 out of 32 participants (15%), in agreement with a recent report[25]. This loss of cross-reactive CD8+ T cell responses could have pathological consequences for some individuals. Further analyses are required to define specific HLA class I profiles and epitopes linked to loss of T cell responses.

T cells are crucial components of the antiviral immune response. Although they do not prevent infection, CD4+ T cells are indispensable for the generation of protective antibody responses and supporting the maturation of CD8+ T cells. Hence, given the ability of variants of concern to escape neutralization, the generation and maintenance of robust SARS-CoV-2-specific T cell responses could contribute to long-term vaccine efficacy against severe disease. Several studies have reported a waning of the neutralizing response after vaccination or infection[26–28]. However, humoral responses can be enhanced upon booster vaccination, improving Omicron neutralization[3,6,29,30]. Vaccine- and infection-induced T cell responses also decay after antigen clearance[31,32], but SARS-CoV-2-specific CD8+ T cells exhibit the hallmarks of long-lived cells[33], and T cell responses to SARS-CoV-1 infection were detectable 17 years later[34]. The longer-term durability of SARS-CoV-2-specific T cells and whether vaccine boosters can further enhance cellular immunity remain to be determined.

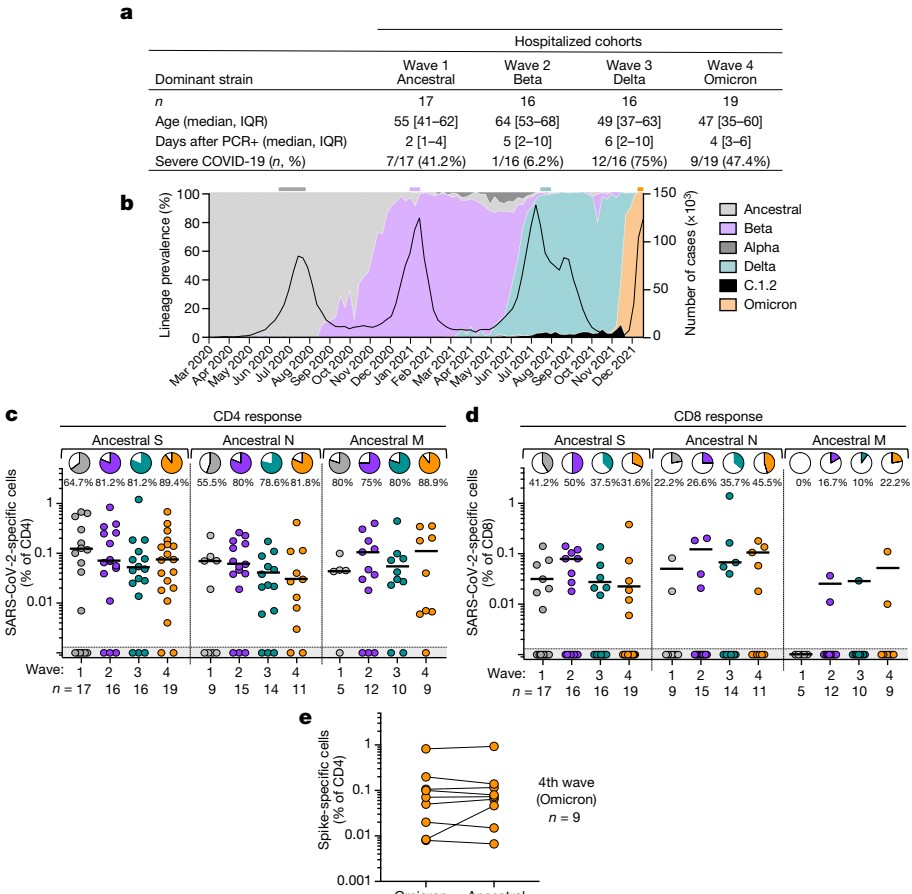

**Fig. 2 | T cell response to ancestral SARS-CoV-2 in unvaccinated hospitalized patients with COVID-19 who were infected with the ancestral, Beta, Delta or Omicron SARS-CoV-2 variants. a**, Clinical characteristics of the study groups. Severe disease was defined on the basis of oxygen therapy requirement according to the WHO ordinal scale scoring system (≥5; $O_2$ via high flow to extracorporeal membrane oxygenation). **b**, SARS-CoV-2 epidemiological dynamics in South Africa showing the prevalence of different SARS-CoV-2 strains (based on 24,762 sequences; left axis) and the number of COVID-19 cases (right axis). The bars on the top of the graph indicate the periods when samples were collected for each epidemic wave. **c**, **d**, Frequency

of SARS-CoV-2-specific CD4+ (**c**) and CD8+ T cells (**d**) producing any of the measured cytokines (IFN-γ, IL-2 or TNF) in response to ancestral SARS-CoV-2 spike (S), nucleocapsid (N) and membrane (M) peptide pools. Pies depict the proportion of participants exhibiting a detectable T cell response to each protein. **e**, Comparison of T cell response to ancestral or Omicron spike in Omicron-infected patients. Bars represent medians of responders. No significant differences were observed between antigens amongst responders using a Kruskal–Wallis test with Dunn's multiple comparisons post test. The number of participants included in each analysis is indicated on the graphs.

Despite the sharp increase in cases in South Africa in the current surge[35], this has not translated into the expected increase in hospitalization or deaths, compared with previous waves[12]. This uncoupling of caseloads and severe outcomes could be attributed to population immunity, including maintenance of cross-reactive T cell responses observed in our study and/or intrinsic differences in Omicron severity. South Africa has high levels of SARS-CoV-2 seropositivity, driven mainly by previous infection (estimated at more than 60%) and a modest proportion of vaccinated people[36] (40%). Emerging data hint at reduced intrinsic severity of Omicron, including reduced infection of lower airway cells[37,38]. The relative contribution of high levels of immunity and potential changes in intrinsic virulence on clinical outcomes are difficult to disentangle. Moreover, it remains to be determined whether the apparently milder outcomes at a population level will be observed in other contexts with different exposure histories and vaccination coverage, or whether the higher transmissibility of Omicron and the expected massive increase in cases in a short period will offset any gains. So far, immune correlates of protection from disease are not clearly defined and large-scale prospective studies would be necessary to evaluate correlates of protection and define the role of T cell responses in disease.

Our study had several limitations. We studied Omicron cross-reactivity of vaccine responses approximately one month after vaccination. Since T cell responses decline over time, the detection of continued cross-reactivity with variants over time will be related to the durability of the T cell response. Recall memory responses in vivo are likely to expand rapidly upon viral infection and contribute to limiting viral replication. We also focused on cytokine production by T helper 1 ($T_H$1) cells to quantify CD4+ and CD8+ T cell responses. Additional approaches such as the activation-induced marker assay may capture the cellular immune response in a more comprehensive manner[39]. The use of 15mer peptides will have underestimated SARS-CoV-2 specific CD8+ T cells, as 9mer or 10mer peptides are optimal for HLA class I binding, and it has been estimated that 15mer peptides capture 77% of the frequency of CD8+ T cells when compared with shorter peptides[40]. Moreover, the saturating concentration of peptides used in these studies may underestimate the effect of mutations on T cells. In addition, the use of peptides does not allow us to define the potential effect of mutations on antigen processing and presentation, thus underestimating the effect of Omicron mutations on T cell cross-recognition. Finally, confirmation of our results from cohorts in other geographical areas and exposure to other vaccines would offer further reassurance of

the maintenance of T cell responses against Omicron. Indeed, emerging data suggest this to be the case[25,41–45].

Overall, our data show that unlike neutralizing antibodies, the SARS-CoV-2 T cell responses generated upon vaccination or previous infection are highly cross-reactive with Omicron. Early reports emerging from South Africa, England and Scotland have reported a lower risk of hospitalization and severe disease compared with the previous Delta wave[9–12]. It remains to be defined whether cell-mediated immunity provides protection from severe disease and contributes to the apparent milder outcomes for Omicron. Moreover, the resilience of the T cell response demonstrated here also bodes well in the event that more highly mutated variants emerge in future.

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

## Methods

### Human participants

At total of 138 participants were included in this study and grouped according to their vaccination and COVID-19 status. Participants were selected based on availability of peripheral blood mononuclear cells (PBMC) and clinical data were recorded by trained clinicians using Red-Cap (v9.5.36). The study was approved by the University of Cape Town Human Research Ethics Committee (ref: HREC 190/2020, 207/2020 and 209/2020) and the University of the Witwatersrand Human Research Ethics Committee (Medical) (ref. M210429 and M210752), the Biomedical Research Ethics Committee at the University of KwaZulu–Natal (ref. BREC/00001275/2020) and the University of Pretoria Health Sciences Research Ethics Committee (ref. 247/2020). Written informed consent was obtained from all participants.

**Participants vaccinated with Ad26.COV2.S (one or two doses) or BNT162b2 (two doses).** PBMC samples from 40 participants (20 who received one dose of Ad26.COV2.S vaccine and 20 who received two doses) were included in this study. These participants are enrolled in the Sisonke phase IIIb trial, an implementation trial of Ad26.COV2.S in healthcare workers. Recruitment took place at Groote Schuur Hospital (Cape Town, Western Cape, South Africa) between July 2020 and December 2021. Prior COVID-19 infection was recorded in 13 out of the 20 participants who had received one dose of the Ad26.COV2.S vaccine and in 14 out of 20 participants who had received two doses. Additionally, we also included samples from 15 participants vaccinated with two doses of BNT162b2 (Pfizer), enrolled in a prospective cohort study in KwaZulu Natal (South Africa). Prior COVID-19 infection was recorded for 6 out of 15 participants. The demographic and clinical characteristics of vaccinated participants are summarized in Extended Data Table 1a, with individual participant details presented in Extended Data Table 4.

**Convalescent COVID-19 participants.** COVID-19 convalescent volunteers (n = 15) were recruited from Groote Schuur Hospital in Cape Town (Western Cape, South Africa). Based on the reported date of infection, seven were probably infected with ancestral SARS-CoV-2 (before August 2020), whereas for the other 8, the infection date occurred in December 2020, suggesting an infection with the Beta variant. Samples were obtained between 19 January and 15 February 2021 before SARS-CoV-2 vaccination became available in South Africa. All had a documented positive SARS-CoV-2 PCR swab result or a positive SARS-CoV-2 nucleocapsid-specific antibody result (Roche Elecsys assay). The median time post positive test was 1.4 months, ranging from 1 to 7 months. The demographic and clinical characteristics of convalescent volunteers are summarized in Extended Data Table 1b, with individual participant details presented in Extended Data Table 3.

**Hospitalized COVID-19 patients.** Sixty-eight hospitalized COVID-19 patients were included in this study. These participants were grouped according to the time of their hospitalization, reflecting four distinct infection waves in South Africa, each dominated by a different SARS-CoV-2 strain (Fig. 2b). Wave 1, 2 and 3 participants were recruited from Groote Schuur Hospital in Cape Town (Western Cape, South Africa) and wave 4 patients were recruited from Groote Schuur Hospital and Tshwane District Hospital in Tshwane (Gauteng, South Africa). Wave 1 patients (n = 17) were enrolled between 11 June and 24 July 2020, at a time when ancestral (Wuhan-1 D614G)-related SARS-CoV-2 strains were circulating. No viral sequences are available for these patients, but we assumed that all were infected with a virus closely related to the ancestral virus, as sampling occurred almost three months before the emergence of the Beta variant in South Africa. Wave 2 patients (n = 16) were recruited between 31 December 2020 and 15 January 2021, when the Beta variant dominated. Viral sequences were available for 6 wave 2 participants, all of whom had confirmed Beta infection (GISAID accession numbers:

EPI_ISL_1040693, 1040658, 1040661, 1040685, 1040657 and 1040663). Wave 3 patients (n = 16) were recruited between 14 July and 21 July 2021. Wave 3 was dominated by the Delta variant. Viral sequences were available for 7 wave 3 participants, all of which were confirmed to be Delta infection (GISAID accession numbers: EPI_ISL_3506484, 3506367, 3957813, 3506504, 3506512 and 3506518). Wave 4 patients (n = 19) were recruited between 1 December and 15 December 2021. The SARS-CoV-2 Omicron variant was dominant during this current wave. Amongst those patients, nine had a Taqpath PCR test performed (Thermofisher), all of which were characterized by SGTF, highly suggestive of an Omicron infection. Although we did not have confirmation of Omicron for the remaining samples, they were recruited at a time when wave 4 was driven by Omicron infection (Fig. 2b; there was no concomitant Delta wave in South Africa as has occurred elsewhere), with the prevalence of Omicron in South Africa at the time of recruitment being over 90% by whole-genome sequencing (WGS). Moreover, in Tshwane, from where the remainder of the samples originated, Omicron was responsible for 98% of infections sequenced at the time of sampling (61 out of 62 samples sequenced).

All hospitalized patients from waves 1, 2 and 4 were unvaccinated at the time of sampling. Third wave participants with known vaccination status were all unvaccinated (n = 8), and the remainder (n = 8) had unknown vaccination status. Moreover, all hospitalized patients from wave 1, 2 and 4 had no clinical record of a previous symptomatic COVID-19 episode, apart from one Wave 4 participant with an unknown history. The majority of wave 3 patients had an unknown history of prior COVID-19. The demographic and clinical characteristics of hospitalized COVID-19 participants are summarized in Extended Data Table 1c, and individual patient clinical data are presented in Extended Data Table 3.

### SARS-CoV-2 spike, WGS and phylogenetic analysis

WGS of SARS-CoV-2 was performed from nasopharyngeal swabs. Sequencing was performed as previously described[2]. In brief, RNA was extracted on an automated Chemagic 360 instrument, using the CMG-1049 kit (Perkin Elmer). Libraries for WGS were prepared using either the Oxford Nanopore Midnight protocol with Rapid Barcoding or the Illumina COVIDseq Assay. The quality control checks on raw sequence data and the genome assembly were performed using Genome Detective 1.133 (https://www.genomedetective.com) which was updated for the accurate assembly and variant calling of tiled primer amplicon Illumina or Oxford Nanopore reads, and the Coronavirus Typing Tool. Phylogenetic classification of the genomes was done using the widespread dynamic lineage classification method from the PANGOLIN software suite (v1.2.106) (https://github.com/hCoV-2019/pangolin).

### Isolation of PBMC

Blood was collected in heparin tubes and processed within 4 h of collection. PBMC were isolated by density gradient sedimentation using Ficoll-Paque (Amersham Biosciences) as per the manufacturer's instructions and cryopreserved in freezing media consisting of heat-inactivated fetal bovine serum (FBS, Thermofisher Scientific) containing 10% DMSO and stored in liquid nitrogen until use.

### SARS-CoV-2 antigens

For T cell assays on hospitalized patients, we used commercially available peptide pools (15mer sequences with 11 amino acids of overlap) covering the full length of the Wuhan-1 SARS-CoV-2 nucleocapsid, membrane and near full-length spike proteins (PepTivator, Miltenyi Biotech). For spike, we combined (1) a pool of peptides (15-mer sequences with 11 amino acids overlap) covering the ancestral N-terminal S1 domain of SARS-CoV-2 spike (GenBank MN908947.3, Protein QHD43416.1) from amino acids 1 to 692, and (2) a pool of peptides (15-mer sequences with 11 amino acids overlap) covering the immunodominant sequence domains of the ancestral C-terminal S2 domain of SARS-CoV-2 (GenBank MN908947.3, Protein QHD43416.1) including the sequence domains

spanning residues 683–707, 741–770, 785–802 and 885–1273. Pools were resuspended in distilled water at a concentration of 50 µg ml⁻¹ and used at a final concentration of 1 µg ml⁻¹. To determine T cell responses to SARS-CoV-2 variants in vaccinated and convalescent volunteers, we used custom mega pools of peptides. These peptides (15-mers overlapping by 10 amino acids) spanned the entire spike protein corresponding to the ancestral Wuhan sequence (GenBank: MN908947), Beta (B.1.351; GISAID: EPI_ISL_736932), Delta SARS-CoV-2 variants (B.1.617.2; GISAID: EPI_ISL_2020950) or Omicron (B.1.1.529), carrying in the spike sequence all the 38 currently described mutations (A67V, H69del, V70del, T95I, G142D, V143del, Y144del, Y145del, S152W, N211del, L212I, ins214EPE, G339D, S371L, S373P, S375F, K417N, N440K, G446S, S477N, T478K, E484A, Q493R, G496S, Q498R, N501Y, Y505H, T547K, D614G, H655Y, N679K, P681H, N764K, D796Y, N856K, Q954H, N969K and L981F). In brief, peptides were synthesized as crude material (TC Peptide Lab). All individual peptides included in each mega pool are listed in Supplementary Table 1. All peptides were individually resuspended in dimethyl sulfoxide (DMSO) at a concentration of 10–20 mg ml⁻¹. Megapools for each antigen were created by pooling aliquots of these individual peptides in the respective SARS-CoV-2 spike sequences, followed by sequential lyophilization steps, and resuspension in DMSO at 1 mg ml⁻¹. There were 253 peptides in the ancestral, Beta and Delta variant pool, and 254 peptides in the Omicron pool. Pools were used at a final concentration of 1 µg ml⁻¹ with an equimolar DMSO concentration in the non-stimulated control.

### Cell stimulation and flow cytometry staining

Cryopreserved PBMC were thawed, washed and rested in RPMI 1640 containing 10% heat-inactivated FCS for 4 h prior to stimulation. PBMC were seeded in a 96-well V-bottom plate at approximately 2 × 10⁶ PBMC per well and stimulated with either the commercial ancestral SARS-CoV-2 spike (S), Nucleocapsid (N) or membrane protein (M) peptide pools (1 µg ml⁻¹) obtained from Miltenyi or custom spike mega pools corresponding to the ancestral (Wuhan-1), Beta, Delta or Omicron variants (1 µg ml⁻¹). All stimulations were performed in the presence of brefeldin A (10 µg ml⁻¹, Sigma-Aldrich) and co-stimulatory antibodies against CD28 (clone 28.2) and CD49d (clone L25) (1 µg ml⁻¹ each; BD Biosciences). As a negative control, PBMC were incubated with co-stimulatory antibodies, Brefeldin A and an equimolar amount of DMSO. After 16 h of stimulation, cells were washed, stained with LIVE/DEAD Fixable VIVID Stain (1/2,500, Invitrogen, Carlsbad, CA, USA) and subsequently surface stained with the following antibodies: CD14 Pac Blue (1/100, TuK4, Invitrogen Thermofisher Scientific), CD19 Pac Blue (1/100, SJ25-C1, Invitrogen Thermofisher Scientific), CD4 PERCP-Cy5.5 (1/100, L200, BD Biosciences), CD8 BV510 (1/100, RPA-8, Biolegend). Cells were then fixed and permeabilized using a Cytofix/Cyto perm buffer (BD Biosciences) and stained with CD3 BV650 (1/100, OKT3) IFN-γ Alexa 700 (1/250, B27), TNF BV786 (1/100, Mab11) and IL-2 APC (1/100, MQ1-17H12) from Biolegend. Finally, cells were washed and fixed in CellFIX (BD Biosciences). Samples were acquired on a BD Fortessa flow cytometer and analyzed using FlowJo (v10.8, FlowJo) and Pestle and Spice v6.1 (https://niaid.github.io/spice). A gating strategy is provided in Extended Data Fig. 1. Results are expressed as the frequency of CD4⁺ or CD8⁺ T cells expressing IFN-γ, TNF or IL-2. Due to high TNF backgrounds, cells producing TNF alone were excluded from the analysis. All data are presented after background subtraction.

### Live virus neutralization assay

A live neutralization assay was performed on plasma obtained from 10 out of the 15 participants vaccinated with BNT162b2 included in this study. H1299-E3 cells were plated in a 96-well plate (Corning) at 30,000 cells per well 1 day pre-infection. Plasma was separated from EDTA-anticoagulated blood by centrifugation at 500*g* for 10 min and stored at −80 °C. Aliquots of plasma samples were heat-inactivated at 56 °C for 30 min and clarified by centrifugation at 10,000*g* for 5 min.

Virus stocks were used at approximately 50–100 focus-forming units per microwell and added to diluted plasma. Antibody-virus mixtures were incubated for 1 h at 37 °C, 5% CO2. Cells were infected with 100 µl of the virus–antibody mixtures for 1 h, then 100 µl of 1× RPMI 1640 (Sigma-Aldrich, R6504), 1.5% carboxymethylcellulose (Sigma-Aldrich, C4888) overlay was added without removing the inoculum. Cells were fixed 18 h post-infection using 4% PFA (Sigma-Aldrich) for 20 min. Foci were stained with a rabbit anti-spike monoclonal antibody (BS-R2B12, GenScript A02058) at 0.5 µg ml⁻¹ in a permeabilization buffer containing 0.1% saponin (Sigma-Aldrich), 0.1% BSA (Sigma-Aldrich) and 0.05% Tween-20 (Sigma-Aldrich) in PBS. Plates were incubated with primary antibody overnight at 4 °C, then washed with wash buffer containing 0.05% Tween-20 in PBS. Secondary goat anti-rabbit horseradish peroxidase (Abcam ab205718) antibody was added at 1 µg ml⁻¹ and incubated for 2 h at room temperature with shaking. TrueBlue peroxidase substrate (SeraCare 5510–0030) was then added at 50 µl per well and incubated for 20 min at room temperature. Plates were imaged in an ELISPOT instrument with built-in image analysis (C.T.L).

### SARS-CoV-2 pseudovirus-based neutralization assay

A pseudovirus-based neutralization assay was performed on plasma obtained from all participants vaccinated with two doses of Ad26.COV2.S (*n* = 20). SARS-CoV-2 pseudotyped lentiviruses were prepared by co-transfecting the HEK 293T cell line with the SARS-CoV-2 614G spike (D614G) or SARS-CoV-2 Beta spike (L18F, D80A, D215G, K417N, E484K, N501Y, A701V and 242–244 del) plasmids with a firefly luciferase encoding lentivirus backbone plasmid. The parental plasmids were provided by E. Landais and D. Sok. For the neutralization assays, heat-inactivated plasma samples were incubated with SARS-CoV-2 pseudotyped virus for 1 h at 37 °C, 5% CO₂. Subsequently, 1 × 10⁴ HEK 293T cells engineered to overexpress ACE-2, provided by M. Farzan, were added and the incubated at 37 °C, 5% CO₂ for 72 h, upon which the luminescence of the luciferase gene was measured. CB6 and CA1 monoclonal antibodies were used as controls.

### Statistical analysis

Statistical analyses were performed in Prism (v9; GraphPad Software). Non-parametric tests were used for all comparisons. The Kruskal–Wallis and Mann–Whitney tests were used for unmatched samples, and the Friedman and Wilcoxon tests for paired samples. *P* values less than 0.05 were considered statistically significant. Details of analysis performed for each experiment are described in the figure legends.

### Reporting summary

Further information on research design is available in the Nature Research Reporting Summary linked to this paper.

### Data availability

Complete genome sequences for the viral isolates were deposited in GISAID. Source data are provided with this paper.

**Acknowledgements** We thank the participants who volunteered for this study. We thank W. van Hougenhouck-Tulleken for assistance with database management; R. Ramlall and J. Semugga for assistance with participant recruitment at Tshwane District Hospital; C. Scheepers and J. Everatt for generous assistance with plotting lineage prevalence; L. Singh and the NHLS Greenpoint Laboratory for PCR testing; and Biocair for assistance with shipping delays of critical reagents. Research reported in this publication was supported by the South African Medical Research Council (SA-MRC) with funds received from the South African Department of Science and Innovation (DSI), including grants 96825, SHIPNCD 76756 and DST/CON 0250/2012. This work was also supported by the Poliomyelitis Research Foundation (21/65) and Wellcome CIDRI-Africa, which is supported by core funding from Wellcome Trust (203135/Z/16/Z and 222574). This project has been funded in part with Federal funds from the National Institute of Allergy and Infectious Diseases, National Institutes of Health, Department of Health and Human Services, under contract no. 75N93021C00016 to A. Sette and contract no. 75N9301900065 to A. Sette and D.W. P.L.M. is supported by the SA Research Chairs Initiative of the DSI and the National Research Foundation (NRF; 9834). W.A.B. and C.R. are supported by the EDCTP2 programme of the European Union's Horizon 2020 programme

(TMA2017SF-1951-TB-SPEC to C.R. and TMA2016SF-1535-CaTCH-22 to W.A.B.). N.A.B.N. acknowledges funding from the SA-MRC, MRC UK and NRF. A. Sette acknowledges funding from the Bill and Melinda Gates award INV-018944, the NIH (AI138546) and the SA-MRC. R.J.W. acknowledges funding from the Francis Crick Institute, which receives funding from Wellcome FC0010218, UKRI FC0010218, CRUK FC0010218 and Rosetrees Trust (M926).

**Author contributions** W.A.B, C.R. and R.K. designed the study. C.R. and W.A.B analysed the data and wrote the manuscript. K.K., S.C., M.B., F.K. and S.V.M. performed neutralization assays and analysed the data. R.K., M.B.T., A.N., R.B., N.B. and A. Suzuki performed flow cytometry assays. M.M., S.S., M.A., D.M., O.A., C.S., E.d.B., M.A.V.D.M., Z.d.B., T.R.d.V. A.B., G.v.d.B, A.M., A. Strydom and M.V. recruited patients and managed cohorts. A.G., D.W. and A. Sette designed and provided variant peptide pools. T.d.O., J.G., S.P., Y.N. and H.T. generated and analysed sequence data. T.M.-G., J.N.B., P.L.M. and A. Sigal supervised the neutralization assay. N.A.B.N., R.J.W., V.U., M.T.B and T.R. led the clinical cohorts. L.-G.B. and G.G. led the Sisonke Ad26.COV2.S vaccination trial. All authors reviewed and edited the manuscript.

**Competing interests** A. Sette is a consultant for Gritstone Bio, Flow Pharma, Arcturus Therapeutics, ImmunoScape, CellCarta, Avalia, Moderna, Fortress and Repertoire. All of the other authors declare no competing interests. La Jolla Institute for Immunology has filed a patent for protection for various aspects of vaccine design and identification of specific epitopes.

**Additional information**
**Correspondence and requests for materials** should be addressed to Wendy A. Burgers or Catherine Riou.

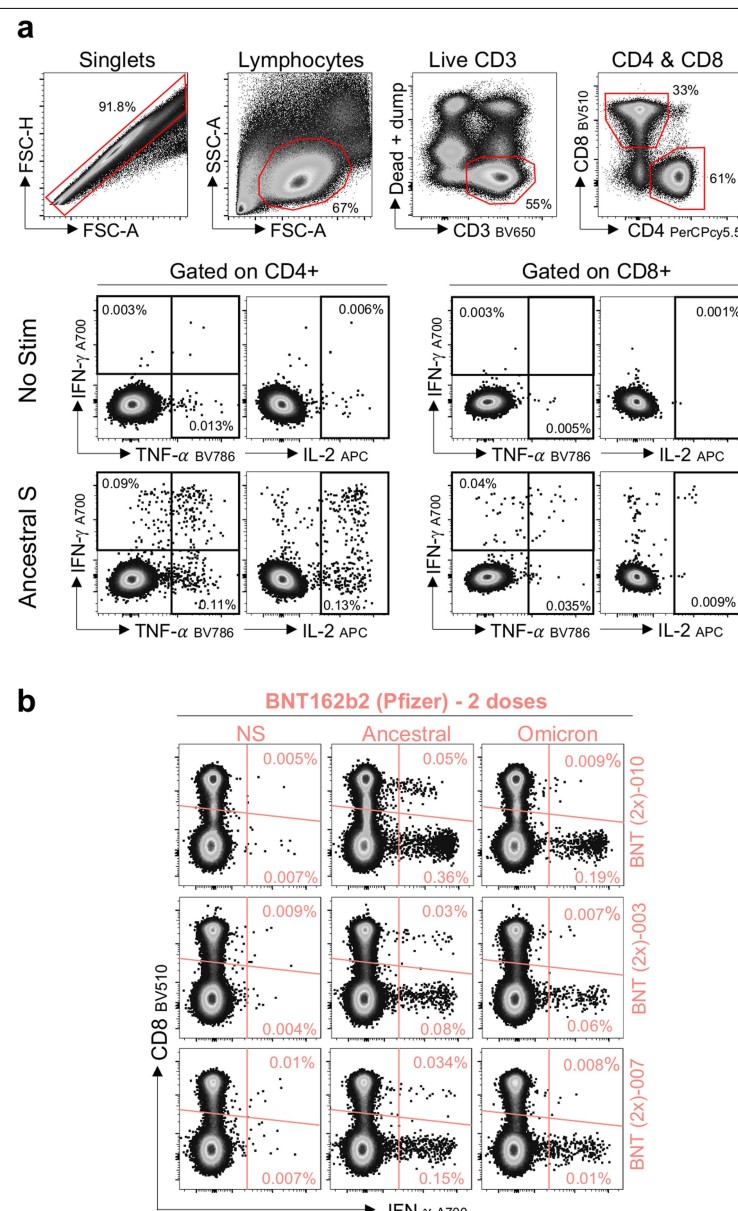

**Extended Data Fig. 1 | Gating strategy and examples of flow cytometry plots. a**, Gating strategy and representative examples of SARS-CoV-2 spike-specific IFN-γ, IL-2 and TNF-α production. **b**, Spike-specific expression of IFN-γ in the T cell compartment of the three BNT162b2-vaccinated participants where Omicron-specific CD8+ T cells were undetectable.

**a**

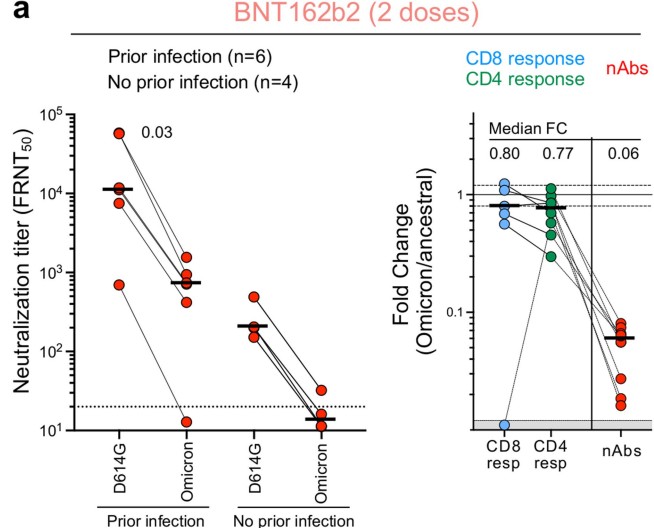

**b**

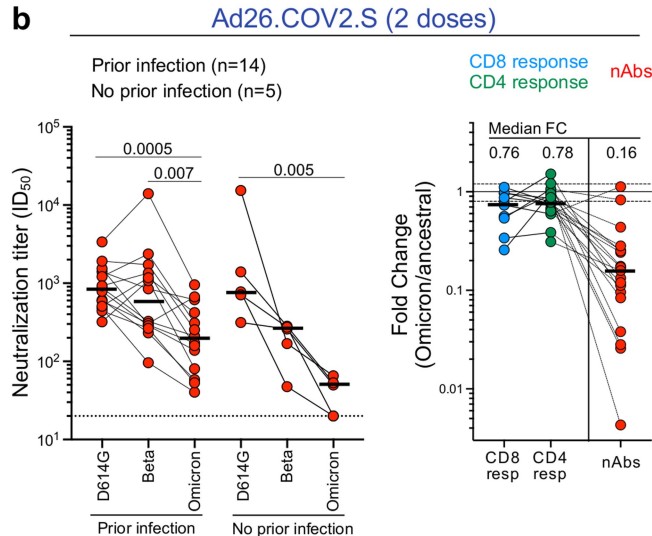

**Extended Data Fig. 2 | Neutralization of Omicron compared to the ancestral SARS-CoV-2 (D614G) by plasma from participants vaccinated with two doses of BNT162b2 or Ad26.COV2.S. a**, Neutralization by BNT162b2 plasma (n = 10), 6 with prior COVID-19 infection and 4 without) was performed using a live virus neutralization assay. The reciprocal plasma dilution ($FRNT_{50}$) resulting in 50% reduction in the number of infection foci is reported. The threshold of detection was set at a $FRNT_{50}$ of 20. A two-tailed paired Wilcoxon test was used to compare ancestral and Omicron titers. Comparison of the fold change in SARS-CoV-2-specific CD8+ and CD4+ T cell responses and neutralization titers (Omicron/ancestral) is depicted in the right panel. Bars represent medians. **b**, Neutralization against ancestral, Beta and Omicron variants by plasma from Ad26.COV2.S vaccinees (two doses; n = 19), including 14 with prior COVID-19 infection and 5 without, was performed using a SARS-CoV-2 pseudovirus-based neutralization assay. The threshold of detection was a 50% inhibitory dilution ($ID_{50}$) of 20. A Friedman test with Dunn´s multiple comparisons post-test was used to compare the titers of the three variants tested. Comparison of the fold change in SARS-CoV-2-specific CD8+ and CD4+ T cell response and neutralization titers (Omicron/ancestral) is depicted in the right panel. Bars represent medians.

## CD4 response

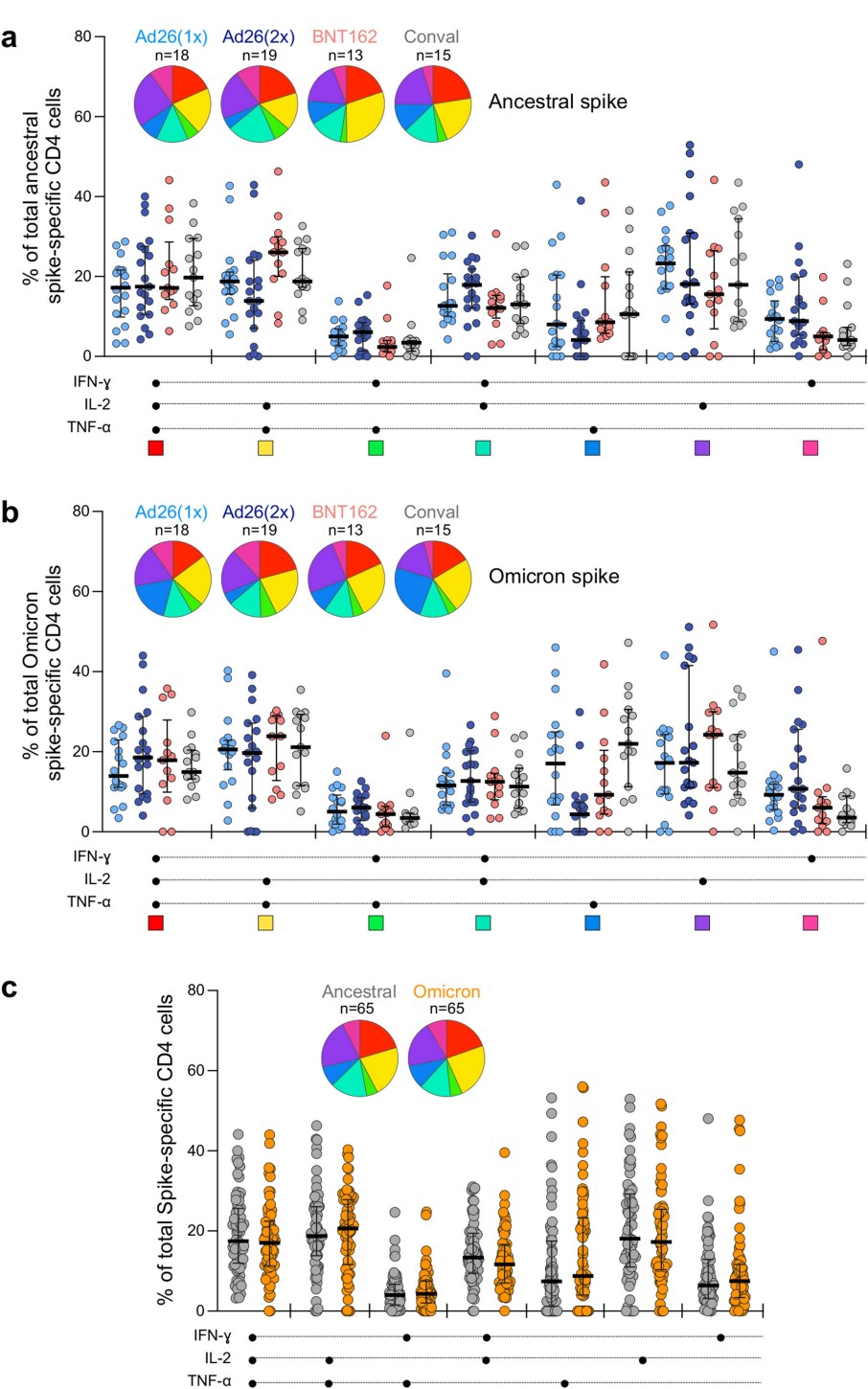

**Extended Data Fig. 3 | Polyfunctional profiles of SARS-CoV-2-specific CD4+ T cells after vaccination and in unvaccinated convalescent volunteers. a, b**, Comparison of the polyfunctional profile of ancestral (**a**) and Omicron (**b**) spike-specific CD4+ T cells between the four groups (Ad26.COV2.S-one dose, Ad26.COV.S-two doses, BNT162b2-two doses and unvaccinated convalescent volunteers). **c**, Comparison of the polyfunctional profile between ancestral and Omicron spike-specific CD4+ T cells including all CD4+ T cell responding participants, irrespective of their clinical grouping. The medians and IQR are shown. Each response pattern (i.e., any possible combination of IFN-γ, IL-2 or TNF-α expression) is color-coded, and data are summarized in the pie charts. No significant differences were observed between pies using a permutation test. The number of participants included in each analysis is indicated on the graphs.

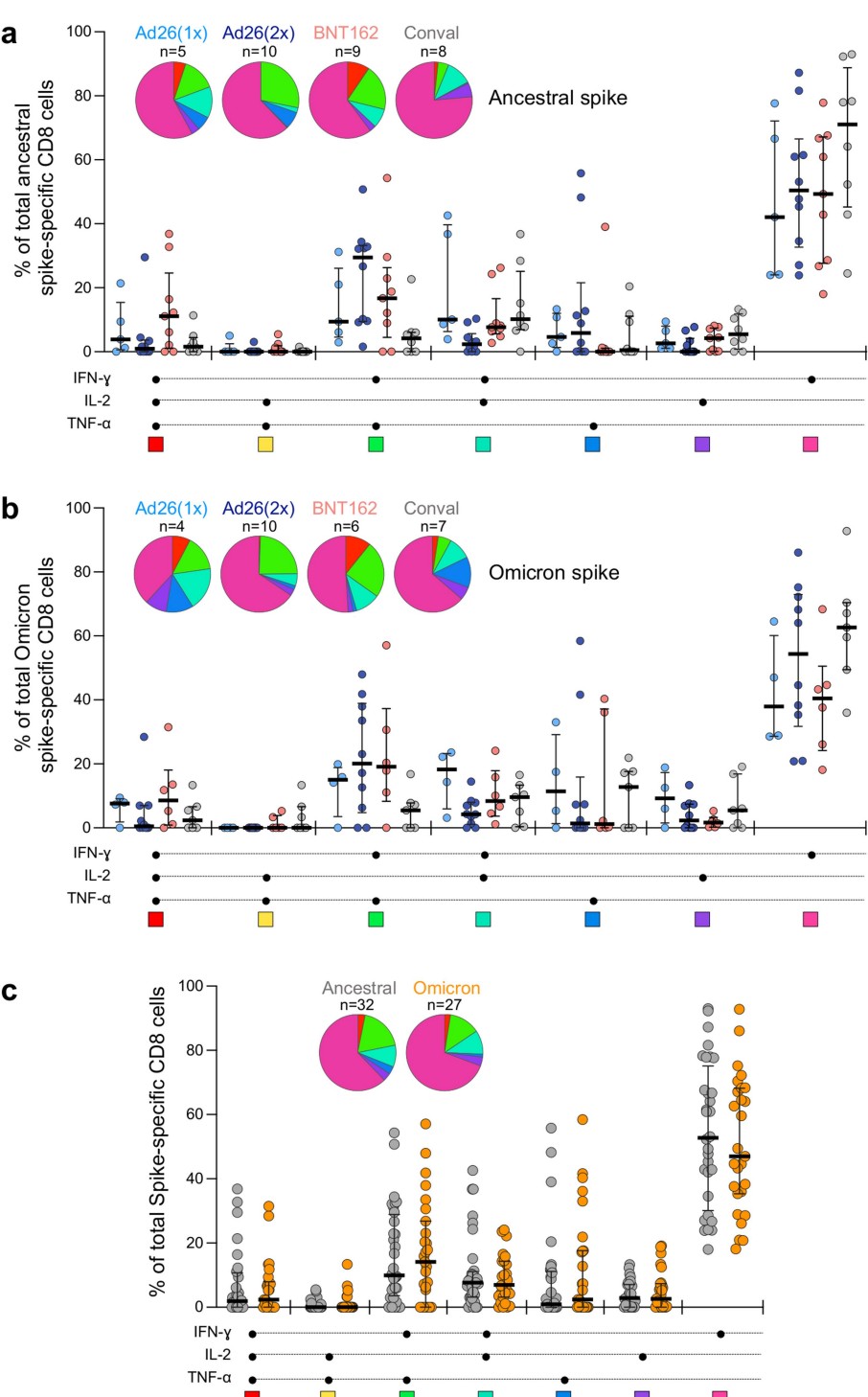

**Extended Data Fig. 4 | Polyfunctional profiles of SARS-CoV-2-specific CD8+ T cells after vaccination and in unvaccinated convalescent volunteers. a, b**, Comparison of the polyfunctional profile of ancestral (**a**) and Omicron (**b**) spike-specific CD8+ T cells between the four groups (Ad26.COV2.S-one dose, Ad26.COV2.S-two doses, BNT162b2-two doses and unvaccinated convalescent COVID-19 volunteers). **c**, Comparison of the polyfunctional profile between ancestral spike and Omicron spike-specific CD8+ T cells including all CD8+ T cell responding participants, irrespective of their clinical grouping. The medians and IQR are shown. Each response pattern (i.e., any possible combination of IFN-γ, IL-2 or TNF-α expression) is color-coded, and data are summarized in the pie charts. No significant differences were observed between pies using a permutation test. The number of participants included in each analysis is indicated on the graphs.

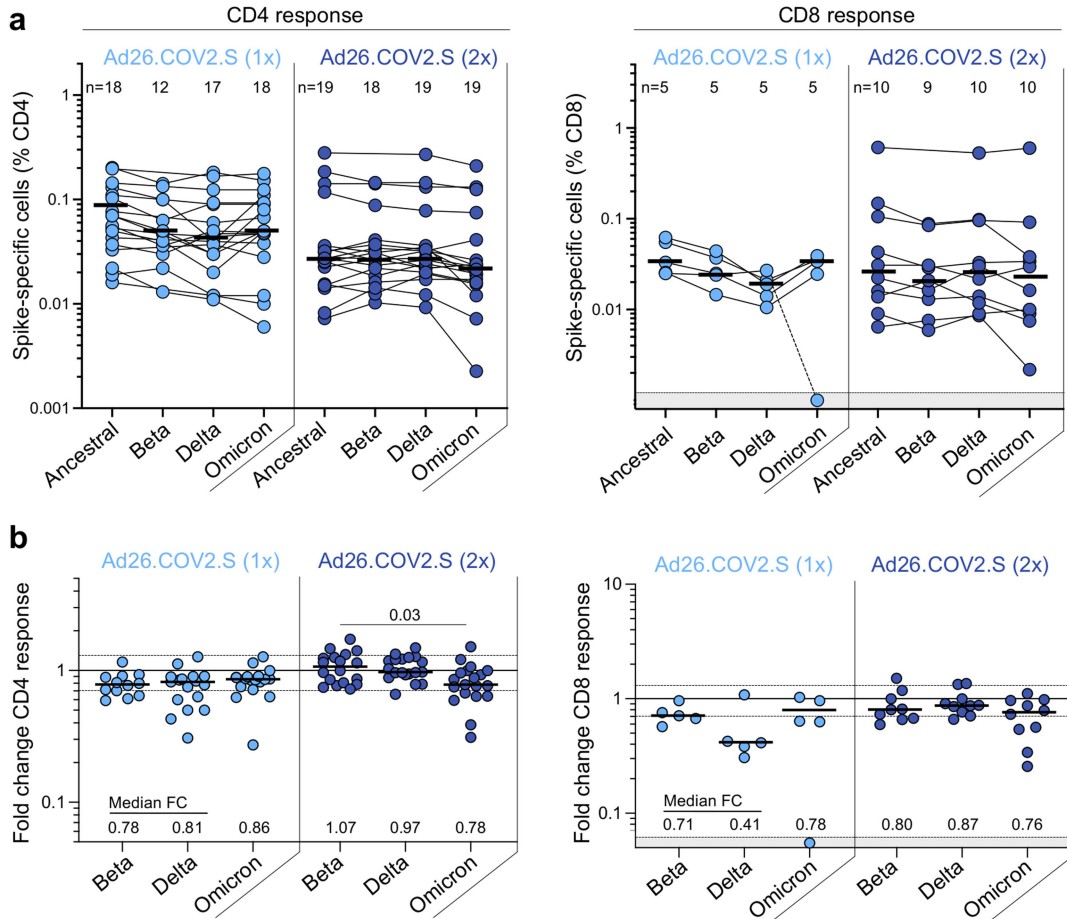

**Extended Data Fig. 5 | T cell responses to the ancestral, Beta, Delta and Omicron SARS-CoV-2 spike in participants who received Ad26.COV2.S (one or two doses). a**, Frequency of spike-specific CD4+ (left panel) and CD8+ T cells (right panel) producing any of the measured cytokines (IFN-γ, IL-2 or TNF-α) in response to ancestral, Beta, Delta and Omicron spike peptide pools. Bars represent median of responders. No significant differences were observed between variants using a Kruskal-Wallis test with Dunn's multiple comparisons post-test. **b**, Fold change in the frequency of spike-specific CD4+ (left panel) and CD8+ T cells (right panel) between ancestral and Omicron spike responses. Bars represent medians. Differences between SARS-CoV-2 variants were calculated using a Kruskal-Wallis test with Dunn's multiple comparisons post-test. Median fold changes are indicated at the bottom of each graph. The number of participants included in each analysis is indicated on the graphs.

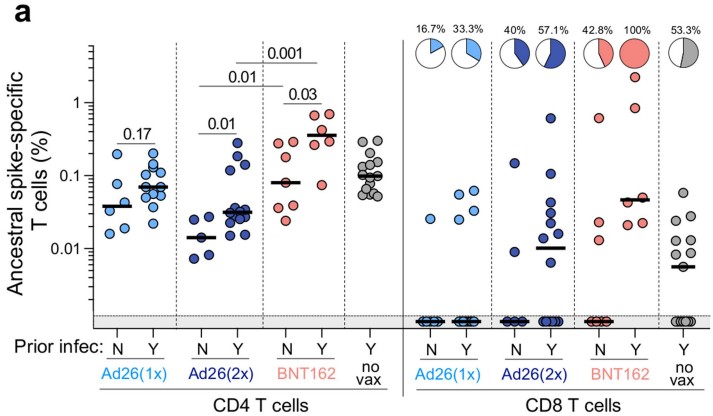

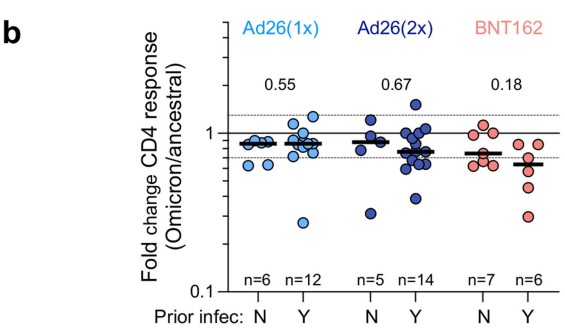

**Extended Data Fig. 6 | Impact of prior COVID-19 infection on T cell responses to the ancestral and Omicron SARS-CoV-2 spike in vaccinated participants. a**, Comparison of the frequency of ancestral spike-specific T cell responses in vaccinated participants who had (Y) or did not have (N) prior SARS-CoV-2 infection. Pies depict the proportion of participants exhibiting a detectable CD8+ T cell response. Bars represent medians. Statistical differences were calculated using a two-tailed Mann-Whitney test. **b**, Fold change in the frequency of spike-specific CD4+ T cells between ancestral and Omicron spike responses in the three vaccine groups. Bars represent medians. Statistical differences were calculated using a two-tailed Mann-Whitney test. The number of participants included in each analysis is indicated on the graphs.

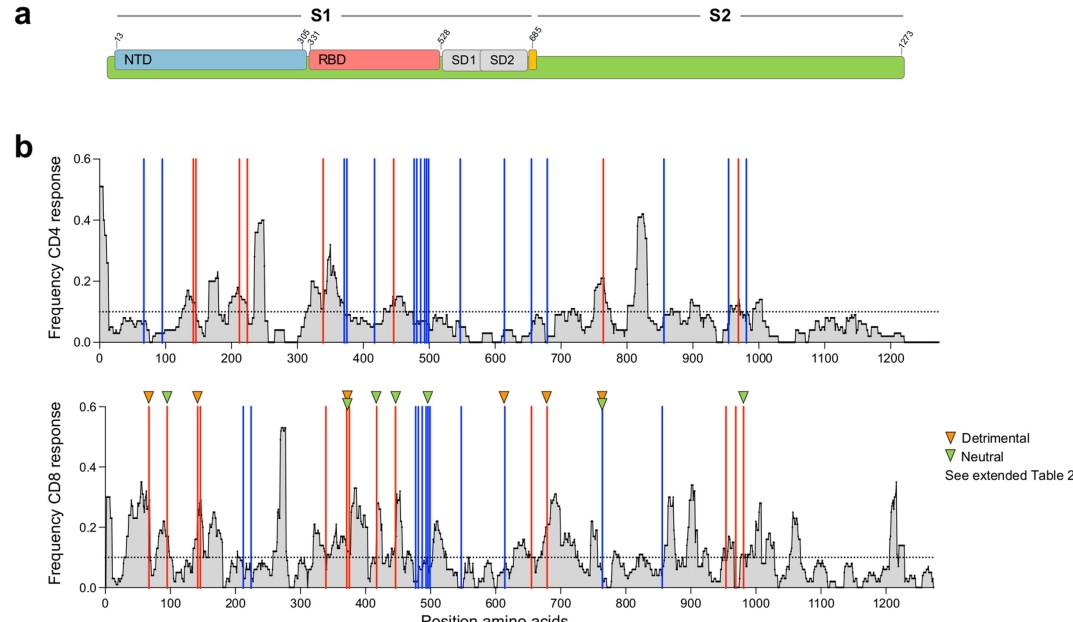

**Extended Data Fig. 7 | Distribution of spike SARS-CoV-2 epitopes targeted by CD4+ and CD8+ T cells. a**, Schematic of SARS-CoV-2 spike protein primary structure colored by domain. NTD: N-terminal domain, RBD: receptor binding domain, SD1: Sub-domain 1, SD2: Sub-domain 2. **b**, Distribution and frequency of recognition of confirmed CD4+ (top) and CD8+ T cell epitopes (bottom) across the entire spike protein. Data represent experimentally confirmed epitopes from the Immune Epitope Database and Analysis Resource (www.iedb.org). Red lines depict the position of Omicron mutations that recorded a frequency of recognition > 10% and blue lines < 10%. The position of variable epitopes associated with specific HLA-class I (see Extended Data Table 2) is indicated by a triangle. Mutations with a detrimental or neutral impact for HLA binding are depicted in orange and green, respectively.

**Extended Data Table 1 | Clinical characteristics of vaccinated, convalescent, and hospitalized COVID-19 participants**

**Table 1a:**

| Vaccinee cohorts | Vaccinees (all) | Without prior infection | With prior infection |
|---|---|---|---|
| **Ad26.COV2.S (Janssen) - 1 dose** | | | |
| Number | 20 | 7 | 13 |
| Age (y) (median, IQR) | 41 [32-52] | 44 [32-54] | 41 [32-48] |
| Gender (% female) | 75% | 71.4% | 76.9% |
| With co-morbidities (n, %) | 6/20 (30%) | 2/7 (28.6%) | 4/13 (30.8%) |
| Days since last vaccine dose (median, IQR) | 28 [28-32] | 31 [28-34] | 28 [27-29] |
| Months since COVID episode (median, IQR) | - | na | 3 [2-6]* |
| **Ad26.COV2.S (Janssen) - 2 doses** | | | |
| Number | 20 | 6 | 14 |
| Age (y) (median, IQR) | 38 [33-56] | 58 [36-62] | 36 [32-44] |
| Gender (% female) | 75% | 83.3% | 71.4% |
| With co-morbidities (n, %) | 9/20 (45%) | 4/6 (66.7%) | 5/14 (35.7%) |
| Days since last vaccine dose (median, IQR) | 22 [21-24] | 23 [21-25] | 22 [21-23] |
| Months since COVID episode (median, IQR) | - | na | 15 [11-18] |
| **BNT162b2 (Pfizer) - 2 doses** | | | |
| Number | 15 | 9 | 6 |
| Age (y) (median, IQR) | 53 [45-71] | 53 [41-74] | 57 [44-66] |
| Gender (% female) | 53.3% | 44.4% | 66.7% |
| With co-morbidities (n, %) | 9/15 (60%) | 4/9 (44.4%) | 5/6 (83.3%) |
| Days since last vaccine dose (median, IQR) | 33 [22-48] | 46 [23-65] | 27 [20-38] |
| Months since COVID episode (median, IQR) | - | na | 14 [13-15] |

**Table 1b:**

| Convalescent cohort (unvaccinated) | |
|---|---|
| Number | 15 |
| Age (y) (median, IQR) | 39 [35-49] |
| Gender (% female) | 12% |
| With co-morbidities (n, %) | 6/15 (40%) |
| Months since COVID episode (median, IQR) | 1.4 [1.3-6] |

**Table 1c:**

| Hospitalized COVID-19 cohort | Total | Wave 1 (Ancestral) | Wave 2 (Beta) | Wave 3 (Delta) | Wave 4 (Omicron) |
|---|---|---|---|---|---|
| Number | 68 | 17 | 16 | 16 | 19 |
| Age (y) (median, range) | 53 [39-64] | 55 [41-62] | 64 [53-68] | 49 [37-63] | 47 [35-60] |
| Gender (% female) | 47% | 29.4% | 56.2% | 56.2% | 47.4% |
| With co-morbidities (n, %) | 53/68 (77.9%) | 11/17 (64.7%) | 15/16 (93.7%) | 13/16 (81.2%) | 14/19 (73.7%) |
| Day since PCR+ (median, range) | 4 [2-8] | 2 [1-3.5] | 5 [2-10] | 6 [2-10] | 4 [3-6] |
| Severe COVID-19 (WHO ≥5) | 29/68 (42.6%) | 7/17 (41.2%) | 1/16 (6.25%) | 12/16 (75%) | 9/19 (47.4%) |

**Table 1a**: Clinical characteristics of vaccinee cohorts. Co-morbidities include: asthma, hypertension, obesity or diabetes mellitus.*: data regarding time post Covid-19 infection were available for only 6 out of the 13 participants who received 1 dose of Ad26-COV.S. **Table 1b:** Clinical characteristics of convalescent COVID-19 patients. Co-morbidities include: asthma, hypertension, obesity or diabetes mellitus. **Table 1c**: Clinical characteristics of hospitalized COVID-19 patient cohort. Co- morbidities include: asthma, hypertension, obesity or diabetes mellitus. Severe COVID-19 was defined based on oxygen therapy requirement according to the WHO ordinal scale scoring system ($O_2$ via high flow to extracorporeal membrane oxygenation).

**Extended Data Table 2 | In silico analysis of the impact of Omicron mutations on epitope recognition by MHC Class I**

| | peptide (WT) | aa start | aa stop | Mutation | Peptide (Omicron) | MHC class-I allele | P rank (WT) | P rank (Omi) | IC$_{50}$ (WT) | IC$_{50}$ (Omi) | Previously described |
|---|---|---|---|---|---|---|---|---|---|---|---|
| Detrimental | VTWFHAIHV | 62 | 70 | A67V, del 69-70 | VTWFHVISG | HLA-A*02:11 | 0.64 | 18 | 18.91 | 8421.29 | Deng, 2021 |
| | | | | | | HLA-A*68:02 | 0.26 | 9.3 | 45.32 | 8237.36 | |
| | GVYYHKNNK | 142 | 150 | G142D, 143-145 del | DHKNNKSWM | HLA-A*03:01 | 0.07 | 89 | 27.86 | 42735.87 | Tarke, 2021 |
| | ASFSTFKCY | 372 | 380 | S373P, S375F | APFFTFKCY | HLA-A*30:02 | 0.06 | 2.8 | 43.85 | 2759.86 | |
| | | | | | | HLA-B*15:03 | 0.26 | 2.5 | 25.16 | 495.99 | |
| | YQDVNCTEV | 612 | 620 | D614G | YQGVNCTEV | HLA-A*02:131 | 0.53 | 1.3 | 49.38 | 149.98 | Tarke, 2021 |
| | | | | | | HLA-A*02:11 | 0.95 | 2 | 31.58 | 93.34 | |
| | SPRRARSVA | 680 | 688 | P681H | SHRRARSVA | HLA-B*07:02 | 0.02 | 2.9 | 4.65 | 4721.01 | Tarke, 2021 |
| | | | | | | HLA-B*07:05 | 0.03 | 3 | 4.51 | 2591.37 | |
| | | | | | | HLA-B*07:06 | 0.03 | 3 | 4.51 | 2591.37 | |
| | | | | | | HLA-B*42:02 | 0.03 | 3.6 | 38.64 | 8401.27 | |
| | QLNRALTGI | 762 | 770 | N764K | QLKRALTGI | HLA-A*02:11 | 0.84 | 4 | 27.13 | 440.69 | |
| Neutral | GVYFASTEK | 89 | 97 | T95I | GVYFASIEK | HLA-A*11:01 | 0.03 | 0.02 | 9.66 | 6.98 | Tarke, 2021 |
| | | | | | | HLA-A*03:01 | 0.05 | 0.04 | 19.15 | 17.74 | |
| | NSASFSTFK | 370 | 379 | S371L, S373P, S375F | NLAPFFTFK | HLA-A*68:01 | 0.02 | 0.06 | 4.4 | 8.51 | Tarke, 2021 |
| | | | | | | HLA-A*34:02 | 0.02 | 0.01 | 16.17 | 11.26 | |
| | | | | | | HLA-A*11:01 | 0.03 | 0.03 | 9.13 | 10.02 | |
| | YNSASFSTF | 369 | 378 | S371L, S373P, S375F | YNLAPFFTF | HLA-B*15:03 | 0.05 | 0.18 | 6.63 | 18.49 | |
| | KIADYNYKL | 417 | 425 | K417N | NIADYNYKL | HLA-A*02:02 | 0.07 | 0.17 | 6.58 | 11.64 | Shomuradova, 2021 |
| | | | | | | HLA-A*02:05 | 0.09 | 0.16 | 13.49 | 27.89 | |
| | VGGNYNYLY | 445 | 453 | G446S | VSGNYNYLY | HLA-A*30:02 | 0.06 | 0.02 | 45.32 | 20.21 | Tarke, 2021 |
| | SKVGGNYNY | 443 | 451 | G446S | SKVSGNYNY | HLA-B*15:03 | 0.26 | 0.17 | 25.38 | 17.47 | Tarke, 2021 |
| | YFPLQSYGF | 489 | 498 | Q493R | YFPLRSYSF | HLA-A*23:01 | 0.12 | 0.06 | 49.01 | 24.54 | Zhang, 2021 |
| | GSFCTQLNR | 757 | 765 | N764K | GSFCTQLKR | HLA-A*11:01 | 0.18 | 0.26 | 37.18 | 53.31 | |
| | QLNRALTGI | 762 | 770 | N764K | QLKRALTGI | HLA-A*02:03 | 0.25 | 0.47 | 15.69 | 27.68 | |
| | VLNDILSRL | 976 | 985 | L981F | VLNDIFSRL | HLA-A*02:02 | 0.04 | 0.05 | 4.16 | 4.36 | Tarke, 2021 |
| | | | | | | HLA-A*02:03 | 0.06 | 0.08 | 5.19 | 6.38 | |
| | | | | | | HLA-A*02:11 | 0.09 | 0.09 | 3.4 | 3.5 | |
| | | | | | | HLA-A*02:05 | 0.09 | 0.09 | 14.04 | 14.07 | |

Putative HLA class I restrictions were inferred using the Immune Epitope Database (IEDB) analysis resource (http://tools.iedb.org/tepitool/, NetMHCpan prediction method). Selected ancestral peptides with predicted a percentile rank (P rank) ≤1 and a IC50 <50 nM are shown, and the binding predictions for the corresponding Omicron mutated epitope. References of previously described immunoreactive peptides (WT) are provided in the last column. Deng et al. (https://doi.org/10.1002/JLB.4MA0621-020R); Tarke et al. (https://doi.org/10.1016/j.xcrm.2021.100204); Shomuradova et al. (https://doi.org/10.1016/j.immuni.2020.11.004), Zhang et al. (https://doi.org/10.1016/j.celrep.2021.109708).

# Extended Data Table 3 | Clinical characteristics of each hospitalized and convalescent COVID-19 participant

| Participant ID | Sex | Age | Date of blood sampling | Time since COVID-19 diagnosis (days) | Vaccination status | Evidence of prior symptomatic COVID-19 | WHO score at sampling | Hyper-tension | Diabetes | Obesity | Asthma | Outcome |
|---|---|---|---|---|---|---|---|---|---|---|---|---|
| Hosp-Wave 4-001 | Male | 36 | 2021/12/07 | 9 | unvaccinated | No | 5 | Yes | Yes | Yes | No | Discharged |
| Hosp-Wave 4-002 | Female | 26 | 2021/12/01 | 5 | unvaccinated | No | 4 | No | Yes | No | No | Still admitted |
| Hosp-Wave 4-003 | Female | 17 | 2021/12/06 | 3 | unvaccinated | No | 4 | No | Yes | No | No | Discharged |
| Hosp-Wave 4-004 | Female | 24 | 2021/12/07 | 4 | unvaccinated | No | 4 | No | Yes | No | No | Discharged |
| Hosp-Wave 4-005 | Female | 63 | 2021/12/08 | 3 | unvaccinated | No | 4 | No | No | No | No | Discharged |
| Hosp-Wave 4-006 | Male | 59 | 2021/12/14 | 4 | unvaccinated | No | 6 | Yes | Yes | No | No | Still admitted |
| Hosp-Wave 4-007 | Female | 62 | 2021/12/07 | 8 | unvaccinated | No | 5 | Yes | No | Yes | No | Discharged |
| Hosp-Wave 4-008 | Male | 58 | 2021/12/13 | 10 | unvaccinated | No | 5 | Yes | Yes | No | No | Discharged |
| Hosp-Wave 4-009 | Male | 49 | 2021/12/13 | 8 | unvaccinated | No | 4 | No | No | No | No | Discharged |
| Hosp-Wave 4-010 | Female | 71 | 2021/12/07 | 4 | unvaccinated | No | 5 | Yes | Yes | No | No | Discharged |
| Hosp-Wave 4-011 | Male | 60 | 2021/12/07 | 1 | unvaccinated | No | 4 | Yes | Yes | No | No | Discharged |
| Hosp-Wave 4-012 | Male | 62 | 2021/12/13 | 3 | unvaccinated | No | 5 | No | No | No | No | Discharged |
| Hosp-Wave 4-013 | Female | 38 | 2021/12/13 | 5 | unvaccinated | No | 5 | No | Yes | No | No | Still admitted |
| Hosp-Wave 4-014 | Male | 49 | 2021/12/13 | 1 | unvaccinated | No | 4 | Yes | No | No | No | Still admitted |
| Hosp-Wave 4-015 | Male | 47 | 2021/12/13 | 4 | unvaccinated | No | 4 | No | No | No | No | Discharged |
| Hosp-Wave 4-016 | Female | 35 | 2021/12/14 | 5 | unvaccinated | No | 5 | No | No | No | No | Discharged |
| Hosp-Wave 4-017 | Female | 21 | 2021/12/14 | 4 | unvaccinated | No | 5 | No | No | No | No | Discharged |
| Hosp-Wave 4-018 | Male | 38 | 2021/12/15 | 6 | unvaccinated | Unk | 3 | Yes | No | No | No | Discharged |
| Hosp-Wave 4-019 | Male | 37 | 2021/12/15 | 6 | unvaccinated | No | 3 | No | No | No | No | Discharged |
| Hosp-Wave 3-001 | Female | 33 | 2021/07/14 | 9 | unvaccinated | No | 5 | Yes | Yes | No | No | Discharged |
| Hosp-Wave 3-002 | Female | 76 | 2021/07/14 | 6 | Unk | No | 5 | Yes | No | Yes | No | Discharged |
| Hosp-Wave 3-003 | Male | 36 | 2021/07/15 | 6 | unvaccinated | No | 3 | No | Yes | No | No | Discharged |
| Hosp-Wave 3-004 | Female | 43 | 2021/07/15 | 6 | unvaccinated | Unk | 5 | Yes | No | No | No | Discharged |
| Hosp-Wave 3-005 | Male | 59 | 2021/07/15 | 2 | Unk | Unk | 5 | Yes | Yes | No | No | Discharged |
| Hosp-Wave 3-006 | Female | 64 | 2021/07/15 | 4 | Unk | Unk | 6 | No | No | Yes | No | Demised |
| Hosp-Wave 3-007 | Male | 64 | 2021/07/16 | 2 | Unk | Unk | 3 | No | No | No | No | Discharged |
| Hosp-Wave 3-008 | Male | 47 | 2021/07/19 | 2 | unvaccinated | Unk | 3 | Yes | Yes | No | No | Discharged |
| Hosp-Wave 3-009 | Male | 29 | 2021/07/19 | 11 | unvaccinated | Unk | 6 | No | Yes | Yes | No | Discharged |
| Hosp-Wave 3-010 | Female | 33 | 2021/07/19 | 6 | unvaccinated | Unk | 5 | Yes | No | No | No | Discharged |
| Hosp-Wave 3-011 | Female | 42 | 2021/07/19 | 12 | unvaccinated | Unk | 5 | No | No | Yes | No | Discharged |
| Hosp-Wave 3-012 | Female | 52 | 2021/07/19 | 26 | Unk | Unk | 6 | Yes | Yes | Yes | No | Demised |
| Hosp-Wave 3-013 | Female | 44 | 2021/07/20 | 2 | unvaccinated | Unk | 5 | Yes | Yes | No | No | Discharged |
| Hosp-Wave 3-014 | Female | 72 | 2021/07/20 | 2 | Unk | Unk | 6 | Yes | Yes | Yes | No | Demised |
| Hosp-Wave 3-015 | Male | 53 | 2021/07/20 | 3 | Unk | Unk | 5 | No | No | No | No | Discharged |
| Hosp-Wave 3-016 | Male | 51 | 2021/07/21 | 12 | Unk | Unk | 4 | No | No | No | No | Discharged |
| Hosp-Wave 2-001 | Male | 64 | 2020/12/31 | 3 | unvaccinated | No | 3 | No | Yes | No | No | Discharged |
| Hosp-Wave 2-002 | Male | 68 | 2021/01/06 | 20 | unvaccinated | No | 4 | Yes | Yes | No | No | Discharged |
| Hosp-Wave 2-003 | Female | 61 | 2021/01/06 | 2 | unvaccinated | No | 4 | Yes | Yes | No | No | Discharged |
| Hosp-Wave 2-004 | Female | 65 | 2021/01/06 | 14 | unvaccinated | No | 4 | No | No | No | No | Discharged |
| Hosp-Wave 2-005 | Female | 53 | 2021/01/08 | 8 | unvaccinated | No | 4 | Yes | Yes | No | No | Discharged |
| Hosp-Wave 2-006 | Female | 68 | 2021/01/08 | 6 | unvaccinated | No | 4 | No | Yes | No | No | Discharged |
| Hosp-Wave 2-007 | Male | 64 | 2021/01/12 | 2 | unvaccinated | No | 4 | Yes | Yes | No | No | Discharged |
| Hosp-Wave 2-008 | Male | 45 | 2021/01/12 | 5 | unvaccinated | No | 4 | Yes | Yes | No | No | Discharged |
| Hosp-Wave 2-009 | Male | 83 | 2021/01/13 | 0 | unvaccinated | No | 4 | Yes | No | No | No | Discharged |
| Hosp-Wave 2-010 | Male | 33 | 2021/01/13 | 5 | unvaccinated | No | 3 | Yes | No | No | No | Discharged |
| Hosp-Wave 2-011 | Female | 73 | 2021/01/13 | 8 | unvaccinated | No | 3 | Yes | No | No | No | Discharged |
| Hosp-Wave 2-012 | Female | 53 | 2021/01/13 | 4 | unvaccinated | No | 4 | Yes | No | Yes | No | Discharged |
| Hosp-Wave 2-013 | Female | 64 | 2021/01/13 | 2 | unvaccinated | No | 4 | Yes | Yes | No | No | Discharged |
| Hosp-Wave 2-014 | Female | 67 | 2021/01/13 | 14 | unvaccinated | No | 4 | Yes | Yes | Yes | No | Discharged |
| Hosp-Wave 2-015 | Female | 68 | 2021/01/15 | 11 | unvaccinated | No | 5 | Yes | Yes | No | No | Discharged |
| Hosp-Wave 2-016 | Male | 49 | 2021/01/15 | 4 | unvaccinated | No | 3 | No | Yes | No | No | Discharged |
| Hosp-Wave 1-001 | Male | 68.1 | 2020/06/11 | 2 | unvaccinated | No | 3 | Yes | Yes | No | No | Discharged |
| Hosp-Wave 1-002 | Male | 36.1 | 2020/06/12 | 11 | unvaccinated | No | 4 | Yes | Yes | Yes | No | Discharged |
| Hosp-Wave 1-003 | Female | 21.2 | 2020/06/18 | 3 | unvaccinated | No | 3 | No | No | No | No | Discharged |
| Hosp-Wave 1-004 | Male | 56.4 | 2020/06/22 | 2 | unvaccinated | No | 4 | No | No | No | No | Discharged |
| Hosp-Wave 1-005 | Male | 49.3 | 2020/06/22 | 2 | unvaccinated | No | 3 | No | Yes | No | No | Discharged |
| Hosp-Wave 1-006 | Female | 55.5 | 2020/06/25 | 4 | unvaccinated | No | 5 | No | No | No | No | Discharged |
| Hosp-Wave 1-007 | Male | 41.2 | 2020/06/25 | 1 | unvaccinated | No | 4 | Yes | No | No | No | Discharged |
| Hosp-Wave 1-008 | Female | 63.5 | 2020/06/25 | 1 | unvaccinated | No | 4 | Yes | Yes | Yes | No | Discharged |
| Hosp-Wave 1-009 | Female | 37.9 | 2020/06/25 | 1 | unvaccinated | No | 4 | No | No | Yes | No | Discharged |
| Hosp-Wave 1-010 | Male | 58.0 | 2020/06/29 | 1 | unvaccinated | No | 4 | Yes | Yes | Yes | No | Discharged |
| Hosp-Wave 1-011 | Male | 49.6 | 2020/07/02 | 1 | unvaccinated | No | 5 | Yes | No | No | No | Discharged |
| Hosp-Wave 1-012 | Male | 63.7 | 2020/07/02 | 1 | unvaccinated | No | 5 | No | No | No | No | Discharged |
| Hosp-Wave 1-013 | Male | 55.0 | 2020/07/03 | 3 | unvaccinated | No | 5 | No | No | No | No | Discharged |
| Hosp-Wave 1-014 | Male | 41.6 | 2020/07/10 | 14 | unvaccinated | No | 5 | No | No | No | No | Discharged |
| Hosp-Wave 1-015 | Male | 60.4 | 2020/07/16 | 1 | unvaccinated | No | 4 | Yes | Yes | No | No | Discharged |
| Hosp-Wave 1-016 | Male | 67.7 | 2020/07/20 | 8 | unvaccinated | No | 5 | Yes | Yes | No | No | Discharged |
| Hosp-Wave 1-017 | Female | 56.4 | 2020/07/24 | 2 | unvaccinated | No | 5 | Yes | No | Yes | No | Discharged |
| Convalescent-001 | Female | 40 | 2021/01/19 | 213 | unvaccinated | No | na | Yes | No | No | No | na |
| Convalescent-002 | Female | 45 | 2021/02/12 | 178 | unvaccinated | No | na | Yes | No | No | No | na |
| Convalescent-003 | Female | 35 | 2021/01/19 | 183 | unvaccinated | No | na | No | No | No | No | na |
| Convalescent-004 | Female | 35 | 2021/01/19 | 148 | unvaccinated | No | na | No | No | No | No | na |
| Convalescent-005 | Male | 31 | 2021/01/19 | 213 | unvaccinated | No | na | No | No | No | No | na |
| Convalescent-006 | Female | 52 | 2021/02/15 | 243 | unvaccinated | No | na | Yes | No | Yes | No | na |
| Convalescent-007 | Female | 35 | 2021/01/26 | 38 | unvaccinated | No | na | Yes | No | No | No | na |
| Convalescent-008 | Female | 55 | 2021/01/19 | 179 | unvaccinated | No | na | No | No | No | No | na |
| Convalescent-009 | Female | 49 | 2021/01/21 | 30 | unvaccinated | No | na | Yes | No | No | No | na |
| Convalescent-010 | Male | 36 | 2021/01/20 | 35 | unvaccinated | No | na | No | No | No | No | na |
| Convalescent-011 | Female | 48 | 2021/01/21 | 40 | unvaccinated | No | na | No | No | No | No | na |
| Convalescent-012 | Male | 39 | 2021/02/03 | 42 | unvaccinated | No | na | No | No | No | No | na |
| Convalescent-013 | Female | 29 | 2021/02/01 | 38 | unvaccinated | No | na | No | No | No | No | na |
| Convalescent-014 | Female | 32 | 2021/02/03 | 38 | unvaccinated | No | na | No | No | No | No | na |
| Convalescent-015 | Female | 57 | 2021/02/03 | 38 | unvaccinated | No | na | Yes | No | No | No | na |

Unk: unknown. Na: not applicable.

## Extended Data Table 4 | Clinical characteristics of each vaccinated participant

| Participant ID | Sex | Age | Date of blood sampling | Vaccine regimen | Date of 1st vaccine dose | Date of 2nd vaccine dose | Time between vaccination and sampling (days) | COVID-19 history prior to sampling | Date of prior infection | Time between prior infection and sampling (months) | Hypertension | Diabetes | Obesity | Asthma |
|---|---|---|---|---|---|---|---|---|---|---|---|---|---|---|
| JJ (1x)-001 | F | 44 | 2021/03/18 | 1 dose Ad26.COV2.S | 2021/02/18 | na | 28 | No prior Infection | na | na | No | No | No | No |
| JJ (1x)-002 | M | 58 | 2021/03/23 | 1 dose Ad26.COV2.S | 2021/02/17 | na | 34 | No prior Infection | na | na | No | No | No | No |
| JJ (1x)-003 | F | 53 | 2021/04/08 | 1 dose Ad26.COV2.S | 2021/02/24 | na | 43 | No prior Infection | na | na | No | No | No | No |
| JJ (1x)-004 | F | 32 | 2021/03/23 | 1 dose Ad26.COV2.S | 2021/02/24 | na | 27 | No prior Infection | na | na | No | No | No | Yes |
| JJ (1x)-005 | F | 54 | 2021/03/18 | 1 dose Ad26.COV2.S | 2021/02/18 | na | 28 | No prior Infection | na | na | No | No | No | No |
| JJ (1x)-006 | M | 30 | 2021/03/24 | 1 dose Ad26.COV2.S | 2021/02/21 | na | 31 | No prior Infection | na | na | No | No | No | No |
| JJ (1x)-007 | F | 33 | 2021/03/23 | 1 dose Ad26.COV2.S | 2021/02/19 | na | 32 | No prior Infection | na | na | No | No | No | Yes |
| JJ (1x)-008 | F | 34 | 2021/03/18 | 1 dose Ad26.COV2.S | 2021/02/18 | na | 28 | Prior Infection | 2021/01/04 | 2.40 | No | No | No | No |
| JJ (1x)-009 | M | 30 | 2021/03/23 | 1 dose Ad26.COV2.S | 2021/02/22 | na | 29 | Prior Infection | 2020/12/18 | 3.13 | No | No | No | No |
| JJ (1x)-010 | M | 42 | 2021/03/18 | 1 dose Ad26.COV2.S | 2021/02/18 | na | 28 | Prior Infection | 2020/12/19 | 2.93 | No | No | No | Yes |
| JJ (1x)-011 | F | 38 | 2021/03/17 | 1 dose Ad26.COV2.S | 2021/02/18 | na | 27 | Prior Infection | 2020/10/07 | 5.30 | No | No | No | No |
| JJ (1x)-012 | M | 58 | 2021/03/24 | 1 dose Ad26.COV2.S | 2021/02/24 | na | 28 | Prior Infection | Unk | Unk | No | No | No | No |
| JJ (1x)-013 | F | 43 | 2021/03/23 | 1 dose Ad26.COV2.S | 2021/02/19 | na | 32 | Prior Infection | Unk | Unk | Yes | No | No | No |
| JJ (1x)-014 | F | 36 | 2021/03/25 | 1 dose Ad26.COV2.S | 2021/02/25 | na | 28 | Prior Infection | Unk | Unk | No | No | No | No |
| JJ (1x)-015 | F | 27 | 2021/04/22 | 1 dose Ad26.COV2.S | 2021/03/24 | na | 29 | Prior Infection | 2020/07/08 | 9.47 | No | No | No | No |
| JJ (1x)-016 | F | 28 | 2021/03/26 | 1 dose Ad26.COV2.S | 2021/02/26 | na | 28 | Prior Infection | 2021/02/16 | 1.25 | No | No | No | No |
| JJ (1x)-017 | F | 53 | 2021/03/18 | 1 dose Ad26.COV2.S | 2021/02/18 | na | 28 | Prior Infection | Unk | Unk | No | No | Yes | No |
| JJ (1x)-018 | F | 46 | 2021/03/31 | 1 dose Ad26.COV2.S | 2021/03/09 | na | 22 | Prior Infection | Unk | Unk | No | No | Yes | No |
| JJ (1x)-019 | F | 51 | 2021/04/01 | 1 dose Ad26.COV2.S | 2021/03/09 | na | 23 | Prior Infection | Unk | Unk | No | No | No | No |
| JJ (1x)-020 | F | 41 | 2021/04/07 | 1 dose Ad26.COV2.S | 2021/02/24 | na | 42 | Prior Infection | Unk | Unk | No | No | No | No |
| JJ (2x)-001 | F | 59 | 2021/12/06 | 2 doses Ad26.COV2.S | 2021/02/19 | 2021/11/12 | 24 | No prior Infection | na | na | Yes | No | No | No |
| JJ (2x)-002 | F | 64 | 2021/12/07 | 2 doses Ad26.COV2.S | 2021/02/27 | 2021/11/11 | 26 | No prior Infection | na | na | No | No | No | Yes |
| JJ (2x)-003 | F | 36 | 2021/12/13 | 2 doses Ad26.COV2.S | 2021/02/26 | 2021/11/19 | 24 | No prior Infection | na | na | No | No | Yes | No |
| JJ (2x)-004 | F | 58 | 2021/12/13 | 2 doses Ad26.COV2.S | 2021/02/22 | 2021/11/22 | 21 | No prior Infection | na | na | No | No | No | No |
| JJ (2x)-005 | M | 36 | 2021/12/06 | 2 doses Ad26.COV2.S | 2021/02/19 | 2021/11/15 | 21 | No prior Infection | na | na | No | No | No | Yes |
| JJ (2x)-006 | F | 62 | 2021/12/07 | 2 doses Ad26.COV2.S | 2021/02/18 | 2021/11/15 | 22 | No prior Infection | na | na | No | No | No | No |
| JJ (2x)-007 | M | 33 | 2021/12/06 | 2 doses Ad26.COV2.S | 2021/02/18 | 2021/11/17 | 21 | Prior Infection | 2020/10/21 | 13.58 | No | No | No | No |
| JJ (2x)-008 | F | 39 | 2021/12/06 | 2 doses Ad26.COV2.S | 2021/02/17 | 2021/11/11 | 25 | Prior Infection | 2020/07/31 | 16.21 | No | No | No | No |
| JJ (2x)-009 | M | 32 | 2021/12/09 | 2 doses Ad26.COV2.S | 2021/02/24 | 2021/11/18 | 21 | Prior Infection | 2020/05/31 | 18.31 | No | No | No | No |
| JJ (2x)-010 | F | 37 | 2021/12/06 | 2 doses Ad26.COV2.S | 2021/02/18 | 2021/11/12 | 24 | Prior Infection | 2021/01/07 | 10.95 | No | No | No | No |
| JJ (2x)-011 | F | 27 | 2021/12/08 | 2 doses Ad26.COV2.S | 2021/02/23 | 2021/11/17 | 21 | Prior Infection | 2020/08/14 | 15.81 | No | No | No | No |
| JJ (2x)-012 | F | 59 | 2021/12/06 | 2 doses Ad26.COV2.S | 2021/02/19 | 2021/11/15 | 21 | Prior Infection | 2021/01/07 | 10.95 | No | No | No | No |
| JJ (2x)-013 | M | 25 | 2021/12/06 | 2 doses Ad26.COV2.S | 2021/02/23 | 2021/11/12 | 24 | Prior Infection | 2021/01/04 | 11.05 | No | No | No | Yes |
| JJ (2x)-014 | M | 52 | 2021/12/08 | 2 doses Ad26.COV2.S | 2021/02/19 | 2021/11/16 | 22 | Prior Infection | 2020/06/02 | 18.21 | No | No | No | No |
| JJ (2x)-015 | F | 34 | 2021/12/08 | 2 doses Ad26.COV2.S | 2021/02/24 | 2021/11/15 | 23 | Prior Infection | 2020/06/01 | 18.24 | No | Yes | No | No |
| JJ (2x)-016 | F | 35 | 2021/12/09 | 2 doses Ad26.COV2.S | 2021/03/16 | 2021/11/17 | 22 | Prior Infection | 2020/06/18 | 17.72 | No | No | No | Yes |
| JJ (2x)-017 | F | 43 | 2021/12/13 | 2 doses Ad26.COV2.S | 2021/02/26 | 2021/11/24 | 19 | Prior Infection | 2020/12/24 | 11.64 | Yes | No | No | No |
| JJ (2x)-018 | F | 31 | 2021/12/13 | 2 doses Ad26.COV2.S | 2021/02/23 | 2021/11/22 | 21 | Prior Infection | 2020/12/21 | 11.74 | No | No | No | No |
| JJ (2x)-019 | F | 48 | 2021/12/07 | 2 doses Ad26.COV2.S | 2021/02/19 | 2021/11/15 | 22 | Prior Infection | 2020/04/28 | 19.33 | Yes | No | No | No |
| JJ (2x)-020 | F | 43 | 2021/12/06 | 2 doses Ad26.COV2.S | 2021/02/22 | 2021/11/17 | 19 | Prior Infection | 2021/01/07 | 10.95 | No | No | No | No |
| BNT (2x)-001 | F | 74 | 2021/07/30 | 2 doses BNT162b2 | 2021/06/22 | 2021/07/20 | 10 | No prior Infection | na | na | No | No | No | No |
| BNT (2x)-002 | F | 36 | 2021/08/03 | 2 doses BNT162b2 | 2021/06/03 | 2021/07/01 | 33 | No prior Infection | na | na | No | No | Yes | No |
| BNT (2x)-003 | M | 53 | 2021/08/11 | 2 doses BNT162b2 | 2021/04/23 | 2021/05/21 | 82 | No prior Infection | na | na | No | No | No | Yes |
| BNT (2x)-004 | F | 46 | 2021/07/26 | 2 doses BNT162b2 | Unk | 2021/05/05 | 82 | No prior Infection | na | na | No | No | Yes | No |
| BNT (2x)-005 | M | 74 | 2021/09/06 | 2 doses BNT162b2 | Unk | 2021/07/21 | 47 | No prior Infection | na | na | No | No | No | No |
| BNT (2x)-006 | F | 74 | 2021/09/06 | 2 doses BNT162b2 | Unk | 2021/07/20 | 48 | No prior Infection | na | na | No | No | No | No |
| BNT (2x)-007 | M | 53 | 2021/07/06 | 2 doses BNT162b2 | Unk | 2021/05/21 | 46 | No prior Infection | na | na | No | No | No | Yes |
| BNT (2x)-008 | M | 35 | 2021/08/02 | 2 doses BNT162b2 | Unk | 2021/07/19 | 14 | No prior Infection | na | na | No | No | Yes | No |
| BNT (2x)-009 | M | 71 | 2021/09/02 | 2 doses BNT162b2 | Unk | 2021/07/19 | 45 | No prior Infection | na | na | No | No | No | No |
| BNT (2x)-010 | F | 66 | 2021/10/12 | 2 doses BNT162b2 | Unk | 2021/08/10 | 63 | Prior Infection | 2020/07/01 | 15.39 | Yes | Yes | Yes | Yes |
| BNT (2x)-011 | F | 43 | 2021/09/02 | 2 doses BNT162b2 | Unk | 2021/08/11 | 22 | Prior Infection | 2020/06/03 | 14.99 | No | Yes | No | No |
| BNT (2x)-012 | M | 51 | 2021/10/06 | 2 doses BNT162b2 | Unk | 2021/09/06 | 30 | Prior Infection | 2020/10/01 | 12.16 | No | No | Yes | No |
| BNT (2x)-013 | F | 67 | 2021/08/16 | 2 doses BNT162b2 | Unk | 2021/07/19 | 28 | Prior Infection | 2020/08/03 | 12.43 | Yes | Yes | Yes | No |
| BNT (2x)-014 | F | 63 | 2021/08/18 | 2 doses BNT162b2 | Unk | 2021/07/23 | 26 | Prior Infection | 2020/08/04 | 12.46 | No | No | No | No |
| BNT (2x)-015 | F | 45 | 2021/10/05 | 2 doses BNT162b2 | Unk | 2021/09/21 | 14 | Prior Infection | 2020/07/04 | 15.06 | Yes | No | No | No |

Unk: unknown. Na: not applicable.

Wendy A. Burgers

# Reporting Summary

## Statistics

For all statistical analyses, confirm that the following items are present in the figure legend, table legend, main text, or Methods section.

| n/a | Confirmed | |
|---|---|---|
| ☐ | ☒ | The exact sample size (*n*) for each experimental group/condition, given as a discrete number and unit of measurement |
| ☐ | ☒ | A statement on whether measurements were taken from distinct samples or whether the same sample was measured repeatedly |
| ☐ | ☒ | The statistical test(s) used AND whether they are one- or two-sided *Only common tests should be described solely by name; describe more complex techniques in the Methods section.* |
| ☒ | ☐ | A description of all covariates tested |
| ☐ | ☒ | A description of any assumptions or corrections, such as tests of normality and adjustment for multiple comparisons |
| ☐ | ☒ | A full description of the statistical parameters including central tendency (e.g. means) or other basic estimates (e.g. regression coefficient) AND variation (e.g. standard deviation) or associated estimates of uncertainty (e.g. confidence intervals) |
| ☐ | ☒ | For null hypothesis testing, the test statistic (e.g. *F*, *t*, *r*) with confidence intervals, effect sizes, degrees of freedom and *P* value noted *Give P values as exact values whenever suitable.* |
| ☒ | ☐ | For Bayesian analysis, information on the choice of priors and Markov chain Monte Carlo settings |
| ☒ | ☐ | For hierarchical and complex designs, identification of the appropriate level for tests and full reporting of outcomes |
| ☒ | ☐ | Estimates of effect sizes (e.g. Cohen's *d*, Pearson's *r*), indicating how they were calculated |

*Our web collection on statistics for biologists contains articles on many of the points above.*

## Software and code

Policy information about availability of computer code

| Data collection | Clinical data were recorded by trained clinicians using RedCap (version 9.5.36) |
|---|---|
| Data analysis | Statistical analysis were performed using GraphPad Prism version 9.3 Flow. Cytometry data were analysed with FlowJo Software (10.8, FlowJo LLC, BD Life Sciences) and Pestle and Spice v6.1 (https://niaid.github.io/spice). HLA prediction was performed using TepiTool from IEDB Analysis Resource (http://tools.iedb.org). The quality control checks on raw sequence data and the genome assembly were performed using Genome Detective 1.133 (https://www.genomedetective.com). Phylogenetic classification of the genomes was done using the PANGOLIN software suite (v1.2.106) (https://github.com/hCoV-2019/pangolin). |

For manuscripts utilizing custom algorithms or software that are central to the research but not yet described in published literature, software must be made available to editors and reviewers. We strongly encourage code deposition in a community repository (e.g. GitHub). See the Nature Portfolio guidelines for submitting code & software for further information.

## Data

Policy information about availability of data

All manuscripts must include a data availability statement. This statement should provide the following information, where applicable:

- Accession codes, unique identifiers, or web links for publicly available datasets
- A description of any restrictions on data availability
- For clinical datasets or third party data, please ensure that the statement adheres to our policy

Datasets (raw data) underlying the figures have been provided as Source Data. Complete genome sequences for the viral isolates were deposited in GISAID.

# Field-specific reporting

Please select the one below that is the best fit for your research. If you are not sure, read the appropriate sections before making your selection.

☒ Life sciences          ☐ Behavioural & social sciences          ☐ Ecological, evolutionary & environmental sciences

For a reference copy of the document with all sections, see nature.com/documents/nr-reporting-summary-flat.pdf

# Life sciences study design

All studies must disclose on these points even when the disclosure is negative.

| | |
|---|---|
| Sample size | Sample size was based on available samples rather than on a pre-defined samples size calculation |
| Data exclusions | Sample (PBMC) with low viability (<60%) or low cell number (CD4+ T cells < 20,000 cells) were excluded from the analysis. |
| Replication | Samples for each patient were analysed once due to limited availability |
| Randomization | As this is a observational study, randomization is not applicable. |
| Blinding | For flow cyctometry assay, samples were stained and acquired in 7 consecutive runs over 2 weeks. While performing the experiments, investigators were blinded to patient groups. |

# Reporting for specific materials, systems and methods

We require information from authors about some types of materials, experimental systems and methods used in many studies. Here, indicate whether each material, system or method listed is relevant to your study. If you are not sure if a list item applies to your research, read the appropriate section before selecting a response.

## Materials & experimental systems

| n/a | Involved in the study |
|---|---|
| ☐ | ☒ Antibodies |
| ☐ | ☒ Eukaryotic cell lines |
| ☒ | ☐ Palaeontology and archaeology |
| ☒ | ☐ Animals and other organisms |
| ☐ | ☒ Human research participants |
| ☐ | ☒ Clinical data |
| ☒ | ☐ Dual use research of concern |

## Methods

| n/a | Involved in the study |
|---|---|
| ☒ | ☐ ChIP-seq |
| ☐ | ☒ Flow cytometry |
| ☒ | ☐ MRI-based neuroimaging |

## Antibodies

| | |
|---|---|
| Antibodies used | Antibodies<br>purified NA/LE mouse anti-human CD28 (clone 28.2) BD Pharmingen Cat# 555725; RRID:AB_2130052, dilution 1/1000<br>purified NA/LE mouse anti-human CD49d (clone L25) BD Pharmingen Cat# 555501; RRID:AB_396068, dilution 1/1000<br>LIVE/DEAD™ Fixable VIVID Stain Invitrogen Cat# L34955, dilution 1/2500<br>CD14 Pac Blue (clone TuK4) Invitrogen Thermofisher Scientific Cat# MHCD1428; RRID:AB_10373537, dilution 1/100<br>CD19 Pac Blue (clone SJ25-C1) Invitrogen Thermofisher Scientific Cat# MHCD1928; RRID:AB_10373689, dilution 1/100<br>CD4 PERCP-Cy5.5 (clone L200) BD Biosciences Cat# 552838; RRID:AB_394488, dilution 1/100<br>CD8 BV510 (clone RPA-8) Biolegend Cat# 301048; RRID:AB_2561942, dilution 1/100<br>CD3 BV650 (clone OKT3) Biolegend Cat# 317324; RRID:AB_2563352, dilution 1/100<br>IFN-g Alexa 700 (clone B27) BD Biosciences Cat# 557995; RRID:AB_396977, dilution 1/250<br>TNF BV786 (clone Mab11) Biolegend Cat# 502948; RRID:AB_2565858, dilution 1/100<br>IL-2 APC (clone MQ1-17H12) Biolegend Cat# 500310; RRID:AB_315097, dilution 1/100 |
| Validation | All antibodies used in this study are commercially available. All antibodies were validated by their manufacturers and were titered to define the optimal titer for positive and negative separation. |

# Eukaryotic cell lines

Policy information about <u>cell lines</u>

| | |
|---|---|
| Cell line source(s) | HEK293T-ACE2 cells were a gift from Dr Michael Farzan, Scripps, USA; H1299-E3 cell line was derived from H1299 (CRL-5803). H1299 cells were a gift from M. Oren, Weizmann Institute of Science. |
| Authentication | Cell lines were not authenticated |
| Mycoplasma contamination | Cell lines were tested for mycoplasma contamination and were mycoplasma negative. |
| Commonly misidentified lines (See <u>ICLAC</u> register) | None |

# Human research participants

Policy information about <u>studies involving human research participants</u>

| | |
|---|---|
| Population characteristics | This study includes participants vaccinated with Ad26.COV2.S or Pfizer BNT162b2, convalescent COVID-19 patients and hospitalized COVID-19 patients. Samples were selected based on PBMC availability. <br> Demographic characteristics are presented in Extended Data Table 1. Clinical characteristics for each participants included in the study are presented in Extended Data Table 3 and 4. |
| Recruitment | Study participants vaccinated with Ad26.COV2.S. and convalescent donors were included in a prospective cohort study conducted at Groote Schuur Hospital (Western Cape). Pfizer BNT162b2 vaccinees were recruited from KwaZulu-Natal. Hospitalized COVID-19 patients were recruited from Groote Schuur and Tshwane hospitals. All participants were older then 18 years old, and all participants gave written informed consent. One bias that may be present is the ethnic background, aince ethnic background differs substantially between the different South African provinces. This could have implications for the prevalence of HLA class I and II molecules between patients recruited from different provinces. |
| Ethics oversight | The study was approved by the University of Cape Town Human Research Ethics Committee (ref: HREC 190/2020, 207/2020 and 209/2020) and the University of the Witwatersrand Human Research Ethics Committee (Medical) (ref. M210429 and M210752), the Biomedical Research Ethics Committee at the University of KwaZulu–Natal (ref. BREC/00001275/2020) and the University of Pretoria Health Sciences Research Ethics Committee (ref. 247/2020). |

Note that full information on the approval of the study protocol must also be provided in the manuscript.

# Clinical data

Policy information about <u>clinical studies</u>
All manuscripts should comply with the ICMJE <u>guidelines for publication of clinical research</u> and a completed <u>CONSORT checklist</u> must be included with all submissions.

| | |
|---|---|
| Clinical trial registration | *Provide the trial registration number from ClinicalTrials.gov or an equivalent agency.* |
| Study protocol | *Note where the full trial protocol can be accessed OR if not available, explain why.* |
| Data collection | *Describe the settings and locales of data collection, noting the time periods of recruitment and data collection.* |
| Outcomes | *Describe how you pre-defined primary and secondary outcome measures and how you assessed these measures.* |

# Flow Cytometry

## Plots

Confirm that:

☒ The axis labels state the marker and fluorochrome used (e.g. CD4-FITC).

☒ The axis scales are clearly visible. Include numbers along axes only for bottom left plot of group (a 'group' is an analysis of identical markers).

☒ All plots are contour plots with outliers or pseudocolor plots.

☒ A numerical value for number of cells or percentage (with statistics) is provided.

## Methodology

| | |
|---|---|
| Sample preparation | Blood was collected in heparin tubes and processed within 4 hours of collection. Peripheral blood mononuclear cells (PBMC) were isolated by density gradient sedimentation using Ficoll-Paque (Amersham Biosciences, Little Chalfont, UK) as per the manufacturer's instructions and cryopreserved in freezing media consisting of heat-inactivated fetal bovine serum (FBS, Thermofisher Scientific) containing 10% DMSO and stored in liquid nitrogen until use. |

| Instrument | BD Fortessa |
|---|---|
| Software | Flowjo V10.8.1, Pestle and Spice V6.1 |
| Cell population abundance | No cell sorting was performed |
| Gating strategy | SARS-Cov-2-specific T cells were identified via the following gating strategy: Viable lymphocytes were identified by successive gating in singlets (FSC-A/FSC-H), time gate, live CD3 (SSC-A/ DUMP channel- dead cells, CD14, CD19,). Then, from live CD3+ T cells, CD4+ and CD8+ T cells were gated in CD4/CD8 plots. Next, SARS-CoV-2 -specific T cells were gated by plotting IFN-g, IL-2 and TNF-a |

☒ Tick this box to confirm that a figure exemplifying the gating strategy is provided in the Supplementary Information.

