## [Peer Review File · Nature]

Manuscript Title: T cell responses to SARS-CoV-2 spike cross-recognize Omicron

Reviewer Comments & Author Rebuttals

Reviewer Reports on the Initial Version:

Referee #1:

In the paper called, "SARS-CoV-2 spike T cell responses induced upon omicron vaccination or infection remain robust against omicron", Keeton et al report CD4 and CD8 T cell responses against ancestral or Omicron spike peptide pools in donors who were either one or two dose J&J (Ad26-COV.S) or two dose Pfizer (BNT162b2) vaccinated, or convalescent after infection during the ancestral Wuhan Hu-1, Beta, Delta or Omicron infection waves in South Africa.

This study is timely and attempts to address an important research question. However, I have some reservations about the cohorts studied, methodology, analysis and interpretation.

Study cohorts

Small study groups, incomplete and missing data about study groups, timing of sampling and missing control groups for interpretation of the T cell studies.

- The vaccine and Omicron infected convalescent cohorts shown in Fig. 1 have very small n numbers (Ad26-COV.S one dose, n=20, Ad26-COV.S two dose, n=20, BNT162b2 two dose, n=15 and unvaccinated convalescent, n=15).
- The wave 1 to 4 hospitalized cohorts shown in Fig. 2 have very small n numbers [wave (ancestral) 1, n=17, wave 2 (beta), n=16, wave 3 (delta), n= 16 and wave 4 (omicron), n=19].
- We are not told the vaccination history of participants included in the hospitalized cohorts. It seems from Supplementary Table 3 that these participants were recruited 2-6d after becoming PCR positive? Also, the Wave 2 (Beta) cohort have a much lower proportion of individuals with severe disease (WHO >5) (6% compared with 38-56%). Higher T cell responses are linked to more severe disease in hospitalized cohorts making the groups poorly matched for comparative analysis?
- An additional control group is missing and needed to determine if there is any pre-existing T cell immune recognition of the ancestral, beta, delta or omicron peptide libraries used. The control groups required are:
 - 1) Pre-pandemic controls (n=20) recruited before 2019 from the same geographic region who could not have been exposed to SARS-CoV-2
 - 2) Uninfected, unvaccinated controls (n=20) recruited after March 2020 from the same geographic region.

The numbers in the vaccinated, convalescent and hospitalized study cohorts for Fig. 1 & 2 need to be increased and the timing of blood sampling for the hospitalised cohorts should be clearly stated in the manuscript text and need to be >14-21d after the acute SARS-CoV-2 infection.

Missing or incomplete data to support the previous SARS-CoV-2 infection history during the first (Wuhan), second (Beta) and third (Delta) waves making interpretation of the results difficult

The vaccination history should be included in Supplementary Table 3. Unvaccinated convalescent, vaccinated and hospitalized groups (wave 1-4) all require more detailed objective longitudinal data to support the participants previous SARS-CoV-2 infection, re-infection and breakthrough infection

history up to December 2021. This could be presented in a Table format to support the definition of their previous infection history during the ancestral Wuhan first wave, beta second wave and delta third wave. The data currently presented in the Methods section and Supplementary Tables is incomplete. Without this information it is simply not possible to interpret the findings in terms of nAb and/or T cell responses. There have been several papers published showing augmented antibody and T cell responses following vaccination in the context of previous infection (including from the authors of this manuscript).

This is especially important in the wave 4 (omicron) group in a geographical region (South Africa) where, as the authors state, >60% of the population have been previously infected with SARS-CoV-2 during the first to third waves. There needs to be additional evidence to support the lack of a previous SARS-CoV-2 infection history in this wave 4 (omicron) group in order to be able to interpret the results. For example, it has been proposed by others that the population in SA has been partly protected from severe disease by the relatively high levels of previous natural infection.

In the current wave 4 (omicron) group of n=19, 7 were identified by S gene target failure on PCR suggestive of omicron infection and further 5 had isolates confirmed by whole genome sequencing (but data is not included). The remaining 7 subjects were assumed to have omicron based solely on the timing of their infection. It would be relatively straightforward to recruit and additional PCR confirmed Omicron cases with longitudinal data to support a lack of previous infection to include in the analysis.

Methodology

Overarching comment: the study draws on individuals with different medical histories, infections and vaccines and sampled at different times. Even referring to the supplementary materials, it was hard to track these between figures to allow the reader a proper chance to gauge key issues such as timing of infection and vaccination history.

The study utilises peptide libraries and flow cytometry to examine the T cell responses. The data from these T cell assays are interesting and informative. Additional information is required regarding the peptides contained in the libraries used. The individual peptides contained in all the libraries should be listed in the supplementary material. For omicron studies it is especially important to report details of peptide panels in some detail (aa sequence as well as length, overlap, number in the pool) as some of the S mutations are clustered together and a single peptide can carry more than one altered T cell epitope, or indeed, get spliced within a variant stretch.

The data reported for CD8 responses is problematic as the peptide libraries used in the study consisted of 15mers. Normally, to specifically study CD8 T cell responses peptide libraries containing peptides that are 9-10aa long are used and peptides that are 15-20aa long are used for CD4 T cell studies. This means that the data shown in Fig. 1g,h, Fig. 2d, Extended Data Fig. 4 and the RHS panels of Extended Data Fig. 5 & Extended Data Fig. 6 should either be removed, repeated using peptide libraries designed to specifically explore a CD8 T cell response.

The use of peptide libraries in the methodology means that the data as presented will by definition include T cell responses against cryptic epitopes never seen in real life settings. So while the peptide libraries used contain peptides of an appropriate aa length (ie 15mer) to demonstrate CD4 T cell responses, some of the peptide epitopes would not be naturally presented. It would, therefore, be important to repeat the CD4 T cell studies with spike protein (ancestral and containing the omicron mutations and deletions) using an ELISpot approach. In this way the spike will be processed by APCs and epitopes presented via the class II pathway to CD4 T cells.

Analysis

The data from the T cell assays are interesting and informative, drawn from analysis of the response to large peptide libraries of the whole ancestral, Delta and Omicron spike protein in the vaccine studies and Wuhan-1 SARS-CoV-2 N, M and S proteins in hospitalized patient studies.

The overall T cell response will be the summation of many T cell responses, in individuals of differing HLA types, responding or not responding to a proportion of many peptides. It is, therefore, not surprising and entirely predicted from the biology that the T cell response overall will be reduced, but

will remain broadly intact overall. There are after all only 38 described mutations in the spike region of the Omicron VOC. This makes the analysis reported here (with incredibly small group numbers) hard to analyze as it is impossible to take account of the many HLA differences between individuals and the impact that this has on their T cell responses. The study needs larger n numbers.

In this context, it seemed perhaps unhelpful (or at least, unconventional) to invoke the very processed-data concept of '70-80% of the CD4 and CD8 T cell response to spike is maintained'. This could mislead those who are unfamiliar with T cell analysis and has virtually no meaning in terms of immune correlates of protection. We still know relatively little about immunodominant epitopes and even less about T cell correlates of protection. This study shows that there are T cell responses present, but there is no data presented here to show that the presence or absence of any such T cell response protects against severe disease and/or death in vaccinated individuals or breakthrough infection. It is important to distinguish between association and causality.

It is unconventional and possibly incorrect to present fold-changes in omicron and ancestral spike median T cell frequency as shown in Fig. 1f,h and Extended Data Fig. 5. This does not have the equivalent functional meaning of a fold change in nAb ID50. It is reasonable to present a reduction in median frequency as in Fig. 1e,h. Please can the authors clarify the statistical test used in Fig. 1e for the paired analysis. Please can the authors consider removing Fig. 1f,g and simply comment on the fall in T cell frequency in the text.

In Fig. 1 the T cell responses were measured approximately one month (range 21 to 64 days) after the most recent vaccine dose. As such, the data very much offers a best-case scenario of very recently vaccinated individuals. The authors should analyze T cell responses at later timepoints as well – much of the concern about breakthrough infection in vaccinees and associated drop in antibody neutralization has been in relation to individuals 6 months or more after their vaccine second dose when the antibody responses have waned.

In Extended Fig 2 the authors explore neutralizing activity against Omicron using plasma from Pfizer or J&J vaccinees. The findings are in line with recent papers and preprints. The writing is imprecise in referring to ability to escape 'the majority of SARS-CoV-2 antibodies.' The precise meaning is unclear: the plasma analyzed would indeed have encompassed many thousands of different anti-S IgG sequences, but no analysis was reported at the level of individual antibodies. In Extended Data Fig. 2a,b the authors should consider removing the fold-change data as it is meaningless to compare fold changes in nAb ID50 and T cell frequencies side-by-side in this way. It's like comparing apples with elephants! (a 10-fold drop in ID50 has widely understood and accepted implications in terms of potential protective correlates, whereas here a largely maintained T cell frequency to a peptide panel is measured on an unrelated scale and confers no protective implication). Also, please can the data for nAb ID50 in Extended Data Fig. 2b be plotted separately for prior infection and no prior infection as it is hard to follow as currently presented.

The data presented in Figure 2 is difficult to interpret for the methodological reasons cited above (lack of longitudinal previous infection history / vaccination history / differences in disease severity between groups / timing of the blood draw). Please can the authors clarify the number of non-responders in Fig. 2c,d. The Introduction cites refs 14-17 in support of the sensible tenet that T cell immunity is likely to modulate COVID-19 severity; it might be useful to word this sentence with greater precision in terms of the specific content of 14-17. The data in Figure 2 and associated extended data offer a potentially rich and informative dataset (if the issues described above can be resolved). T cell responses are compared between convalescent patients from the Wuhan Hu-1, beta, delta and omicron waves. There are many interesting points from these studies that may merit further analysis and discussion. In Figure 2C, it looks as if those infected in the Omicron wave have a T cell response to N that is more impaired than to S or M. It would be valuable to dissect this out at an epitope level. In 2e, it looks as if Omicron-infected people respond very similarly to the ancestral or Omicron peptide pools. This offers a slightly different window onto the epitope conservation question from the data presented in Figure 1, producing a different answer?

Extended data Fig 3 offers interesting analysis of polyfunctionality in the CD4 Omicron T cell

responses, again showing largely conserved recognition and response capacity. However, the case for polyfunctionality as a proxy for T cell affinity had been overstated and no data has been presented showing affinity / avidity measurements in this manuscript.

Interpretation

The paper is timely and complements the recent wave of papers on diminished Omicron neutralization. The wording of the title is not precise. Referring to T cell responses that 'remain robust' is unscientific and misleading: the intention was perhaps to imply a strength of response that could be protective (as subsequently discussed). What the paper actually shows is that, set in the context of all T cell recognition that is intact and detectable, the loss of T cell recognition appears less profound than that shown with respect to antibody evasion.

The Abstract ends on unsupported speculation: it's unclear why these immune findings should be specifically and causally related to clinical observations of mild Omicron disease in South Africa and elsewhere.

Referee #2:

The manuscript of Keeton et al. aims to characterize the impact that Spike Omicron mutations exert on Spike-specific T cells induced by vaccinations with Adenovirus- or mRNA-based vaccines (n=55) or by infection (15 convalescents). Furthermore, the authors tested the ability of Omicron infection (in 19 patients) to induce T cells against Spike, Np and membrane (ancestral sequence) in comparison to other patients (~ 50 patients) infected by other VoCs.

T cell data were generated by PBMC activation with peptide pools covering the Spike protein of the Wuhan strain and of the Omicron VoC and NP and membrane proteins of the Wuhan strain. Intracellular cytokine staining was used to evaluate the quantity of CD8 and CD4 T cells responding to the distinct peptide pools and their cytokine production (IFN-gamma, IL-2 and TNF-alpha).

The results show that Omicron Spike mutations do not suppress the majority of T cells induced by vaccines or infection and that Omicron infection triggers a T cell response against different structural proteins similar to other SARS-CoV-2 VoCs. These data are novel, important and timely.

In light of the fast world-wide diffusion of Omicron and of the profound impact that Omicron mutations have on the neutralizing ability of antibodies, knowledge related to the impact that these mutations have on T cells induced by vaccination is of great interest and importance. Having said this, the results presented are really only limited to the frequency of T cells stimulated by the different peptide pools. There are no attempts to define, for example, why T cells in the majority of vaccinated individuals tolerate Omicron mutations. Is this due to the fact that T cells are always targeting conserved regions of Spike, or it is because the mutations are present within the epitopes but they don't have any effect on HLA-class I and class II binding or in TCR recognition?

In addition, the discussion (and the abstract) depicts only a very positive scenario: a "well-preserved T cell immunity to Omicron". Data in the majority of individuals tested support this conclusion, but the data reported also that in 5 of the 32 (15%) vaccinated individuals, Omicron mutations appear to completely abolish CD8 T cell recognition. 15% is not a completely insignificant number and I think the authors should point out that a quantifiable number of individuals might experience a loss of T cell response, which might have some virological (high and long viral spread?) and perhaps even pathological consequences. It might be time to show that biology is not only black or white but some grey exists.

In normal circumstances, this reviewer would have thought that defining the epitopes and the restriction elements of the CD8 T cells that are completely inhibited by the Omicron mutations represent a logic and indispensable part of a research work investigating the effect of mutations on T cells. Such information is not only scholarly important but has practical consequences since, for

example, this information could link specific HLA-class I profiles of individuals to increased susceptibility of prolonged Omicron infection.

It is however clear that definition of T cell epitopes and of their HLA-class I restrictions requires time that might severely delay the publication of the initial observation of the preserved T cell reactivity in the majority of vaccinated individuals.

2) The observation about the ability of Omicron infection to elicit a T cell response targeting structural proteins of SARS-CoV-2 of similar magnitude to what can be observed in individuals infected by others VoCs is novel and of interest. Clearly, also here it would have been nice to define a little better the quality of T cells in particular whether the T cells induced by Omicron infection target conserved regions.

3) Limitations: the authors list some limitations, like the lack of utilization of an AIM assay to confirm and better define T cell responses and the single use of 15-mers. This is ok but perhaps, more related to the focus of the work that aims to define the impact of mutations on T cell responses, the authors should also list and discuss other potential limitations. First, they should point out that all their experiments were performed with a robust concentration of peptides (1ug/ml). The use of such high concentration might underestimate the impact of mutations on T cells. Second, it must be highlighted that mutations outside the epitopes might also have an effect on processing and presentation. As such, testing T cells only by utilizing peptides might underestimate the impact of mutations on the T cell response. These limitations should be included.

Referee #3:

Keeton et al studied immune responses towards the Omicron variant. The authors assessed T cell responses, in context of neutralizing antibodies, in individuals who were vaccinated with Ad26.CoV2.S and BNT162b2 as well as in unvaccinated convalescent COVID-19 patients. The main findings are that 70-80% of the spike-specific CD4+ and CD8+ T cell responses cross-react with the Omicron variant, similar to the cross-reactivity observed against Delta and Beta variants. This is a well performed study, manuscript is written clearly and the data are presented in a nice and accessible way. The study provides key insights into pre-existing T cell immunity to the Omicron variant, established either by prior vaccination or infection.

Specific comments:

Data presented in Fig 2cd show that a proportion of COVID-19 patients have not elicited T cell responses. What were the antibody levels in those patients? Did COVID-19 patients with non-detectable T cell responses seroconvert? Was the time post disease onset or disease severity different amongst patients who lacked T cell responses?

The authors should at least comment on the elevated antibody levels towards the Omicron variant following the third vaccine booster.

'CD4' should be replaced with 'CD4+' and 'CD8' should be replaced with 'CD8+' T cells

The authors speculate that specific HLA molecules can be adversely affected by mutations in particular CD8 epitopes, which seems like a possible explanation. Were the participants HLA typed so this could be explored further?

Extended Fig 1a: frequencies should be added to the FACS plots.

Extended Fig 6a: should be 'specific' rather than 'spe'.

The authors used PepTivator, Miltenyi Biotech peptide pools. According to the Miltenyi website, not all Spike peptide pools cover the entire Spike protein. Please clarify whether the entire Spike protein was covered by the overlapping spike peptides. The authors state that they combined S1 peptide pool and "the majority of the C-terminal S2 domain".

From Miltenyi website: "The PepTivator SARS-CoV-2 Prot_S covers selected immunodominant sequence domains of the spike protein (aa 304–338, 421–475, 492–519, 683–707, 741–770, 785–802, and 885–1273). In contrast, PepTivator SARS-CoV-2 Prot_S Complete covers all functional domains (aa 5–1273), Prot_S1 the complete N-terminal S1 domain (aa 1–692) and Prot_S+ parts of the C-terminal S2 domain (aa 689–895). The complete S2 domain (and parts of the S1 domain: aa 304–338, 421–475, and 492–519) is covered, when PepTivator SARS-CoV-2 Prot_S and Prot_S+ are combined."

Author Rebuttals to Initial Comments:

Referees' comments:

Referee #1 (Remarks to the Author):

In the paper called, "SARS-CoV-2 spike T cell responses induced upon omicron vaccination or infection remain robust against omicron", Keeton et al report CD4 and CD8 T cell responses against ancestral or Omicron spike peptide pools in donors who were either one or two dose J&J (Ad26-COV.S) or two dose Pfizer (BNT162b2) vaccinated, or convalescent after infection during the ancestral Wuhan Hu-1, Beta, Delta or Omicron infection waves in South Africa.

This study is timely and attempts to address an important research question. However, I have some reservations about the cohorts studied, methodology, analysis and interpretation.

Study cohorts

1.1. Small study groups, incomplete and missing data about study groups, timing of sampling and missing control groups for interpretation of the T cell studies.

- The vaccine and Omicron infected convalescent cohorts shown in Fig. 1 have very small n numbers (Ad26-COV.S one dose, n=20, Ad26-COV.S two dose, n=20, BNT162b2 two dose, n=15 and unvaccinated convalescent, n=15).
- The wave 1 to 4 hospitalized cohorts shown in Fig. 2 have very small n numbers [wave (ancestral) 1, n=17, wave 2 (beta), n=16, wave 3 (delta), n= 16 and wave 4 (omicron), n=19].

>>Response: We evaluated a total of 138 patients in this study. Cross-reactivity to Omicron spike was measured in 55 vaccinees and 15 unvaccinated convalescents. When vaccinees were delineated according to vaccine type, results were remarkably consistent across the groups, with a similar median decrease in T cell frequencies to Omicron compared to ancestral peptide reagents. Since our submission, six preprints have been posted that concur with our results of well-preserved T cell responses against Omicron (using sample sizes comparable to ours), as follows:

Authors	Source	Date posted	Total n	Sub-gps	n/group	Setting	Groups	Assay
De Marco et al (Borsellino)	MedRxiv	30-Dec-21	61	5	10-15	EU	Vaccinated / Prior infection + vax / Vax + infection	AIM
Gao et al (Bruggert)	Research Square	03-Jan-22	103	3	15-40	EU	Vaccinated / convalescent / seronegative	AIM + ICS
GeurtsvanKessel et al (De Vries)	MedRxiv	29-Dec-21	100	8	5-15	EU	Vaccinated	AIM
Liu et al (Barouch)	MedRxiv	03-Jan-22	51	2	20-31	US	Vaccinated	Elispot + ICS
Naranbhai et al (Gaiha)	MedRxiv	05-Jan-22	76	3	11-31	US	Vaccinated / Prior infection + vax / Prior infection	Elispot + proliferation
Tarke et al (Sette)	BioRxiv	28-Dec-21	17	1	17	US	Vaccinated	AIM
OUR STUDY	MedRxiv	28-Dec-21	138	8	15-20	Africa	Vaccinated / Convalescent / Omicron-infected	ICS

We also compared 19 hospitalised patients in the Omicron wave to 49 hospitalised patients infected with other variants. It was of interest to delineate the patients according to variant infection, and despite differences in age, disease severity, co-morbidities and timing, there were broadly similar ranges of responses across the groups.

Thus, we assert that the internal consistency of our data, and independent corroboration from a sizeable number of emerging studies from different settings using different T cell assays, would argue that increasing the n would not alter our conclusions, and that our data are well-supported by the number of participants studied.

1.2. We are not told the vaccination history of participants included in the hospitalized cohorts. It seems from Supplementary Table 3 that these participants were recruited 2-6d after becoming PCR positive? Also, the Wave 2 (Beta) cohort have a much lower proportion of individuals with severe disease (WHO >5) (6% compared with 38-56%). Higher T cell responses are linked to more severe disease in hospitalized cohorts making the groups poorly matched for comparative analysis?

>>Response: We have now added a detailed Extended Data Table 5, with individual patient data that contains vaccination history and all other relevant details. All hospitalised patients from wave 1, 2 and 4 were unvaccinated. Third wave participants with known vaccination status were all unvaccinated (n=8), and the remainder (n=8) had unknown vaccination status. These details have been added to the Methods section (page 16).

With regard to T cell responses being linked to more severe disease: we have previously shown that the frequency of SARS-CoV-2-specific CD4 T cells in hospitalised patients was comparable irrespective of disease severity (defined based on the WHO ordinal scale) (Riou et al, JCI 2021 PMC8203446). Additional studies also present a more nuanced link between disease severity and T cell responses in hospitalised patients, which is influenced by multiple factors including co-morbidities and age (Sattler et al, JCI 2020 PMID:32833687; Neidleman et al, 2021 PMID:34260965; Nielsen et al, 2021 PMC8176920). The intention of the analysis in Figure 2 was to compare variant responses across waves in patients with a range of ages, disease severity, co-morbidities and time since diagnosis (which was not time since start of symptoms, which was not known for most patients).

1.3. An additional control group is missing and needed to determine if there is any pre-existing T cell immune recognition of the ancestral, beta, delta or omicron peptide libraries used. The control groups required are:

- 1) Pre-pandemic controls (n=20) recruited before 2019 from the same geographic region who could not have been exposed to SARS-CoV-2
- 2) Uninfected, unvaccinated controls (n=20) recruited after March 2020 from the same geographic region.

>>Response: As stated by the reviewer, it is well-established that cross-reactive SARS-CoV-2 T cell responses are found in a proportion of the population who have not been exposed to SARS-CoV-2, from studies of pre-pandemic samples (Braun et al. 2020; Grifoni et al. 2020; Mateus et al. 2020). Indeed, we also identified ancestral Spike-specific responses in a minority of COVID-naïve individuals prior to vaccination (Keeton et al. 2021). These responses have been attributed to exposure to endemic human coronaviruses, and reactivity is more common for non-spike antigens than spike (Grifoni et al. 2020). Whilst we could perform such experiments, the addition of these control groups would add little value to the current study, and neither alter nor strengthen our conclusions.

1.4. The numbers in the vaccinated, convalescent and hospitalized study cohorts for Fig. 1 & 2 need to be increased and the timing of blood sampling for the hospitalised cohorts should be clearly stated in the manuscript text and need to be >14-21d after the acute SARS-CoV-2 infection.

>>**Response:** We have addressed the issue of increasing the numbers in point 1.1 above, and we have added timing of blood sampling and details at an individual patient level in the new Extended Data Tables 5 and 6). As mentioned, for hospitalised patients we have included time since diagnosis, which followed the onset of symptoms, the date of which was not known for most patients.

1.5. Missing or incomplete data to support the previous SARS-CoV-2 infection history during the first (Wuhan), second (Beta) and third (Delta) waves making interpretation of the results difficult

>>**Response:** We have addressed the issue of missing or incomplete clinical data in point 1.2 above, with the inclusion of a detailed Extended Data Table 5 including infection history, to better interpret the results (see Methods section, page 16).

1.6. The vaccination history should be included in Supplementary Table 3. Unvaccinated convalescent, vaccinated and hospitalized groups (wave 1-4) all require more detailed objective longitudinal data to support the participants previous SARS-CoV-2 infection, re-infection and breakthrough infection history up to December 2021. This could be presented in a Table format to support the definition of their previous infection history during the ancestral Wuhan first wave, beta second wave and delta third wave. The data currently presented in the Methods section and Supplementary Tables is incomplete. Without this information it is simply not possible to interpret the findings in terms of nAb and/or T cell responses. There have been several papers published showing augmented antibody and T cell responses following vaccination in the context of previous infection (including from the authors of this manuscript).

>>**Response:** We have addressed the issue of additional detail on the clinical data in point 1.2 above, with the inclusion of detailed Extended Data Tables 5 and 6, including vaccination and infection history, to better interpret the results.

1.7. This is especially important in the wave 4 (omicron) group in a geographical region (South Africa) where, as the authors state, >60% of the population have been previously infected with SARS-CoV-2 during the first to third waves. There needs to be additional evidence to support the lack of a previous SARS-CoV-2 infection history in this wave 4 (omicron) group in order to be able to interpret the results. For example, it has been proposed by others that the population in SA has been partly protected from severe disease by the relatively high levels of previous natural infection.

>>**Response:** All wave 4 participants included in our study were unvaccinated and had no clinical record of previous symptomatic COVID-19. These details are now included in the additional detailed Extended Data Table 5 provided and in the Methods section.

1.8. In the current wave 4 (omicron) group of n=19, 7 were identified by S gene target failure on PCR suggestive of omicron infection and further 5 had isolates confirmed by whole genome sequencing (but data is not included). The remaining 7 subjects were assumed to have omicron based solely on the timing of their infection. It would be relatively

straightforward to recruit and additional PCR confirmed Omicron cases with longitudinal data to support a lack of previous infection to include in the analysis.

>>**Response:** We believe that it is reasonable to assume that the majority of patients recruited during the Omicron wave were infected with Omicron. A further two swabs were located and confirmed to be SGTF so we have amended the text to 9/19 SGTF, and this is regarded as an acceptable proxy for Omicron. We have removed the reference to the whole genome sequencing confirmation of 5 of the samples, since whilst they were assigned as Omicron by Nextclade, the assemblies had multiple frameshifts (poor sequencing quality) and were rejected by GISAID. Wave 4 did not occur with a concomitant Delta wave in South Africa as has occurred elsewhere, but was driven by the Omicron variant, and the prevalence of Omicron in South Africa at the time of recruitment was >90% (see Fig. 2b). Moreover, in Tshwane from where the remainder of the samples originated, Omicron was responsible for 98% of infections sequenced at the time of sampling in our study (determined from 60/61 samples sequenced). We have corrected the text and elaborated on this in the Methods (page 15).

Methodology

1.9. Overarching comment: the study draws on individuals with different medical histories, infections and vaccines and sampled at different times. Even referring to the supplementary materials, it was hard to track these between figures to allow the reader a proper chance to gauge key issues such as timing of infection and vaccination history.

>>**Response:** As referred to in point 1.1, 1.5 and 1.6, we have included Extended Data Tables 5 and 6 with individual participant details, including timing of infection and vaccination history.

The study utilises peptide libraries and flow cytometry to examine the T cell responses. The data from these T cell assays are interesting and informative. Additional information is required regarding the peptides contained in the libraries used. The individual peptides contained in all the libraries should be listed in the supplementary material. For omicron studies it is especially important to report details of peptide panels in some detail (aa sequence as well as length, overlap, number in the pool) as some of the S mutations are clustered together and a single peptide can carry more than one altered T cell epitope, or indeed, get spliced within a variant stretch.

>>**Response:** We have added an Extended Data Table indicating all individual peptide sequences for Ancestral, Beta, Delta and Omicron peptide libraries (Extended Data Table 7). The methods section includes additional detail of peptide panels (page 16):

“There were a total of 253 peptides in the Ancestral, Beta and Delta variant pool, and 254 peptides in the Omicron pool.”

1.10. The data reported for CD8 responses is problematic as the peptide libraries used in the study consisted of 15mers. Normally, to specifically study CD8 T cell responses peptide libraries containing peptides that are 9-10aa long are used and peptides that are 15-20aa long are used for CD4 T cell studies. This means that the data shown in Fig. 1g,h, Fig. 2d, Extended Data Fig. 4 and the RHS panels of Extended Data Fig. 5 & Extended Data Fig. 6 should either be removed, repeated using peptide libraries designed to specifically explore a

CD8 T cell response.

>>**Response:** The use of 15mer peptides is an approach widely used to screen for CD8 T cell responses (a few examples cited below). Peptides of 15 amino acids in length are slightly less efficient at activating CD8 T cells than 9-10mers, but is a broadly acceptable compromise for reasons of cost and practicality (in particular, enabling CD4 and CD8 responses to be measured in the same sample given the large cell numbers required for cellular immunology assays). Kiecker et al. 2004 (PMID: 15172453) estimated that 15mers capture 77% of the frequency of the CD8 response compared to shorter peptides. Thus, 15mer peptides still capture the majority of the CD8 response, exemplified by the abundant CD8 responses we detect in our study. The assertion that the CD8 data is invalid since we used 15mers and the request that we remove all CD8 data from the paper or repeat the study using 9-10mers is rather surprising and we believe quite an unreasonable request.

We did raise the issue of peptide length as a limitation in our discussion:

“The use of 15mer peptides may also have underestimated SARS-CoV-2-specific CD8 T cells, as shorter peptides are more optimal for HLA class I binding³⁹.”

A selection of the many studies that use 15mers to screen for CD8 responses:

Goel et al, Science 2021; Guerra et al, Science Immunology 2021; Barouch et al, NEJM 2021, Alter et al Nature 2021; Cohen et al, Cell Reports Med 2021.

In addition, the six emerging studies on Omicron T cell cross-reactivity we referred to earlier also used 15mer peptide sets and assessed CD8 T cell responses (De Marco et al, 2021; Gao et al, 2021, GeurtsvanKessel et al, 2021; Liu et al, 2021, Naranbhai et al, 2021, Tarke et al, 2021).

1.11. The use of peptide libraries in the methodology means that the data as presented will by definition include T cell responses against cryptic epitopes never seen in real life settings. So while the peptide libraries used contain peptides of an appropriate aa length (ie 15mer) to demonstrate CD4 T cell responses, some of the peptide epitopes would not be naturally presented. It would, therefore, be important to repeat the CD4 T cell studies with spike protein (ancestral and containing the omicron mutations and deletions) using an ELISpot approach. In this way the spike will be processed by APCs and epitopes presented via the class II pathway to CD4 T cells.

>>**Response:** We used peptides to assess T cell responses and identify mutations in Omicron that affect HLA binding (epitope presentation) and T cell recognition (TCR contacts), whereas mutations in Omicron that affect peptide processing would only be identified with the use of whole protein antigens that have undergone appropriate intracellular processing. This is certainly an additional way in which Omicron mutations could affect T cells. However, not having data on the potential for Omicron mutations to affect processing does not minimise the important body of data on epitope recognition that we do present, addressing HLA binding and T cell recognition, and we feel that identifying potential processing mutations is beyond the scope of the current study. We have highlighted this point in the discussion, that further work could be performed to assess the range of ways Omicron’s mutations may affect T cell responses. We have added the following in the Discussion section:

“In addition, the use of peptides does not permit us to define the potential effect of mutations on antigen processing, thus underestimating the impact of Omicron mutations on T cell cross-recognition”.

Analysis

The data from the T cell assays are interesting and informative, drawn from analysis of the response to large peptide libraries of the whole ancestral, Delta and Omicron spike protein in the vaccine studies and Wuhan-1 SARS-CoV-2 N, M and S proteins in hospitalized patient studies.

1.12. The overall T cell response will be the summation of many T cell responses, in individuals of differing HLA types, responding or not responding to a proportion of many peptides. It is, therefore, not surprising and entirely predicted from the biology that the T cell response overall will be reduced, but will remain broadly intact overall. There are after all only 38 described mutations in the spike region of the Omicron VOC. This makes the analysis reported here (with incredibly small group numbers) hard to analyze as it is impossible to take account of the many HLA differences between individuals and the impact that this has on their T cell responses. The study needs larger n numbers.

>>Response: We agree wholeheartedly with the reviewer, that the experimental data we present confirms what we would have predicted with respect to the effect of only 38 mutations in the Omicron spike. We have responded to the issue of increased n in points 1.1 and 1.4. We are heartened that six additional studies that have appeared as preprints since our submission have come to the same conclusion of preserved cross-reactivity to spike. Thus, despite the (very likely) many HLA differences between individuals and groups that will shape the magnitude and specificity of their T cell responses, both in our study and those from North America and Europe, there is a remarkable consistency in the reduction in the response to Omicron compared to ancestral spike. This consistency, generated from 55 vaccinees and 15 convalescents, and the experimental demonstration of what would be predicted, argues that the number of participants in our study is sufficient to observe reliable changes in the cross-reactivity to Omicron.

1.13. In this context, it seemed perhaps unhelpful (or at least, unconventional) to invoke the very processed-data concept of '70-80% of the CD4 and CD8 T cell response to spike is maintained'. This could mislead those who are unfamiliar with T cell analysis and has virtually no meaning in terms of immune correlates of protection. We still know relatively little about immunodominant epitopes and even less about T cell correlates of protection. This study shows that there are T cell responses present, but there is no data presented here to show that the presence or absence of any such T cell response protects against severe disease and/or death in vaccinated individuals or breakthrough infection. It is important to distinguish between association and causality.

>>Response: We agree that it is inaccurate to link preserved T cell recognition to immune correlates of protection, and do not make such direct claims. However, that 70-80% of the response is cross-reactive to Omicron is an accurate reflection of our data in Fig. 1f and h (see below). We have tempered the final sentence of the abstract to emphasise that the link with protection is uncertain, and added text to the Discussion.

“It remains to be determined whether well-preserved T cell immunity to Omicron contributes to protection from severe COVID-19, and is linked to early clinical observations from South Africa and elsewhere⁹⁻¹².”

Discussion: *“To date, immune correlates of protection from disease are not clearly defined and large-scale prospective studies testing both humoral and cellular responses would be necessary to evaluate correlates of protection and define the role of T cell responses in virological control.”*

1.14. It is unconventional and possibly incorrect to present fold-changes in omicron and ancestral spike median T cell frequency as shown in Fig. 1f,h and Extended Data Fig. 5. This does not have the equivalent functional meaning of a fold change in nAb ID50. It is reasonable to present a reduction in median frequency as in Fig. 1e,h. Please can the authors clarify the statistical test used in Fig. 1e for the paired analysis. Please can the authors consider removing Fig. 1f,g and simply comment on the fall in T cell frequency in the text.

>> **Response:** We are surprised by this comment, as fold change is relatively commonly used to compare the difference of many biological measurements (including flow cytometry assay data), between different experiment conditions or time points (Tarke, Cell Rep Med. 2021, PMC8249675; Smits, JCI, 2020, PMC6994124; Chew, ARHR, 2020, PMC7864091; Samson, Science Trans Med. 2017, PMC6276984; Ghoneim, Cell, 2017, PMC5568784). This type of calculation is merely a way to represent normalized differences between paired samples. In this study, we believe that showing both the raw data (Fig. 1e and g, where a Wilcoxon matched-pairs signed rank test was used, see legend to Fig. 1) and the fold change (Fig. 1f and h) is of value, as it illustrates both the range of T cell responses and quantifies the differences between ancestral and Omicron spike.

1.15. In Fig. 1 the T cell responses were measured approximately one month (range 21 to 64 days) after the most recent vaccine dose. As such, the data very much offers a best-case scenario of very recently vaccinated individuals. The authors should analyze T cell responses at later timepoints as well – much of the concern about breakthrough infection in vaccinees and associated drop in antibody neutralization has been in relation to individuals 6 months or more after their vaccine second dose when the antibody responses have waned.

>>**Response:** It has been demonstrated that, as for antibodies, vaccine-induced T cell responses decline over time, albeit more slowly. Goel et al (2021) estimated a half-life of 187d for the CD4 response after mRNA vaccination. Consistent with this, 6 months after mRNA vaccination in another study, spike-specific immunity had declined roughly 2-fold, but remained detectable and cross-reactive to the Delta variant (Woldemeskel et al, 2021). Thus, the detection of continued cross-reactivity with variants over time will be related to the durability of the T cell response. If T cell responses to vaccination decline, we can reliably predict that there will be a concomitant decline in the cross-reactivity to variants detected by short term *in vitro* assays, as employed in our study. However, the relevance of these results would be questionable, since recall memory responses *in vivo* are likely to expand rapidly upon viral infection and contribute to limiting viral replication. Rather, studies of Omicron breakthrough infection in individuals whose responses have declined since vaccination, would demonstrate whether low level memory T cell responses expand in response to challenge with Omicron. However, studying breakthrough infections is beyond the scope of our study, but we have added discussion to cover these important points raised by the reviewer.

Discussion (page 6): “We studied Omicron cross-reactivity of vaccine responses approximately one month after vaccination. Since T cell responses decline over time (albeit more slowly than antibodies), the detection of continued cross-reactivity with variants over time will be related to the durability of the T cell response. Recall memory responses in vivo are likely to expand rapidly upon viral infection and contribute to limiting viral replication. Studies of Omicron breakthrough infection in individuals whose responses have declined since vaccination would be of great interest, and would demonstrate whether low level vaccine memory T cell responses expand in response to challenge with Omicron.”

1.1. In Extended Fig 2 the authors explore neutralizing activity against Omicron using plasma from Pfizer or J&J vaccinees. The findings are in line with recent papers and preprints. The writing is imprecise in referring to ability to escape ‘the majority of SARS-CoV-2 antibodies.’ The precise meaning is unclear: the plasma analyzed would indeed have encompassed many thousands of different anti-S IgG sequences, but no analysis was reported at the level of individual antibodies.

>>Response: We have rephrased this sentence to refer to neutralising ability.

In Extended Data Fig. 2a,b the authors should consider removing the fold-change data as it is meaningless to compare fold changes in nAb ID50 and T cell frequencies side-by-side in this way. It’s like comparing apples with elephants! (a 10-fold drop in ID50 has widely understood and accepted implications in terms of potential protective correlates, whereas here a largely maintained T cell frequency to a peptide panel is measured on an unrelated scale and confers no protective implication).

>>Response: Our intention with this Figure was to join the dots *i.e.* to demonstrate that the loss of T cell recognition appears less profound than the decrease in neutralization to Omicron when measured in the same individual vaccinated participants. We make no claims in our results on the protective ability of the T cell response, and have emphasised this point in the discussion section.

“To date, immune correlates of protection from disease are not clearly defined and large- scale prospective studies testing both humoral and cellular responses would be necessary to evaluate correlates of protection and define the role of T cell responses in virological control.”

Also, please can the data for nAb ID50 in Extended Data Fig. 2b be plotted separately for prior infection and no prior infection as it is hard to follow as currently presented.

>>Response: We have adjusted Extended Data Fig. 2b and these groups are now plotted separately.

1.2. The data presented in Figure 2 is difficult to interpret for the methodological reasons cited above (lack of longitudinal previous infection history / vaccination history / differences in disease severity between groups / timing of the blood draw).

>>Response: We have provided all clinical details (previous infection history, vaccination history, disease severity and time of blood draw post vaccination or COVID-19 episode) for each participant included in this study in the new Extended Data Table 4.

1.3. Please can the authors clarify the number of non-responders in Fig. 2c,d.

>>Response: The frequency of responders has been added to Fig. 2c,d.

1.4. The Introduction cites refs 14-17 in support of the sensible tenet that T cell immunity is

likely to modulate COVID-19 severity; it might be useful to word this sentence with greater precision in terms of the specific content of 14-17.

>>Response: We have elaborated on this brief sentence in the introduction to provide more details on these studies (page 3):

“SARS-CoV-2-specific T cells play a role in modulating COVID-19 severity. A study of acute COVID-19 suggested, through combined measurement of CD4+ T cells, CD8+ T cells, and neutralizing antibodies in COVID-19, that co-ordination of these three arms of the adaptive response leads to lower disease severity¹⁴. A greater CD8+ T cell response in blood and highly clonally expanded CD8+ T cells in bronchoalveolar lavage were observed in convalescent patients who experienced mild or moderate disease compared to more severe disease¹⁵⁻¹⁶, and CD8+ T cells provided partial protective immunity in the context of suboptimal antibody titers in a macaque model¹⁷.”

1.1. The data in Figure 2 and associated extended data offer a potentially rich and informative dataset (if the issues described above can be resolved). T cell responses are compared between convalescent patients from the Wuhan Hu-1, beta, delta and omicron waves. There are many interesting points from these studies that may merit further analysis and discussion. In Figure 2C, it looks as if those infected in the Omicron wave have a T cell response to N that is more impaired than to S or M. It would be valuable to dissect this out at an epitope level.

>>Response: As mentioned by Reviewer #3, a comprehensive *in vitro* analysis, comparing the recognition of Omicron vs ancestral spike at a single epitope level, would require a considerable amount of time and a different study design. To further explain the potential impact of Omicron mutations on T cell cross-recognition, we have now included *in silico* analyses assessing 1) the immunogenicity of epitopes across conserved and variable regions of spike and 2) predicted HLA Class I restriction for Omicron variable epitopes. Please refer to Reviewer # 2 point 2.1).

1.2. In 2e, it looks as if Omicron-infected people respond very similarly to the ancestral or Omicron peptide pools. This offers a slightly different window onto the epitope conservation question from the data presented in Figure 1, producing a different answer?

>>Response: Figure 1 and 2 represent different scenarios. Figure 1 are individuals who have encountered primarily ancestral spike (through vaccination or prior infection or a combination of the two), and we measure their theoretical ability to cross-recognise Omicron spike should they encounter it. Figure 2 are those whose first encounter with spike is the Omicron version (since these are unvaccinated individuals with no record of prior symptomatic infection), and we measure their actual ability to respond to Omicron spike after encountering it. We agree that superficially this may read as identifying some loss of T cell response (albeit small) in Fig 1, compared to near complete conservation of the T cell response in Fig 2, which would seem to be slightly at odds. We would assert that Figure 2 rather corroborates the message of Figure 1: when we look at potential cross-reactivity with Omicron, it is mostly conserved; when we subsequently examine actual recognition of Omicron, we demonstrate conservation to an even greater degree.

To better highlight the distinction between what the two data sets are showing, we have included additional discussion, and reversed the order of the data on Fig 1e.

Discussion (page 5): "In this study, we measured the ability of individuals to cross-recognise Omicron spike should they become infected, following an exposure to a previous version of the viral antigen (primarily the ancestral spike) through vaccination or prior infection or a combination of the two. We also studied unvaccinated individuals who had no history of previous infection, and whose first encounter with spike was the Omicron version. ... While we assessed experimentally the potential to cross-recognise Omicron, we also measured the actual ability to mount a response to Omicron (Fig. 2e), demonstrating near-complete preservation of the response between Omicron and ancestral spike and corroborating our initial observations."

1.1. Extended data Fig 3 offers interesting analysis of polyfunctionality in the CD4 Omicron T cell responses, again showing largely conserved recognition and response capacity. However, the case for polyfunctionality as a proxy for T cell affinity had been overstated and no data has been presented showing affinity / avidity measurements in this manuscript.

>>Response: We have removed this sentence so as not to imply polyfunctionality is a proxy for T cell affinity.

Interpretation

1.2. The paper is timely and complements the recent wave of papers on diminished Omicron neutralization. The wording of the title is not precise. Referring to T cell responses that 'remain robust' is unscientific and misleading: the intention was perhaps to imply a strength of response that could be protective (as subsequently discussed). What the paper actually shows is that, set in the context of all T cell recognition that is intact and detectable, the loss of T cell recognition appears less profound than that shown with respect to antibody evasion.

>>Response: We have modified the title to more precisely reflect the results: "SARS-CoV-2 spike T cell responses induced upon vaccination or infection cross-recognize Omicron"

1.3. The Abstract ends on unsupported speculation: it's unclear why these immune findings should be specifically and causally related to clinical observations of mild Omicron disease in South Africa and elsewhere.

>>Response: We have modified this sentence so as not to imply a direct link between retention of T cell responses and lower clinical severity of Omicron infection.

"It remains to be determined whether well-preserved T cell immunity to Omicron contributes to protection from severe COVID-19, and is linked to early clinical observations from South Africa and elsewhere⁹⁻¹²."

Referee #2 (Remarks to the Author):

The manuscript of Keeton et al. aims to characterize the impact that Spike Omicron mutations exert on Spike-specific T cells induced by vaccinations with Adenovirus- or mRNA-based vaccines (n=55) or by infection (15 convalescents). Furthermore, the authors tested the ability of Omicron infection (in 19 patients) to induce T cells against Spike, Np and membrane (ancestral sequence) in comparison to other patients (~ 50 patients) infected by

other VoCs.

T cell data were generated by PBMC activation with peptide pools covering the Spike protein of the Wuhan strain and of the Omicron VoC and NP and membrane proteins of the Wuhan strain. Intracellular cytokine staining was used to evaluate the quantity of CD8 and CD4 T cells responding to the distinct peptide pools and their cytokine production (IFN-gamma, IL-2 and TNF-alpha).

The results show that Omicron Spike mutations do not suppress the majority of T cells induced by vaccines or infection and that Omicron infection triggers a T cell response against different structural proteins similar to other SARS-CoV-2 VoCs. These data are novel, important and timely.

2.1. In light of the fast world-wide diffusion of Omicron and of the profound impact that Omicron mutations have on the neutralizing ability of antibodies, knowledge related to the impact that these mutations have on T cells induced by vaccination is of great interest and importance. Having said this, the results presented are really only limited to the frequency of T cells stimulated by the different peptide pools. There are no attempts to define, for example, why T cells in the majority of vaccinated individuals tolerate Omicron mutations. Is this due to the fact that T cells are always targeting conserved regions of Spike, or it is because the mutations are present within the epitopes but they don't have any effect on HLA-class I and class II binding or in TCR recognition?

In normal circumstances, this reviewer would have thought that defining the epitopes and the restriction elements of the CD8 T cells that are completely inhibited by the Omicron mutations represent a logic and indispensable part of a research work investigating the effect of mutations on T cells. Such information is not only scholarly important but has practical consequences since, for example, this information could link specific HLA-class I profiles of individuals to increased susceptibility of prolonged Omicron infection.

It is however clear that definition of T cell epitopes and of their HLA-class I restrictions requires time that might severely delay the publication of the initial observation of the preserved T cell reactivity in the majority of vaccinated individuals.

The observation about the ability of Omicron infection to elicit a T cell response targeting structural proteins of SARS-CoV-2 of similar magnitude to what can be observed in individuals infected by others VoCs is novel and of interest. Clearly, also here it would have been nice to define a little better the quality of T cells in particular whether the T cells induced by Omicron infection target conserved regions.

>> **Response:** A comprehensive *in vitro* analysis, comparing the recognition of Omicron vs ancestral spike at a single epitope level, would require a considerable amount of time and a different study design. Thus, to alleviate this weakness, we have added *in silico* analyses to the manuscript showing that Omicron spike mutations occur preferentially in regions poorly targeted by CD4+ T cells, but are more common in regions frequently targeted by CD8+ T cells. This suggests that while immunogenic conserved regions should cross-react with Omicron, some mutations may lead to T cell escape (Extended Figure 7, see below).

Extended Fig. 7: Structure and distribution of spike SARS-CoV-2 epitopes targeted by CD4 and CD8 T cells.

a, Schematic of SARS-CoV-2 spike protein primary structure colored by domain. NTD: N-terminal domain, RBD: receptor binding domain, SD1: Sub-domain 1, SD2: Sub-domain 2. **b**, Distribution and frequency of recognition of confirmed CD4+ (top) and CD8+ T cell epitopes (bottom) across the entire spike protein. Data represent experimentally confirmed epitopes from the Immune Epitope Database and Analysis Resource (www.iedb.org).

Red lines depict the position of Omicron mutations that recorded a frequency of recognition

>10% and blue lines <10%.

Moreover, we also performed *in silico* analysis to define predicted HLA Class I restriction for Omicron variable epitopes. Our results show that six confirmed Spike epitopes containing Omicron mutations (A67V/del 69-70, G142D/143-145 del, S373P, S375F, D614G, P681H and N764K) would be detrimentally affected for binding to specific class I alleles. However, we also found another seven confirmed epitopes that contained Omicron mutations (T95I, S371L/S373P/S375F, K417N, G446S, Q493R, N764K, L981F) but had no impact on class I binding compared to the ancestral sequence. Overall, this suggests that while some Omicron mutations may mediate escape from specific HLA-restricted CD8+ T cells, not all mutations appear to have an impact on HLA class I binding (see Extended Data Table 4 below). These analyses have been included in the manuscript text (page 5).

	peptide (WT)	aa start	aa stop	Mutation	Peptide (Omicron)	MHC class-I allele	P rank (WT)	P rank (Omicron)	IC ₅₀ (WT)	IC ₅₀ (Omicron)
Detrimental	VTWFHALHV	62	70	A67V, del 69-70	VTWFHVI SG	HLA-A*02:11	0.64	18	18.91	8421.29
						HLA-A*68:02	0.26	9.3	45.32	8237.36
	GVYYHKNNK	142	150	G142D, 143-145 del	DHKNNK SWM	HLA-A*03:01	0.07	89	27.86	42735.87
	ASFSTFKCY	372	380	S373P, S375F	A FF STFKCY	HLA-A*30:02	0.06	2.8	43.85	2759.86
						HLA-B*15:03	0.26	2.5	25.16	495.99
	YQDVNCTEV	612	620	D614G	YQ G VNCTEV	HLA-A*02:131	0.53	1.3	49.38	149.98
						HLA-A*02:11	0.95	2	31.58	93.34
						HLA-B*07:02	0.02	2.9	4.65	4721.01
	SPRRARSVA	680	688	P681H	S H RRARSVA	HLA-B*07:05	0.03	3	4.51	2591.37
						HLA-B*07:06	0.03	3	4.51	2591.37
					HLA-B*42:02	0.03	3.6	38.64	8401.27	
QLNRALTGI	762	770	N764K	Q L KRALTGI	HLA-A*02:11	0.84	4	27.13	440.69	
Neutral	GVYFAST E K	89	97	Of note,	GVYFAST E K	HLA-A*11:01	0.03	0.02	9.66	6.98
						HLA-A*03:01	0.05	0.04	19.15	17.74
						HLA-A*68:01	0.02	0.06	4.4	8.51
	NSASFS T FK	370	379	S371L, S373P, S375F	N L A P FS T FK	HLA-A*34:02	0.02	0.01	16.17	11.26
						HLA-A*11:01	0.03	0.03	9.13	10.02
	YNSASFS T P	369	378	S371L, S373P, S375F	Y N L A P F ST P	HLA-B*15:03	0.05	0.18	6.63	18.49
	KIADY N YKL	417	425	K417N	K I ADY N YKL	HLA-A*02:02	0.07	0.17	6.58	11.64
						HLA-A*02:05	0.09	0.16	13.49	27.89
	VGG N YNYLY	445	453	G446S	V S G N YNYLY	HLA-A*30:02	0.06	0.02	45.32	20.21
	SKV G NYNY	443	451	G446S	S K V S GNYNY	HLA-B*15:03	0.26	0.17	25.38	17.47
	YFPLQ S YGF	489	498	Q493R	Y F PL R S S YGF	HLA-A*23:01	0.12	0.06	49.01	24.54
	GSF C TQLNR	757	765	N764K	G S F C TQL R	HLA-A*11:01	0.18	0.26	37.18	53.31
	QLNRAL T GI	762	770	N764K	Q L KRAL T GI	HLA-A*02:03	0.25	0.47	15.69	27.68
						HLA-A*02:02	0.04	0.05	4.16	4.36
	VLND I FSRL	976	985	L981F	V L ND I FS R L	HLA-A*02:03	0.06	0.08	5.19	6.38
					HLA-A*02:11	0.09	0.09	3.4	3.5	
					HLA-A*02:05	0.09	0.09	14.04	14.07	

New Extended Data Table 4: *In silico* analysis of the impact of Omicron mutations on epitope recognition by MHC Class I. Confirmed epitopes containing Omicron mutations are listed, together with their putative HLA class I restrictions. These were inferred using the Immune Epitope Database (IEDB) analysis resource (<http://tools.iedb.org/tepitool/>, NetMHCpan prediction method). Selected ancestral peptides with predicted a percentile rank (P rank) ≤ 1 and a IC₅₀ < 50 nM are shown, and the corresponding values for Omicron mutated epitope versions.

2.2. In addition, the discussion (and the abstract) depicts only a very positive scenario: a “well-preserved T cell immunity to Omicron”. Data in the majority of individuals tested support this conclusion, but the data reported also that in 5 of the 32 (15%) vaccinated individuals, Omicron mutations appear to completely abolish CD8 T cell recognition. 15% is not a completely insignificant number and I think the authors should point out that a quantifiable number of individuals might experience a loss of T cell response, which might have some virological (high and long viral spread?) and perhaps even pathological consequences. It might be time to show that biology is not only black or white but some grey exists.

>> **Response:** Thank you for this important point. We have extended the discussion on these data.

“It is noteworthy that Omicron mutations appear to abolish CD8 T cell recognition in 5 out of 32 participants (15%). These data are in agreement with a recent preprint (Naranbhai et al, 2021). It is thus possible that for some individuals, such loss of cross-reactive CD8+ T cell responses could have some virological and/or pathological consequences. Further analyses are required to define specific HLA-class I profiles and epitopes linked to loss of T cell responses.”

2.3. Limitations: the authors list some limitations, like the lack of utilization of an AIM assay to confirm and better define T cell responses and the single use of 15-mers. This is ok but perhaps, more related to the focus of the work that aims to define the impact of mutations on T cell responses, the authors should also list and discuss other potential limitations. First, they should point out that all their experiments were performed with a robust concentration of peptides (1ug/ml). The use of such high concentration might underestimate the impact of mutations on T cells. Second, it must be highlighted that mutations outside the epitopes might also have an effect on processing and presentation. As such, testing T cells only by utilizing peptides might underestimate the impact of mutations on the T cell response. These limitations should be included.

>> **Response:** We agree, and these limitations have now been included in the discussion (page 6).

“Moreover, the saturating concentration of peptide reagents used in these studies (1ug/ml) may underestimate the impact of mutations on T cells. In addition, the use of peptides does not permit us to define the potential effect of mutations on antigen processing and presentation, thus underestimating the impact of Omicron mutations on T cell cross- recognition.”

Referee #3 (Remarks to the Author):

Keeton et al studied immune responses towards the Omicron variant. The authors assessed T cell responses, in context of neutralizing antibodies, in individuals who were vaccinated with Ad26.CoV2.S and BNT162b2 as well as in unvaccinated convalescent COVID-19 patients. The main findings are that 70-80% of the spike-specific CD4+ and CD8+ T cell responses cross-react with the Omicron variant, similar to the cross-reactivity observed against Delta and Beta variants. This is a well performed study, manuscript is written clearly and the data are presented in a nice and accessible way. The study provides key insights into pre-existing T cell immunity to the Omicron variant, established either by prior vaccination or infection.

Specific comments:

3.1. Data presented in Fig 2cd show that a proportion of COVID-19 patients have not elicited T cell responses. What were the antibody levels in those patients? Did COVID-19 patients with non-detectable T cell responses seroconvert? Was the time post disease onset or disease severity different amongst patients who lacked T cell responses?

>> **Response:** For the hospitalised patients included in this study, SARS-CoV-2 antibodies (i.e. SARS-CoV-2 nucleocapsid-specific IgG) were measured only in first wave patients using the commercial Roche Elecsys® Anti-SARS-CoV-2 immunoassay (Roche Diagnostics, Basel, Switzerland). All patients were positive for N-specific IgG (with a median cut-off index [signal sample/cut-off] of 10.2, IQR: 4.1-75.3). No significant difference in the magnitude of N-specific Abs was observed between CD4 responders and non-responders ($p=0.24$). No antibody data were available for the patients from other waves.

Additionally, time post disease was comparable between CD4 non-responders (n=15) and responders (n=53) (median: 2.5 and 4.5 days, respectively, p=0.13). Finally, the proportion of patients with severe disease (i.e. WHO>5) was also comparable between CD4 responders and non-responders (38% vs 40%).

These results are now included in the text (page 4):

“The frequency of responders also did not differ markedly across the waves. Of note, we did not find any association between the absence of detectable CD4+ T cell responses and the time post COVID-19 diagnosis or disease severity.”

3.1. The authors should at least comment on the elevated antibody levels towards the Omicron variant following the third vaccine booster.

>> **Response:** We have included the following in the discussion:

“However, humoral responses can be enhanced upon booster vaccination, including the improvement of Omicron neutralization^{3,6,28,29}, further highlighting the importance of vaccine boosters.”

3.2. ‘CD4’ should be replaced with ‘CD4+’ and ‘CD8’ should be replaced with ‘CD8+’ T cells

>> **Response:** We have changed this throughout the manuscript.

3.3. The authors speculate that specific HLA molecules can be adversely affected by mutations in particular CD8 epitopes, which seems like a possible explanation. Were the participants HLA typed so this could be explored further?

>> **Response:** We have not performed HLA typing of the cohorts included in this study. This would indeed concretely address our speculation. We have recently shown this to be the case for mutations in the Beta variant for CD4 T cell recognition (Riou et al, Science Translational Medicine 2021), demonstrating that some epitopes restricted by specific HLA alleles could lead to T cell escape. To further explain the potential impact of Omicron mutations on T cell cross-recognition, we have added *in silico* analyses assessing 1) the immunogenicity of epitopes across spike; and 2) predicted HLA Class I restriction for Omicron variable epitopes. Please refer to Reviewer #2 point 2.1.

3.4. Extended Fig 1a: frequencies should be added to the FACS plots.

>> **Response:** Frequencies have been added to the plots.

3.5. Extended Fig 6a: should be ‘specific’ rather than ‘spe’.

>> **Response:** This has been corrected.

3.6. The authors used PepTivator, Miltenyi Biotech peptide pools. According to the Miltenyi website, not all Spike peptide pools cover the entire Spike protein. Please clarify whether the entire Spike protein was covered by the overlapping spike peptides. The authors state that they combined S1 peptide pool and “the majority of the C-terminal S2 domain”.

From Miltenyi website: “The PepTivator SARS-CoV-2 Prot_S covers selected immunodominant sequence domains of the spike protein (aa 304–338, 421–475, 492–519, 683–707, 741–770, 785–802, and 885–1273). In contrast, PepTivator SARS-CoV-2 Prot_S Complete covers all functional domains (aa 5–1273), Prot_S1 the complete N-terminal S1

domain (aa 1–692) and Prot_S+ parts of the C-terminal S2 domain (aa 689–895). The complete S2 domain (and parts of the S1 domain: aa 304–338, 421–475, and 492–519) is covered, when PepTivator SARS-CoV-2 Prot_S and Prot_S+ are combined.”

>>**Response:** The spike PepTivator pools we used for hospitalised patients have now been described in detail in the Methods section. We have corrected the text to read “near full-length Spike”, as the combination of pools that we used misses three small stretches of S2, namely aa 708-740, 771-784, and 803-884:

“For spike, we combined i) a pool of peptides (15-mer sequences with 11 amino acids (aa) overlap) covering the ancestral N-terminal S1 domain of SARS-CoV-2 (GenBank MN908947.3, Protein QHD43416.1) from aa 1 to 692 and ii) a pool of peptides (15-mer sequences with 11 aa overlap) covering the immunodominant sequence domains of the ancestral C-terminal S2 domain of SARS-CoV-2 (GenBank MN908947.3, Protein QHD43416.1) including the sequence domains aa 683-707, aa 741-770, aa 785-802, and aa 885-1273.

Reviewer Reports on the First Revision:

Referee #1:

In the rebuttal the authors concede that 15mer peptides are suboptimal to study CD8+ T cell responses.

1) The authors themselves cite a manuscript by Kiecker et al that reports, “Peptides of 15 amino acids length used at the same concentration (in microg/ml) stimulated CD8+ T cells somewhat less efficiently (on average 77% of the frequencies induced with the respective shorter peptides).”

The peptide pools used in any study should ideally be fit for the purpose of the study design being reported. The purpose of this study was to measure CD4+ and CD8+ T cell responses against omicron and ancestral spike. Using a 15mer peptide pool to study both CD4+ and CD8+ T cell responses against omicron compared to ancestral spike undermines somewhat differential conclusions about CD8+ T cell responses.

For example, “Both vaccination and infection induced spike-specific CD4+ T cell responses, while a CD8 response was less consistently detected (Fig. 1c).”

For example, “Similar results were observed for the CD8+ T cell response (Fig. 1g-h), where vaccinees who had received two doses of Ad26.COVID.S and convalescent donors demonstrated a significant decrease in the magnitude of Omicron spike-specific CD8+ T cells, although the other groups did not. There was a median reduction of 17-25% of the CD8 response to Omicron compared to the ancestral virus.”

The authors may wish to consider replacing the following sentence.....

‘The use of 15 mer peptides may also have underestimated SARS-CoV-2 specific CD8+ T cells as shorter peptides are more optimal for HLA class I binding’

With.....

‘The use of 15 mer peptides will have underestimated SARS-CoV-2 specific CD8+ T cells as 9-10mer peptides are optimal for HLA class I binding and it has been estimated that 15mer peptides capture 77% of the frequency of CD8+ T cells when compared to shorter peptidesref’

ref - Kiecker F, Streitz M, Ay B, Cherepnev G, Volk HD, Volkmer-Engert R, Kern F. Analysis of antigen-specific T-cell responses with synthetic peptides - what kind of peptide for which purpose? Hum Immunol. 2004 May;65(5):523-36.

2) "Overall, the limited effect of Omicron's mutations on the T cell response suggests that vaccination or prior infection may still provide substantial protection from severe disease." - This sentence is problematic. This study has not provided any data to support a causal link between T cell immunity against Omicron and protection from severe disease through prior infection and / or vaccination.

3) "Cross-reactive T cell responses acquired through vaccination or infection may be contributing to these apparent milder outcomes for Omicron." This sentence is problematic. Again there is no data in this manuscript looking at the role of T cell immunity in SARS-CoV-2 disease severity in the context of infection with the omicron variant to support this statement.

Referee #2:

The authors addressed my comments and modify the manuscript adding some of the limitations of their work.

The topic is of unquestioned interest since knowledge of vaccine and infection induced T cell response against Omicron are needed . The authors analyzed a good number of vaccinated and convalescent individuals and the results supported the conclusions.

Referee #3:

In the revised manuscript, the authors addressed my previous concerns.

Minor comment:

The authors included additional in silico data to show the effect of Omicron mutations on epitope recognition by HLAs (Extended Data Table 4). It would be helpful if the authors could comment on the immunogenicity and prominence of the epitopes defined as 'detrimentally' affected by the Omicron mutations. This would provide some insights into the importance of these mutations on T cell responses.

Author Rebuttals to First Revision:

Referee #1:

In the rebuttal the authors concede that 15mer peptides are suboptimal to study CD8+ T cell responses.

1) The authors themselves cite a manuscript by Kiecker et al that reports, "Peptides of 15 amino acids length used at the same concentration (in microg/ml) stimulated CD8+ T cells somewhat less efficiently (on average 77% of the frequencies induced with the respective shorter peptides)."

The peptide pools used in any study should ideally be fit for the purpose of the study design being reported. The purpose of this study was to measure CD4+ and CD8+ T cell responses against omicron and ancestral spike. Using a 15mer peptide pool to study both CD4+ and CD8+ T cell responses against omicron compared to ancestral spike undermines somewhat differential conclusions about CD8+ T cell responses.

For example, "Both vaccination and infection induced spike-specific CD4+ T cell responses, while a CD8 response was less consistently detected (Fig. 1c)."

For example, "Similar results were observed for the CD8+ T cell response (Fig. 1g-h), where

vaccinees who had received two doses of Ad26.COV2.S and convalescent donors demonstrated a significant decrease in the magnitude of Omicron spike-specific CD8+ T cells, although the other groups did not. There was a median reduction of 17-25% of the CD8 response to Omicron compared to the ancestral virus."

The authors may wish to consider replacing the following sentence.....

'The use of 15 mer peptides may also have underestimated SARS-CoV-2 specific CD8+ T cells as shorter peptides are more optimal for HLA class I binding'

With.....

'The use of 15 mer peptides will have underestimated SARS-CoV-2 specific CD8+ T cells as 9-10mer peptides are optimal for HLA class I binding and it has been estimated that 15mer peptides capture 77% of the frequency of CD8+ T cells when compared to shorter peptidesref

ref - Kiecker F, Streitz M, Ay B, Cherepnev G, Volk HD, Volkmer-Engert R, Kern F. Analysis of antigen-specific T-cell responses with synthetic peptides - what kind of peptide for which purpose? Hum Immunol. 2004 May;65(5):523-36.

>> **Response:** The text has been amended in the limitations section as requested by the referee and the article from Kiecker et al. has been added to the references, as follows:

"The use of 15 mer peptides will have underestimated SARS-CoV-2 specific CD8+ T cells as 9-10mer peptides are optimal for HLA class I binding and it has been estimated that 15mer peptides capture 77% of the frequency of CD8+ T cells when compared to shorter peptides"⁴⁰

2) "Overall, the limited effect of Omicron's mutations on the T cell response suggests that vaccination or prior infection may still provide substantial protection from severe disease." - This sentence is problematic. This study has not provided any data to support a causal link between T cell immunity against Omicron and protection from severe disease through prior infection and / or vaccination.

2.1) "Cross-reactive T cell responses acquired through vaccination or infection may be contributing to these apparent milder outcomes for Omicron." This sentence is problematic. Again there is no data in this manuscript looking at the role of T cell immunity in SARS-CoV-2 disease severity in the context of infection with the omicron variant to support this statement.

>> **Response:** The sentence suggesting a potential causal link between T cell response and disease severity has been removed and replaced by the following statement in the discussion:

“Overall, our data show that unlike neutralizing antibodies, the SARS-CoV-2 T cell response generated upon vaccination or prior infection are highly cross-reactive with Omicron. Early reports emerging from South Africa, England and Scotland have reported a lower risk of hospitalization and severe disease compared to the previous Delta wave⁹⁻¹². It remains to be defined whether cell-mediated immunity provides protection from severe disease and contributes to the apparent milder outcomes for Omicron.”

Referee #2:

The authors addressed my comments and modify the manuscript adding some of the limitations of their work.

The topic is of unquestioned interest since knowledge of vaccine and infection induced T cell response against Omicron are needed . The authors analyzed a good number of vaccinated and convalescent individuals and the results supported the conclusions.

>> No response required.

Referee #3:

In the revised manuscript, the authors addressed my previous concerns.

Minor comment:

The authors included additional in silico data to show the effect of Omicron mutations on epitope recognition by HLAs (Extended Data Table 4). It would be helpful if the authors could comment on the immunogenicity and prominence of the epitopes defined as ‘detrimentally’ affected by the Omicron mutations. This would provide some insights into the importance of these mutations on T cell responses.

>> **Response:** Extended data Figure 7 was modified to identify the peptides listed in Extended Table 4 on the “prevalence of recognition” graph. And the text has also been amended as follow:

"Six confirmed spike epitopes containing Omicron mutations (A67V/del 69-70, G142D/143-145 del, S373P, S375F, D614G, P681H and N764K) would be detrimentally affected for binding to specific class I alleles, four of which were located at a position that recorded a frequency of recognition greater than 10%. However, we also found another seven confirmed epitopes that contained Omicron mutations (T95I, S371L/S373P/S375F, K417N, G446S, Q493R, N764K, L981F) but had no impact on class I binding compared to the ancestral sequence, five of which were located at a position with a frequency of recognition greater than 10%."

Response to Reviewers – Round 1

Nature Manuscript 2021-12-20627, Keeton et al.

Referee #1:

In the paper called, "SARS-CoV-2 spike T cell responses induced upon omicron vaccination or infection remain robust against omicron", Keeton et al report CD4 and CD8 T cell responses against ancestral or Omicron spike peptide pools in donors who were either one or two dose J&J (Ad26-COV.S) or two dose Pfizer (BNT162b2) vaccinated, or convalescent after infection during the ancestral Wuhan Hu-1, Beta, Delta or Omicron infection waves in South Africa. This study is timely and attempts to address an important research question. However, I have some reservations about the cohorts studied, methodology, analysis and interpretation.

Study cohorts

1.1. Small study groups, incomplete and missing data about study groups, timing of sampling and missing control groups for interpretation of the T cell studies.

- The vaccine and Omicron infected convalescent cohorts shown in Fig. 1 have very small n numbers (Ad26-COV.S one dose, n=20, Ad26-COV.S two dose, n=20, BNT162b2 two dose, n=15 and unvaccinated convalescent, n=15).

- The wave 1 to 4 hospitalized cohorts shown in Fig. 2 have very small n numbers [wave (ancestral) 1, n=17, wave 2 (beta), n=16, wave 3 (delta), n= 16 and wave 4 (omicron), n=19].

>>**Response:** We evaluated a total of 138 patients in this study. Cross-reactivity to Omicron spike was measured in 55 vaccinees and 15 unvaccinated convalescents. When vaccinees were delineated according to vaccine type, results were remarkably consistent across the groups, with a similar median decrease in T cell frequencies to Omicron compared to ancestral peptide reagents. Since our submission, six preprints have been posted that concur with our results of well-preserved T cell responses against Omicron (using sample sizes comparable to ours), as follows:

Authors	Source	Date	Total	Sub-	n/group	Setting	Groups	Assay
---------	--------	------	-------	------	---------	---------	--------	-------

		posted	n	gps				
De Marco et al (Borsellino)	MedRxiv	30-Dec-21	61	5	10-15	EU	Vaccinated / Prior infection + vax / Vax + infection	AIM
Gao et al (Bruggert)	Research Square	03-Jan-22	103	3	15-40	EU	Vaccinated / convalescent / seronegative	AIM + ICS
GeurtsvanKessel et al (De Vries)	MedRxiv	29-Dec-21	100	8	5-15	EU	Vaccinated	AIM
Liu et al (Barouch)	MedRxiv	03-Jan-22	51	2	20-31	US	Vaccinated	Elispot + ICS
Naranbhai et al (Gaiha)	MedRxiv	05-Jan-22	76	3	11-31	US	Vaccinated / Prior infection + vax / Prior infection	Elispot + proliferation
Tarke et al (Sette)	BioRxiv	28-Dec-21	17	1	17	US	Vaccinated	AIM
OUR STUDY	MedRxiv	28-Dec-21	138	8	15-20	Africa	Vaccinated / Convalescent / Omicron-infected	ICS

We also compared 19 hospitalised patients in the Omicron wave to 49 hospitalised patients infected with other variants. It was of interest to delineate the patients according to variant infection, and despite differences in age, disease severity, co-morbidities and timing, there were broadly similar ranges of responses across the groups.

Thus, we assert that the internal consistency of our data, and independent corroboration from a sizeable number of emerging studies from different settings using different T cell assays, would argue that increasing the n would not alter our conclusions, and that our data are well-supported by the number of participants studied.

1.2. We are not told the vaccination history of participants included in the hospitalized cohorts. It seems from Supplementary Table 3 that these participants were recruited 2-6d after becoming PCR positive? Also, the Wave 2 (Beta) cohort have a much lower proportion of individuals with severe disease (WHO >5) (6% compared with 38-56%). Higher T cell responses are linked to more severe disease in hospitalized cohorts making the groups poorly matched for comparative analysis?

>>Response: We have now added a detailed Extended Data Table 5, with individual patient data that contains vaccination history and all other relevant details. All hospitalised patients from wave 1, 2 and 4 were unvaccinated. Third wave participants with known vaccination status were all unvaccinated (n=8), and the remainder (n=8) had unknown vaccination status. These details have been added to the Methods section (page 16).

With regard to T cell responses being linked to more severe disease: we have previously shown that the frequency of SARS-CoV-2-specific CD4 T cells in hospitalised patients was comparable irrespective of disease severity (defined based on the WHO ordinal scale) (Riou et al, JCI 2021 PMC8203446). Additional studies also present a more nuanced link between disease severity and T cell responses in hospitalised patients, which is influenced by multiple factors including co-morbidities and age (Sattler et al, JCI 2020 PMID:32833687; Neidleman et al, 2021 PMID:34260965; Nielsen et al, 2021 PMC8176920). The intention of the analysis in Figure 2 was to compare variant responses across waves in patients with a range of ages, disease severity, co-morbidities and time since diagnosis (which was not time since start of symptoms, which was not known for most patients).

1.3. An additional control group is missing and needed to determine if there is any pre-existing T cell immune recognition of the ancestral, beta, delta or omicron peptide libraries used. The

control groups required are:

- 1) Pre-pandemic controls (n=20) recruited before 2019 from the same geographic region who could not have been exposed to SARS-CoV-2
- 2) Uninfected, unvaccinated controls (n=20) recruited after March 2020 from the same geographic region.

>>**Response:** As stated by the reviewer, it is well-established that cross-reactive SARS-CoV-2 T cell responses are found in a proportion of the population who have not been exposed to SARS-CoV-2, from studies of pre-pandemic samples (Braun et al. 2020; Grifoni et al. 2020; Mateus et al. 2020). Indeed, we also identified ancestral Spike-specific responses in a minority of COVID-naïve individuals prior to vaccination (Keeton et al. 2021). These responses have been attributed to exposure to endemic human coronaviruses, and reactivity is more common for non-spike antigens than spike (Grifoni et al. 2020). Whilst we could perform such experiments, the addition of these control groups would add little value to the current study, and neither alter nor strengthen our conclusions.

1.4. The numbers in the vaccinated, convalescent and hospitalized study cohorts for Fig. 1 & 2 need to be increased and the timing of blood sampling for the hospitalised cohorts should be clearly stated in the manuscript text and need to be >14-21d after the acute SARS-CoV-2 infection.

>>**Response:** We have addressed the issue of increasing the numbers in point 1.1 above, and we have added timing of blood sampling and details at an individual patient level in the new Extended Data Tables 5 and 6). As mentioned, for hospitalised patients we have included time since diagnosis, which followed the onset of symptoms, the date of which was not known for most patients.

1.5. Missing or incomplete data to support the previous SARS-CoV-2 infection history during the first (Wuhan), second (Beta) and third (Delta) waves making interpretation of the results difficult

>>**Response:** We have addressed the issue of missing or incomplete clinical data in point 1.2 above, with the inclusion of a detailed Extended Data Table 5 including infection history, to better interpret the results (see Methods section, page 16).

1.6. The vaccination history should be included in Supplementary Table 3. Unvaccinated convalescent, vaccinated and hospitalized groups (wave 1-4) all require more detailed objective longitudinal data to support the participants previous SARS-CoV-2 infection, re-infection and breakthrough infection history up to December 2021. This could be presented in a Table format to support the definition of their previous infection history during the ancestral Wuhan first wave, beta second wave and delta third wave. The data currently presented in the Methods section and Supplementary Tables is incomplete. Without this information it is simply not possible to interpret the findings in terms of nAb and/or T cell responses. There have been several papers published showing augmented antibody and T cell responses following vaccination in the context of previous infection (including from the authors of this manuscript).

>>**Response:** We have addressed the issue of additional detail on the clinical data in point 1.2 above, with the inclusion of detailed Extended Data Tables 5 and 6, including vaccination and infection history, to better interpret the results.

1.7. This is especially important in the wave 4 (omicron) group in a geographical region (South Africa) where, as the authors state, >60% of the population have been previously infected with SARS-CoV-2 during the first to third waves. There needs to be additional evidence to support the lack of a previous SARS-CoV-2 infection history in this wave 4 (omicron) group in order to be able to interpret the results. For example, it has been proposed by others that the population in SA has been partly protected from severe disease by the relatively high levels of previous natural infection.

>>**Response:** All wave 4 participants included in our study were unvaccinated and had no clinical record of previous symptomatic COVID-19. These details are now included in the additional detailed Extended Data Table 5 provided and in the Methods section.

1.8. In the current wave 4 (omicron) group of n=19, 7 were identified by S gene target failure on PCR suggestive of omicron infection and further 5 had isolates confirmed by whole genome sequencing (but data is not included). The remaining 7 subjects were assumed to have omicron based solely on the timing of their infection. It would be relatively straightforward to recruit and additional PCR confirmed Omicron cases with longitudinal data to support a lack of previous infection to include in the analysis.

>>**Response:** We believe that it is reasonable to assume that the majority of patients recruited during the Omicron wave were infected with Omicron. A further two swabs were located and confirmed to be SGTF so we have amended the text to 9/19 SGTF, and this is regarded as an acceptable proxy for Omicron. We have removed the reference to the whole genome sequencing confirmation of 5 of the samples, since whilst they were assigned as Omicron by Nextclade, the assemblies had multiple frameshifts (poor sequencing quality) and were rejected by GISAID. Wave 4 did not occur with a concomitant Delta wave in South Africa as has occurred elsewhere, but was driven by the Omicron variant, and the prevalence of Omicron in South Africa at the time of recruitment was >90% (see Fig. 2b). Moreover, in Tshwane from where the remainder of the samples originated, Omicron was responsible for 98% of infections sequenced at the time of sampling in our study (determined from 60/61 samples sequenced). We have corrected the text and elaborated on this in the Methods (page 15).

Methodology

1.9. Overarching comment: the study draws on individuals with different medical histories, infections and vaccines and sampled at different times. Even referring to the supplementary materials, it was hard to track these between figures to allow the reader a proper chance to gauge key issues such as timing of infection and vaccination history.

>>**Response:** As referred to in point 1.1, 1.5 and 1.6, we have included Extended Data Tables 5 and 6 with individual participant details, including timing of infection and vaccination history.

The study utilises peptide libraries and flow cytometry to examine the T cell responses. The data from these T cell assays are interesting and informative. Additional information is required regarding the peptides contained in the libraries used. The individual peptides contained in all the libraries should be listed in the supplementary material. For omicron studies it is especially important to report details of peptide panels in some detail (aa sequence as well as length, overlap, number in the pool) as some of the S mutations are clustered together and a single peptide can carry more than one altered T cell epitope, or indeed, get spliced within a variant stretch.

>>**Response:** We have added an Extended Data Table indicating all individual peptide sequences for Ancestral, Beta, Delta and Omicron peptide libraries (Extended Data Table 7). The methods section includes additional detail of peptide panels (page 16):

“There were a total of 253 peptides in the Ancestral, Beta and Delta variant pool, and 254 peptides in the Omicron pool.”

1.10. The data reported for CD8 responses is problematic as the peptide libraries used in the study consisted of 15mers. Normally, to specifically study CD8 T cell responses peptide libraries containing peptides that are 9-10aa long are used and peptides that are 15-20aa long are used for CD4 T cell studies. This means that the data shown in Fig. 1g,h, Fig. 2d, Extended Data Fig. 4 and the RHS panels of Extended Data Fig. 5 & Extended Data Fig. 6 should either be removed, repeated using peptide libraries designed to specifically explore a CD8 T cell response.

>>**Response:** The use of 15mer peptides is an approach widely used to screen for CD8 T cell responses (a few examples cited below). Peptides of 15 amino acids in length are slightly less efficient at activating CD8 T cells than 9-10mers, but is a broadly acceptable compromise for reasons of cost and practicality (in particular, enabling CD4 and CD8 responses to be measured in the same sample given the large cell numbers required for cellular immunology assays). Kiecker et al. 2004 (PMID: **15172453**) estimated that 15mers capture 77% of the frequency of the CD8 response compared to shorter peptides. Thus, 15mer peptides still capture the majority of the CD8 response, exemplified by the abundant CD8 responses we detect in our study. The assertion that the CD8 data is invalid since we used 15mers and the request that we remove all CD8 data from the paper or repeat the study using 9-10mers is rather surprising and we believe quite an unreasonable request.

We did raise the issue of peptide length as a limitation in our discussion:

“The use of 15mer peptides may also have underestimated SARS-CoV-2-specific CD8 T cells, as shorter peptides are more optimal for HLA class I binding³⁹.”

A selection of the many studies that use 15mers to screen for CD8 responses:

Goel et al, Science 2021; Guerra et al, Science Immunology 2021; Barouch et al, NEJM 2021,

Alter et al Nature 2021; Cohen et al, Cell Reports Med 2021.

In addition, the six emerging studies on Omicron T cell cross-reactivity we referred to earlier also used 15mer peptide sets and assessed CD8 T cell responses (De Marco et al, 2021; Gao et al, 2021, GeurtsvanKessel et al, 2021; Liu et al, 2021, Naranbhai et al, 2021, Tarke et al, 2021).

1.11. The use of peptide libraries in the methodology means that the data as presented will by definition include T cell responses against cryptic epitopes never seen in real life settings. So while the peptide libraries used contain peptides of an appropriate aa length (ie 15mer) to demonstrate CD4 T cell responses, some of the peptide epitopes would not be naturally presented. It would, therefore, be important to repeat the CD4 T cell studies with spike protein (ancestral and containing the omicron mutations and deletions) using an ELISpot approach. In this way the spike will be processed by APCs and epitopes presented via the class II pathway to CD4 T cells.

>>**Response:** We used peptides to assess T cell responses and identify mutations in Omicron that affect HLA binding (epitope presentation) and T cell recognition (TCR contacts), whereas mutations in Omicron that affect peptide processing would only be identified with the use of whole protein antigens that have undergone appropriate intracellular processing. This is certainly an additional way in which Omicron mutations could affect T cells. However, not having data on the potential for Omicron mutations to affect processing does not minimise the important body of data on epitope recognition that we do present, addressing HLA binding and T cell recognition, and we feel that identifying potential processing mutations is beyond the scope of the current study. We have highlighted this point in the discussion, that further work could be performed to assess the range of ways Omicron's mutations may affect T cell responses. We have added the following in the Discussion section:

"In addition, the use of peptides does not permit us to define the potential effect of mutations on antigen processing, thus underestimating the impact of Omicron mutations on T cell cross-recognition".

Analysis

The data from the T cell assays are interesting and informative, drawn from analysis of the response to large peptide libraries of the whole ancestral, Delta and Omicron spike protein in the vaccine studies and Wuhan-1 SARS-CoV-2 N, M and S proteins in hospitalized patient studies.

1.12. The overall T cell response will be the summation of many T cell responses, in individuals of differing HLA types, responding or not responding to a proportion of many peptides. It is, therefore, not surprising and entirely predicted from the biology that the T cell response overall will be reduced, but will remain broadly intact overall. There are after all only 38 described mutations in the spike region of the Omicron VOC. This makes the analysis reported here (with incredibly small group numbers) hard to analyze as it is impossible to take account of the many

HLA differences between individuals and the impact that this has on their T cell responses. The study needs larger n numbers.

>>**Response:** We agree wholeheartedly with the reviewer, that the experimental data we present confirms what we would have predicted with respect to the effect of only 38 mutations in the Omicron spike. We have responded to the issue of increased n in points 1.1 and 1.4. We are heartened that six additional studies that have appeared as preprints since our submission have come to the same conclusion of preserved cross-reactivity to spike.

Thus, despite the (very likely) many HLA differences between individuals and groups that will shape the magnitude and specificity of their T cell responses, both in our study and those from North America and Europe, there is a remarkable consistency in the reduction in the response to Omicron compared to ancestral spike. This consistency, generated from 55 vaccinees and 15 convalescents, and the experimental demonstration of what would be predicted, argues that the number of participants in our study is sufficient to observe reliable changes in the cross-reactivity to Omicron.

1.13. In this context, it seemed perhaps unhelpful (or at least, unconventional) to invoke the very processed-data concept of '70-80% of the CD4 and CD8 T cell response to spike is maintained'. This could mislead those who are unfamiliar with T cell analysis and has virtually no meaning in terms of immune correlates of protection. We still know relatively little about immunodominant epitopes and even less about T cell correlates of protection. This study shows that there are T cell responses present, but there is no data presented here to show that the presence or absence of any such T cell response protects against severe disease and/or death in vaccinated individuals or breakthrough infection. It is important to distinguish between association and causality.

>>**Response:** We agree that it is inaccurate to link preserved T cell recognition to immune correlates of protection, and do not make such direct claims. However, that 70-80% of the response is cross-reactive to Omicron is an accurate reflection of our data in Fig. 1f and h (see below). We have tempered the final sentence of the abstract to emphasise that the link with protection is uncertain, and added text to the Discussion.

"It remains to be determined whether well-preserved T cell immunity to Omicron contributes to protection from severe COVID-19, and is linked to early clinical observations from South Africa and elsewhere⁹⁻¹²."

Discussion: *"To date, immune correlates of protection from disease are not clearly defined and large-scale prospective studies testing both humoral and cellular responses would be necessary to evaluate correlates of protection and define the role of T cell responses in virological control."*

1.14. It is unconventional and possibly incorrect to present fold-changes in omicron and ancestral spike median T cell frequency as shown in Fig. 1f,h and Extended Data Fig. 5. This does not have the equivalent functional meaning of a fold change in nAb ID50. It is reasonable to present a reduction in median frequency as in Fig. 1e,h. Please can the authors clarify the statistical test used in Fig. 1e for the paired analysis. Please can the authors consider removing Fig. 1f,g and simply comment on the fall in T cell frequency in the text.

>> **Response:** We are surprised by this comment, as fold change is relatively commonly used to compare the difference of many biological measurements (including flow cytometry assay data), between different experiment conditions or time points (Tarke, Cell Rep Med. 2021, PMC8249675; Smits, JCI, 2020, PMC6994124; Chew, ARHR, 2020, PMC7864091; Samson, Science Trans Med. 2017, PMC6276984; Ghoneim, Cell, 2017, PMC5568784). This type of calculation is merely a way to represent normalized differences between paired samples. In this study, we believe that showing both the raw data (Fig. 1e and g, where a Wilcoxon matched-pairs signed rank test was used, see legend to Fig. 1) and the fold change (Fig. 1f and h) is of value, as it illustrates both the range of T cell responses and quantifies the differences between ancestral and Omicron spike.

1.15. In Fig. 1 the T cell responses were measured approximately one month (range 21 to 64 days) after the most recent vaccine dose. As such, the data very much offers a best-case scenario of very recently vaccinated individuals. The authors should analyze T cell responses at later timepoints as well – much of the concern about breakthrough infection in vaccinees and associated drop in antibody neutralization has been in relation to individuals 6 months or more after their vaccine second dose when the antibody responses have waned.

>>**Response:** It has been demonstrated that, as for antibodies, vaccine-induced T cell responses decline over time, albeit more slowly. Goel et al (2021) estimated a half-life of 187d for the CD4 response after mRNA vaccination. Consistent with this, 6 months after mRNA vaccination in another study, spike-specific immunity had declined roughly 2-fold, but remained detectable and cross-reactive to the Delta variant (Woldemeskel et al, 2021). Thus, the detection of continued cross-reactivity with variants over time will be related to the durability of the T cell response. If T cell responses to vaccination decline, we can reliably predict that there will be a concomitant decline in the cross-reactivity to variants detected by short term *in vitro* assays, as employed in our study. However, the relevance of these results would be questionable, since recall memory responses *in vivo* are likely to expand rapidly upon viral infection and contribute to limiting viral replication. Rather, studies of Omicron breakthrough infection in individuals whose responses have declined since vaccination, would demonstrate whether low level memory T cell responses expand in response to challenge with Omicron. However, studying breakthrough infections is beyond the scope of our study, but we have added discussion to cover these important points raised by the reviewer.

Discussion (page 6): *"We studied Omicron cross-reactivity of vaccine responses approximately one month after vaccination. Since T cell responses decline over time (albeit more slowly than antibodies), the detection of continued cross-reactivity with variants over time will be related to the durability of the T cell response. Recall memory responses in vivo are likely to expand rapidly upon viral infection and contribute to limiting viral replication. Studies of Omicron*

breakthrough infection in individuals whose responses have declined since vaccination would be of great interest, and would demonstrate whether low level vaccine memory T cell responses expand in response to challenge with Omicron.”

1.16. In Extended Fig 2 the authors explore neutralizing activity against Omicron using plasma from Pfizer or J&J vaccinees. The findings are in line with recent papers and preprints. The writing is imprecise in referring to ability to escape ‘the majority of SARS-CoV-2 antibodies.’ The precise meaning is unclear: the plasma analyzed would indeed have encompassed many thousands of different anti-S IgG sequences, but no analysis was reported at the level of individual antibodies.

>>Response: We have rephrased this sentence to refer to neutralising ability.

In Extended Data Fig. 2a,b the authors should consider removing the fold-change data as it is meaningless to compare fold changes in nAb ID50 and T cell frequencies side-by-side in this way. It’s like comparing apples with elephants! (a 10-fold drop in ID50 has widely understood and accepted implications in terms of potential protective correlates, whereas here a largely maintained T cell frequency to a peptide panel is measured on an unrelated scale and confers no protective implication).

>>Response: Our intention with this Figure was to join the dots *i.e.* to demonstrate that the loss of T cell recognition appears less profound than the decrease in neutralization to Omicron when measured in the same individual vaccinated participants. We make no claims in our results on the protective ability of the T cell response, and have emphasised this point in the discussion section.

“To date, immune correlates of protection from disease are not clearly defined and large-scale prospective studies testing both humoral and cellular responses would be necessary to evaluate correlates of protection and define the role of T cell responses in virological control.”

Also, please can the data for nAb ID50 in Extended Data Fig. 2b be plotted separately for prior infection and no prior infection as it is hard to follow as currently presented.

>>Response: We have adjusted Extended Data Fig. 2b and these groups are now plotted separately.

1.17. The data presented in Figure 2 is difficult to interpret for the methodological reasons cited above (lack of longitudinal previous infection history / vaccination history / differences in disease severity between groups / timing of the blood draw).

>>**Response:** We have provided all clinical details (previous infection history, vaccination history, disease severity and time of blood draw post vaccination or COVID-19 episode) for each participant included in this study in the new Extended Data Table 4.

1.18. Please can the authors clarify the number of non-responders in Fig. 2c,d.

>>**Response:** The frequency of responders has been added to Fig. 2c,d.

1.19. The Introduction cites refs 14-17 in support of the sensible tenet that T cell immunity is likely to modulate COVID-19 severity; it might be useful to word this sentence with greater precision in terms of the specific content of 14-17.

>>**Response:** We have elaborated on this brief sentence in the introduction to provide more details on these studies (page 3):

"SARS-CoV-2-specific T cells play a role in modulating COVID-19 severity. A study of acute COVID-19 suggested, through combined measurement of CD4+ T cells, CD8+ T cells, and neutralizing antibodies in COVID-19, that co-ordination of these three arms of the adaptive response leads to lower disease severity¹⁴. A greater CD8+ T cell response in blood and highly clonally expanded CD8+ T cells in bronchoalveolar lavage were observed in convalescent patients who experienced mild or moderate disease compared to more severe disease¹⁵⁻¹⁶, and CD8+ T cells provided partial protective immunity in the context of suboptimal antibody titers in a macaque model¹⁷."

1.20. The data in Figure 2 and associated extended data offer a potentially rich and informative dataset (if the issues described above can be resolved). T cell responses are compared between convalescent patients from the Wuhan Hu-1, beta, delta and omicron waves. There are many interesting points from these studies that may merit further analysis and discussion. In Figure 2C, it looks as if those infected in the Omicron wave have a T cell response to N that is more impaired than to S or M. It would be valuable to dissect this out at an epitope level.

>>**Response:** As mentioned by Reviewer #3, a comprehensive *in vitro* analysis, comparing the recognition of Omicron vs ancestral spike at a single epitope level, would require a considerable amount of time and a different study design. To further explain the potential impact of Omicron mutations on T cell cross-recognition, we have now included *in silico* analyses assessing 1) the immunogenicity of epitopes across conserved and variable regions of spike and 2) predicted HLA Class I restriction for Omicron variable epitopes. Please refer to Reviewer # 2 point 2.1).

1.21. In 2e, it looks as if Omicron-infected people respond very similarly to the ancestral or Omicron peptide pools. This offers a slightly different window onto the epitope conservation question from the data presented in Figure 1, producing a different answer?

>>**Response:** Figure 1 and 2 represent different scenarios. Figure 1 are individuals who have encountered primarily ancestral spike (through vaccination or prior infection or a combination of the two), and we measure their theoretical ability to cross-recognise Omicron spike should they encounter it. Figure 2 are those whose first encounter with spike is the Omicron version (since these are unvaccinated individuals with no record of prior symptomatic infection), and we measure their actual ability to respond to Omicron spike after encountering it. We agree that superficially this may read as identifying some loss of T cell response (albeit small) in Fig 1, compared to near complete conservation of the T cell response in Fig 2, which would seem to be slightly at odds. We would assert that Figure 2 rather corroborates the message of Figure 1: when we look at potential cross-reactivity with Omicron, it is mostly conserved; when we subsequently examine actual recognition of Omicron, we demonstrate conservation to an even greater degree.

To better highlight the distinction between what the two data sets are showing, we have included additional discussion, and reversed the order of the data on Fig 1e.

Discussion (page 5): "In this study, we measured the ability of individuals to cross-recognise Omicron spike should they become infected, following an exposure to a previous version of the viral antigen (primarily the ancestral spike) through vaccination or prior infection or a combination of the two. We also studied unvaccinated individuals who had no history of previous infection, and whose first encounter with spike was the Omicron version. ... While we assessed experimentally the potential to cross-recognise Omicron, we also measured the actual ability to mount a response to Omicron (Fig. 2e), demonstrating near-complete preservation of the response between Omicron and ancestral spike and corroborating our initial observations."

1.22. Extended data Fig 3 offers interesting analysis of polyfunctionality in the CD4 Omicron T cell responses, again showing largely conserved recognition and response capacity. However, the case for polyfunctionality as a proxy for T cell affinity had been overstated and no data has been presented showing affinity / avidity measurements in this manuscript.

>>**Response:** We have removed this sentence so as not to imply polyfunctionality is a proxy for T cell affinity.

Interpretation

1.23. The paper is timely and complements the recent wave of papers on diminished Omicron neutralization. The wording of the title is not precise. Referring to T cell responses that 'remain robust' is unscientific and misleading: the intention was perhaps to imply a strength of response that could be protective (as subsequently discussed). What the paper actually shows

is that, set in the context of all T cell recognition that is intact and detectable, the loss of T cell recognition appears less profound than that shown with respect to antibody evasion.

>>**Response:** We have modified the title to more precisely reflect the results:

"SARS-CoV-2 spike T cell responses induced upon vaccination or infection cross-recognize Omicron"

1.24. The Abstract ends on unsupported speculation: it's unclear why these immune findings should be specifically and causally related to clinical observations of mild Omicron disease in South Africa and elsewhere.

>>**Response:** We have modified this sentence so as not to imply a direct link between retention of T cell responses and lower clinical severity of Omicron infection.

"It remains to be determined whether well-preserved T cell immunity to Omicron contributes to protection from severe COVID-19, and is linked to early clinical observations from South Africa and elsewhere⁹⁻¹²."

Referee #2:

The manuscript of Keeton et al. aims to characterize the impact that Spike Omicron mutations exert on Spike-specific T cells induced by vaccinations with Adenovirus- or mRNA-based vaccines (n=55) or by infection (15 convalescents). Furthermore, the authors tested the ability of Omicron infection (in 19 patients) to induce T cells against Spike, Np and membrane (ancestral sequence) in comparison to other patients (~ 50 patients) infected by other VoCs.

T cell data were generated by PBMC activation with peptide pools covering the Spike protein of the Wuhan strain and of the Omicron VoC and NP and membrane proteins of the Wuhan strain. Intracellular cytokine staining was used to evaluate the quantity of CD8 and CD4 T cells responding to the distinct peptide pools and their cytokine production (IFN-gamma, IL-2 and TNF-alpha).

The results show that Omicron Spike mutations do not suppress the majority of T cells induced by vaccines or infection and that Omicron infection triggers a T cell response against different structural proteins similar to other SARS-CoV-2 VoCs.

These data are novel, important and timely.

2.1. In light of the fast world-wide diffusion of Omicron and of the profound impact that Omicron mutations have on the neutralizing ability of antibodies, knowledge related to the impact that these mutations have on T cells induced by vaccination is of great interest and importance. Having said this, the results presented are really only limited to the frequency of T

cells stimulated by the different peptide pools. There are no attempts to define, for example, why T cells in the majority of vaccinated individuals tolerate Omicron mutations. Is this due to the fact that T cells are always targeting conserved regions of Spike, or it is because the mutations are present within the epitopes but they don't have any effect on HLA-class I and class II binding or in TCR recognition?

In normal circumstances, this reviewer would have thought that defining the epitopes and the restriction elements of the CD8 T cells that are completely inhibited by the Omicron mutations represent a logic and indispensable part of a research work investigating the effect of mutations on T cells. Such information is not only scholarly important but has practical consequences since, for example, this information could link specific HLA-class I profiles of individuals to increased susceptibility of prolonged Omicron infection.

It is however clear that definition of T cell epitopes and of their HLA-class I restrictions requires time that might severely delay the publication of the initial observation of the preserved T cell reactivity in the majority of vaccinated individuals.

The observation about the ability of Omicron infection to elicit a T cell response targeting structural proteins of SARS-CoV-2 of similar magnitude to what can be observed in individuals infected by others VoCs is novel and of interest. Clearly, also here it would have been nice to define a little better the quality of T cells in particular whether the T cells induced by Omicron infection target conserved regions.

>> **Response:** A comprehensive *in vitro* analysis, comparing the recognition of Omicron vs ancestral spike at a single epitope level, would require a considerable amount of time and a different study design. Thus, to alleviate this weakness, we have added *in silico* analyses to the manuscript showing that Omicron spike mutations occur preferentially in regions poorly targeted by CD4+ T cells, but are more common in regions frequently targeted by CD8+ T cells. This suggests that while immunogenic conserved regions should cross-react with Omicron, some mutations may lead to T cell escape (Extended Figure 7, see below).

Extended Fig. 7: Structure and distribution of spike SARS-CoV-2 epitopes targeted by CD4 and CD8 T cells.

a, Schematic of SARS-CoV-2 spike protein primary structure colored by domain. NTD: N-terminal domain, RBD: receptor binding domain, SD1: Sub-domain 1, SD2: Sub-domain 2. **b**, Distribution and frequency of recognition of confirmed CD4+ (top) and CD8+ T cell epitopes (bottom) across the entire spike protein. Data represent experimentally confirmed epitopes from the Immune Epitope Database and Analysis Resource (www.iedb.org). Red lines depict the position of Omicron mutations that recorded a frequency of recognition >10% and blue lines <10%.

blue lines <10%.

Moreover, we also performed *in silico* analysis to define predicted HLA Class I restriction for Omicron variable epitopes. Our results show that six confirmed Spike epitopes containing Omicron mutations (A67V/del 69-70, G142D/143-145 del, S373P, S375F, D614G, P681H and N764K) would be detrimentally affected for binding to specific class I alleles. However, we also found another seven confirmed epitopes that contained Omicron mutations (T95I, S371L/S373P/S375F, K417N, G446S, Q493R, N764K, L981F) but had no impact on class I binding compared to the ancestral sequence. Overall, this suggests that while some Omicron mutations may mediate escape from specific HLA-restricted CD8+ T cells, not all mutations appear to have an impact on HLA class I binding (see Extended Data Table 4 below). These analyses have been included in the manuscript text (page 5).

	peptide (WT)	aa start	aa stop	Mutation	Peptide (Omicron)	MHC class-I allele	P rank (WT)	P rank (Omicron)	IC ₅₀ (WT)	IC ₅₀ (Omicron)
Detrimental	VTWFHAIHV	62	70	A67V, del 69-70	VTWFHVIISG	HLA-A*02:11	0.64	18	18.91	8421.29
	GVIYHKNNK	142	150	G142D, 143-145 del	DKNNKNSM	HLA-A*68:02	0.26	9.3	45.32	8237.36
	ASFSTFKCY	372	380	S373P, S375F	APFFTFKCY	HLA-A*30:01	0.07	89	27.86	42735.87
	ASFSTFKCY	372	380	S373P, S375F	APFFTFKCY	HLA-A*30:02	0.06	2.8	43.85	2759.86
	ASFSTFKCY	372	380	S373P, S375F	APFFTFKCY	HLA-B*15:03	0.26	2.5	25.16	495.99
	YQDVNCTEV	612	620	D614G	YQGVNCTEV	HLA-A*02:131	0.53	1.3	49.38	149.98
	YQDVNCTEV	612	620	D614G	YQGVNCTEV	HLA-A*02:11	0.95	2	31.58	93.34
	SPRRARSVA	680	688	P681H	SHRRARSVA	HLA-B*07:02	0.02	2.9	4.65	4721.01
	SPRRARSVA	680	688	P681H	SHRRARSVA	HLA-B*07:05	0.03	3	4.51	2591.37
	SPRRARSVA	680	688	P681H	SHRRARSVA	HLA-B*07:06	0.03	3	4.51	2591.37
QLNRALTGI	762	770	N764K	QLKRALTGI	HLA-B*42:02	0.03	3.6	38.64	8401.27	
QLNRALTGI	762	770	N764K	QLKRALTGI	HLA-A*02:11	0.84	4	27.13	440.69	
Neutral	GVYFASTEK	89	97	Of note,	GVYFASIEK	HLA-A*11:01	0.03	0.02	9.66	6.98
	GVYFASTEK	89	97	Of note,	GVYFASIEK	HLA-A*03:01	0.05	0.04	19.15	17.74
	NSASFSTFK	370	379	S371L, S373P, S375F	NLAPFFTFK	HLA-A*68:01	0.02	0.06	4.4	8.51
	NSASFSTFK	370	379	S371L, S373P, S375F	NLAPFFTFK	HLA-A*34:02	0.02	0.01	16.17	11.26
	NSASFSTFK	370	379	S371L, S373P, S375F	NLAPFFTFK	HLA-A*11:01	0.03	0.03	9.13	10.02
	YNSASFSTF	369	378	S371L, S373P, S375F	YNLAPFFTF	HLA-B*15:03	0.05	0.18	6.63	18.49
	KIADYNYKL	417	425	K417N	NIADYNYKL	HLA-A*02:02	0.07	0.17	6.58	11.64
	KIADYNYKL	417	425	K417N	NIADYNYKL	HLA-A*02:05	0.09	0.16	13.49	27.89
	VGGNYNYLY	445	453	G446S	VSGNYNYLY	HLA-A*30:02	0.06	0.02	45.32	20.21
	SKVGGNYNY	443	451	G446S	SKVSGNYNY	HLA-B*15:03	0.26	0.17	25.38	17.47
	YFPLQSYGF	489	498	Q493R	YFPLRSYSF	HLA-A*23:01	0.12	0.06	49.01	24.54
	GSFCTQLNR	757	765	N764K	GSFCTQLKR	HLA-A*11:01	0.18	0.26	37.18	53.31
	QLNRALTGI	762	770	N764K	QLKRALTGI	HLA-A*02:03	0.25	0.47	15.69	27.68
VLNDILSRL	976	985	L981F	VLNDIFSRL	HLA-A*02:02	0.04	0.05	4.16	4.36	
VLNDILSRL	976	985	L981F	VLNDIFSRL	HLA-A*02:03	0.06	0.08	5.19	6.38	
VLNDILSRL	976	985	L981F	VLNDIFSRL	HLA-A*02:11	0.09	0.09	3.4	3.5	
VLNDILSRL	976	985	L981F	VLNDIFSRL	HLA-A*02:05	0.09	0.09	14.04	14.07	

New Extended Data Table 4: *In silico* analysis of the impact of Omicron mutations on epitope recognition by MHC Class I.

Confirmed epitopes containing Omicron mutations are listed, together with their putative HLA class I restrictions. These were inferred using the Immune Epitope Database (IEDB) analysis resource (<http://tools.iedb.org/tepitool/>, NetMHCpan prediction method). Selected ancestral peptides with predicted a percentile rank (P rank) ≤ 1 and a IC₅₀ < 50 nM are shown, and the corresponding values for Omicron mutated epitope versions.

2.2. In addition, the discussion (and the abstract) depicts only a very positive scenario: a “well-preserved T cell immunity to Omicron”. Data in the majority of individuals tested support this conclusion, but the data reported also that in 5 of the 32 (15%) vaccinated individuals, Omicron mutations appear to completely abolish CD8 T cell recognition. 15% is not a completely insignificant number and I think the authors should point out that a quantifiable number of individuals might experience a loss of T cell response, which might have some virological (high and long viral spread?) and perhaps even pathological consequences. It might be time to show that biology is not only black or white but some grey exists.

>> **Response:** Thank you for this important point. We have extended the discussion on these data.

“It is noteworthy that Omicron mutations appear to abolish CD8 T cell recognition in 5 out of 32 participants (15%). These data are in agreement with a recent preprint (Naranbhai et al, 2021). It is thus possible that for some individuals, such loss of cross-reactive CD8+ T cell

responses could have some virological and/or pathological consequences. Further analyses are required to define specific HLA-class I profiles and epitopes linked to loss of T cell responses."

2.3. Limitations: the authors list some limitations, like the lack of utilization of an AIM assay to confirm and better define T cell responses and the single use of 15-mers. This is ok but perhaps, more related to the focus of the work that aims to define the impact of mutations on T cell responses, the authors should also list and discuss other potential limitations. First, they should point out that all their experiments were performed with a robust concentration of peptides (1ug/ml). The use of such high concentration might underestimate the impact of mutations on T cells. Second, it must be highlighted that mutations outside the epitopes might also have an effect on processing and presentation. As such, testing T cells only by utilizing peptides might underestimate the impact of mutations on the T cell response. These limitations should be included.

>>**Response:** We agree, and these limitations have now been included in the discussion (page 6).

"Moreover, the saturating concentration of peptide reagents used in these studies (1ug/ml) may underestimate the impact of mutations on T cells. In addition, the use of peptides does not permit us to define the potential effect of mutations on antigen processing and presentation, thus underestimating the impact of Omicron mutations on T cell cross-recognition."

Referee #3:

Keeton et al studied immune responses towards the Omicron variant. The authors assessed T cell responses, in context of neutralizing antibodies, in individuals who were vaccinated with Ad26.CoV2.S and BNT162b2 as well as in unvaccinated convalescent COVID-19 patients. The main findings are that 70-80% of the spike-specific CD4+ and CD8+ T cell responses cross-react with the Omicron variant, similar to the cross-reactivity observed against Delta and Beta variants. This is a well performed study, manuscript is written clearly and the data are presented in a nice and accessible way. The study provides key insights into pre-existing T cell immunity to the Omicron variant, established either by prior vaccination or infection.

Specific comments:

3.1. Data presented in Fig 2cd show that a proportion of COVID-19 patients have not elicited T cell responses. What were the antibody levels in those patients? Did COVID-19 patients with non-detectable T cell responses seroconvert? Was the time post disease onset or disease severity different amongst patients who lacked T cell responses?

>> **Response:** For the hospitalised patients included in this study, SARS-CoV-2 antibodies (i.e. SARS-CoV-2 nucleocapsid-specific IgG) were measured only in first wave patients using the

commercial Roche Elecsys® Anti-SARS-CoV-2 immunoassay (Roche Diagnostics, Basel, Switzerland). All patients were positive for N-specific IgG (with a median cut-off index [signal sample/cut-off] of 10.2, IQR: 4.1-75.3). No significant difference in the magnitude of N-specific Abs was observed between CD4 responders and non-responders (p=0.24). No antibody data were available for the patients from other waves.

Additionally, time post disease was comparable between CD4 non-responders (n=15) and responders (n=53) (median: 2.5 and 4.5 days, respectively, p=0.13). Finally, the proportion of patients with severe disease (i.e. WHO>5) was also comparable between CD4 responders and non-responders (38% vs 40%).

These results are now included in the text (page 4):

“The frequency of responders also did not differ markedly across the waves. Of note, we did not find any association between the absence of detectable CD4+ T cell responses and the time post COVID-19 diagnosis or disease severity.”

3.2. The authors should at least comment on the elevated antibody levels towards the Omicron variant following the third vaccine booster.

>> **Response:** We have included the following in the discussion:

“However, humoral responses can be enhanced upon booster vaccination, including the improvement of Omicron neutralization^{3,6,28,29}, further highlighting the importance of vaccine boosters.”

3.3. ‘CD4’ should be replaced with ‘CD4+’ and ‘CD8’ should be replaced with ‘CD8+’ T cells

>> **Response:** We have changed this throughout the manuscript.

3.4. The authors speculate that specific HLA molecules can be adversely affected by mutations in particular CD8 epitopes, which seems like a possible explanation. Were the participants HLA typed so this could be explored further?

>> **Response:** We have not performed HLA typing of the cohorts included in this study. This would indeed concretely address our speculation. We have recently shown this to be the case for mutations in the Beta variant for CD4 T cell recognition (Riou et al, Science Translational Medicine 2021), demonstrating that some epitopes restricted by specific HLA alleles could lead to T cell escape. To further explain the potential impact of Omicron mutations on T cell cross-recognition, we have added *in silico* analyses assessing 1) the immunogenicity of epitopes across spike; and 2) predicted HLA Class I restriction for Omicron variable epitopes. Please refer to Reviewer #2 point 2.1.

3.5. Extended Fig 1a: frequencies should be added to the FACS plots.

>> **Response:** Frequencies have been added to the plots.

3.6. Extended Fig 6a: should be 'specific' rather than 'spe'.

>> **Response:** This has been corrected.

3.7. The authors used PepTivator, Miltenyi Biotech peptide pools. According to the Miltenyi website, not all Spike peptide pools cover the entire Spike protein. Please clarify whether the entire Spike protein was covered by the overlapping spike peptides. The authors state that they combined S1 peptide pool and "the majority of the C-terminal S2 domain".

From Miltenyi website: "The PepTivator SARS-CoV-2 Prot_S covers selected immunodominant sequence domains of the spike protein (aa 304–338, 421–475, 492–519, 683–707, 741–770, 785–802, and 885–1273). In contrast, PepTivator SARS-CoV-2 Prot_S Complete covers all functional domains (aa 5–1273), Prot_S1 the complete N-terminal S1 domain (aa 1–692) and Prot_S+ parts of the C-terminal S2 domain (aa 689–895). The complete S2 domain (and parts of the S1 domain: aa 304–338, 421–475, and 492–519) is covered, when PepTivator SARS-CoV-2 Prot_S and Prot_S+ are combined."

>>**Response:** The spike PepTivator pools we used for hospitalised patients have now been described in detail in the Methods section. We have corrected the text to read "near full-length Spike", as the combination of pools that we used misses three small stretches of S2, namely aa 708-740, 771-784, and 803-884:

"For spike, we combined i) a pool of peptides (15-mer sequences with 11 amino acids (aa) overlap) covering the ancestral N-terminal S1 domain of SARS-CoV-2 (GenBank MN908947.3, Protein QHD43416.1) from aa 1 to 692 and ii) a pool of peptides (15-mer sequences with 11 aa overlap) covering the immunodominant sequence domains of the ancestral C-terminal S2 domain of SARS-CoV-2 (GenBank MN908947.3, Protein QHD43416.1) including the sequence domains aa 683-707, aa 741-770, aa 785-802, and aa 885-1273."

nature portfolio